# Subseasonal-to-seasonal forecasts of Heat waves in West African cities

Cedric G. Ngoungue Langue[1,2], Christophe Lavaysse[2,3], and Cyrille Flamant[1]

[1]Laboratoire Atmosphères, Milieux, Observations Spatiales (LATMOS) - UMR 8190 CNRS/Sorbonne Université/UVSQ, 78280 Guyancourt, France.
[2]Université Grenoble Alpes, CNRS, IRD, G-INP, IGE,38000 Grenoble, France
[3]European Commission, Joint Research Centre (JRC), 21027 Ispra, VA, Italy

**Correspondence:** Ngoungue Langue Cedric Gacial (cedric-gacial.ngoungue-langue@latmos.ipsl.fr)

**Abstract.** Heat waves are one of the most dangerous climatic hazards for human and ecosystem health worldwide. Accurate forecasts of these dramatic events are useful for policy makers and climate services to anticipate risks and develop appropriate responses. Subseasonal forecasts are of great importance for actions to mitigate the human and health consequences of extreme heat. In this perspective, the present study addresses the predictability of heat waves at subseasonal time scale in West African cities over the period 2001-2020. Two types of heat waves were analyzed : dry and wet heat waves using 2-meter temperature (T2m) and wet bulb temperature (Tw) respectively. Two models that are part of the subseasonal to seasonal (S2S) forecasting project, namely the European Centre for Medium-Range Weather Forecasts (ECMWF) and the United Kingdom Meteorological Office (UKMO) models, were evaluated using two state-of-the-art reanalysis products, namely the fifth generation ECMWF reanalysis (ERA5) and the Modern-Era Retrospective analysis for Research and Application. The skill of the models to detect hot extreme events is evaluated using the Brier score. The models show significant skills in detecting hot days both for medium- and long-range forecasts (2- and 5-week lead times, respectively). The predictability of heat waves in the forecast models is assessed by calculating categorical metrics such as the hit-rate, the Gilbert score and the false alarm ratio (FAR). The forecast models show greater skills in predicting heat wave days compared to a baseline climatology, mainly for medium-range forecasts (two weeks lead time) in three climatic regions in West Africa, but the hit-rate values remain very low. Nevertheless, they are important in order to assess the predictability of heat waves in the tropical region which is characterized by a strong convective activity. We find that wet heat waves are more predictable than dry heat waves. On average, the False Alarm Ratio (FAR) is excessively high and tends to increase with the lead time. Only approximately 15% to 30% of the predicted heat wave days are actually observed for Week 5 and Week 2, respectively. This suggests that the models overestimate the duration of the heat waves with respect to ERA5. Although the models demonstrate skills on heat wave detection compared to a baseline climatology, the accurate forecasts of the intensity of heat waves remains challenging over all lead times.

## 1 Introduction

The impact of heat waves on different sectors, in particular the economy and health, makes them one of the most dangerous climate hazards globally (Perkins, 2015). Heat waves pose a significant threat to human health, as they cause discomfort and

stress to body temperature regulation. Increased atmospheric humidity can exacerbate heat stress and lead to deaths, particularly among vulnerable populations such as children and the elderly (Russo et al., 2017). Climatic projections show an increase in the frequency, intensity and duration of extreme temperatures over the next century and beyond [e.g. Kharin et al. (2007); Fischer and Schär (2010); Perkins et al. (2012)]. Under these warming conditions, the frequency of extreme events such as heat waves will increase. In the latest Intergovernmental Panel on Climate Change report (IPCC report 2023), the authors show that equatorial regions will be more affected by climate change than mid- and high- latitudes. In some equatorial regions such as sub-Saharan Africa, the predictability of heat waves remains poorly documented.

Heat waves are often associated with extreme dry and/or night-time temperatures. A Heat wave is defined as a period of marked unusual hot weather (maximum, minimum and daily average temperatures) over a region persisting at least three consecutive days during the warm period of the year based on local (station-based) climatological conditions, with thermal conditions recorded above given thresholds. Most often, heat waves are considered as a period of at least three consecutive days during which daily temperatures exceed a specific threshold [e.g. Perkins and Alexander (2013); Déqué et al. (2017); Barbier et al. (2018); Batté et al. (2018); Ngoungue Langue et al. (2023)]. Many factors can affect the definition of a heat wave, including the end-user sectors (human health, infrastructures, transport, agriculture) and also the climatic conditions of the regions (Perkins and Alexander, 2013). Therefore, there is no standard definition of a heat wave [e.g. Perkins (2015); Oueslati et al. (2017); Shafiei Shiva et al. (2019)]. Different thresholds, duration and indicators contribute to the divergence in the definition of heat waves (Smith et al., 2013). Heat waves can be defined from daily meteorological variables such as daily raw temperature [e.g. Batté et al. (2018); Lavaysse et al. (2018); Engdaw et al. (2022); Ngoungue Langue et al. (2023)], min, mean or max daily wet bulb temperature [e.g. Yu et al. (2021); Ngoungue Langue et al. (2023)] or heat stress indices [e.g. Robinson (2001); Fischer and Schär (2010); Perkins et al. (2012); Guigma et al. (2020); Ngoungue Langue et al. (2023)] using relative or absolute thresholds. Heat stress indices refers to indices resulting from a combination of some atmospheric variables useful to assess the human body comfort (wind speed, relative humidity, and incoming solar radiation) such as apparent temperature, Universal Thermal Comfort index, excess heat factor (EHF) and excess heat index (EHI) (McGregor et al., 2015). Some other authors use the daily anomalies of temperature to define heat waves [e.g. Stefanon et al. (2012); Barbier et al. (2018)].

The impact of heat waves on human activities and health increases the need for skillful and reliable climate forecasts on subseasonal to seasonal time scales in order to anticipate risks and develop appropriate responses (Lowe et al., 2016). There-fore, early warning systems are of crucial importance to provide information on the occurrence of such events. In general, early warning systems integrated shorter and medium-range forecasts of potential weather hazards. This type of forecast window refers to subseasonal time scale from 2 up to 6 weeks. The subseasonal range is highly relevant for actions aimed at mitigating the human and health consequences of extreme heat [e.g. White et al. (2017); Moron et al. (2018); Tompkins et al. (2019); Osman et al. (2023)]. Subseasonal forecasts are used to monitor the evolution of specific weather patterns that have been iden-tified in advance with seasonal forecasts. Subseasonal and seasonal forecasts (S2S) are of great importance for humanitarian services in order to build up a "Ready-Set-Go" early warning concept that allows early action to be taken before a potential disaster [e.g., Bazo et al. (2019); Lala et al. (2022); Domeisen et al. (2022)].

Lin et al. (2022) assessed the predictability of the unprecedented 2021 summer heat wave over Western North America using 10 subseasonal to seasonal forecast models. They focused on the predictability of anomalous temperatures at 2 metres a few weeks in advance to derive the occurrence of heat waves. The results showed that the 10 models were able to predict anomalous hot temperatures around 3 weeks in advance. Regarding the amplitude of temperature anomalies, the models were able to detect realistic values up to two weeks in advance. Katsafados et al. (2014) also assessed the predictability of the 2010 Russian heat wave at seasonal time scale using the National Center for Atmospheric Research Community Atmosphere Model version 3 (NCAR CAM3) and the latest ECMWF operational forecasts for validation. Their forecasting approach is based on the predictability of atmospheric blocking using temperature at 850 hPa and the geopotential height at 500 hPa. They found that only a few members reproduced the main features of the blocking system three months in advance. Most of the members predicted the blocking system in the wrong location and underestimated its duration. In West Africa, few studies have been conducted on heat wave predictability. Batté et al. (2018) evaluated heat wave forecasting at subseasonal and seasonal time scales using the Météo-France model in the S2S project and the Météo-France seasonal forecasting System 5 (MF5) respectively. To assess the skills of the models at seasonal time scale, they defined two indices from the Expert Team on Climate Change Detection and Indices (ETCCDI, (Omondi et al., 2014)) database,namely, the heat wave duration index (HWDI) and the number of heat waves per season (HWPP) using the apparent temperature and 2-meter temperature. Apparent temperature represents the temperature actually felt by humans, caused by the combined effects of air temperature, relative humidity and wind speed. At the subseasonaltime scale, they used the apparent temperature and T2m anomalies instead of the HWDI and HWPP. They found that at the seasonal time scale, the skills of MF5 to reproduce inter-annual anomalies of heat wave duration is limited at the grid point level because of the high spatial variability in the region, but it is significant when taking the average over the Sahel region [10°N-20°N, 10°W-20°E]. At subseasonal time, the skills of the model decrease beyond one week. Guigma et al. (2021) assessed the predictability of Sahelian heat waves at subseasonal time scale using the ECMWF extended long-range forecast system (ENS-ext), ERA5 and BEST gridded data for the evaluation. Their approach is based on forecasting the probability of occurrence of heat waves. They show that ENS-ext is able to forecast Sahelian heat waves with significant skill up to 2 weeks ahead; and with increasing lead time, wet heat waves are more predictable than dry events.

Most previous studies assess heat wave predictability using T2m anomalies and large-scale predictors such as temperature at 850 hPa and geopotential height at 500 hPa [e.g. Vitart and Robertson (2018); Katsafados et al. (2014)]. While this approach is adequate for obtaining information about the weather situation for the future days, it cannot provide useful information about the onset and duration of heat waves. In this study, we will adapt the method proposed by Lavaysse et al. (2019) when assessing the predictability of heat waves over Europe. This method is more robust because the occurrence and duration of heatwaves are assessed directly using daily minimum or maximum temperatures. This involves the computation of evaluation metrics to assess the skills of the forecasts.

The present study assesses the predictability of heat wave frequency and characteristics in West African cities over the period 1993 to 2020 using two subseasonalto seasonal (S2S) models namely, ECMWF and UKMO. To the author's knowledge, this work is the first of its kind in the region and represents a first assessment of heat wave predictability at subseasonal time

scale in West Africa. To carry out this study, we first analyze the representation of T2m and wet bulb temperature (Tw) in the forecast models with respect to the reanalysis data used as references (see Section 2). Secondly, we evaluate the models on the representation of extreme heat events. Finally, the predictive skill of the models in forecasting heat waves is assessed.

The remainder of this article is organized as follows: in section2, we present the region of study and the data used for this work; the description of the methodology is also provided. Section3 contains the main results of this study following the methodology presented in section 2.

## 2 Region, Data and Methods

### 2.1 Region of interest

This work is carried out in West Africa, covering an area between 5-20°N and 15°W-10°E, extending from the Atlantic coast to Chad and from the Gulf of Guinea to the southern fringes of the Sahara desert [Fig.1]. In April 2010, North Africa was affected by a severe heatwave, with daily maximum and minimum temperatures exceeding 40°C and 27°C respectively over a period of 5 days. This event was disastrous for the population and caused significant material damage. In May 2013, the Senegalese city of Matam, experienced an intense heat wave with temperatures sometimes reaching 50°C in the shade. The event was persistent both during the day and night, and it caused 18 deaths among the elderly people in 10 days. Mauritania also experienced a devastating heatwave in May 2013, with maximum temperatures exceeding 46°C, causing the death of more than 25 elderly people and children.

The climate of West Africa is mainly influenced by the West African monsoon, which regulates the rainy season and therefore affects agriculture. The West African climate is characterized by a short wet season followed by a long dry season. West Africa exhibits high climate variability at regional- and local- scale. The present study focuses on major cities in the coastal and continental parts of West Africa. The cities were grouped in three regions based on their location, climate variability and the evolution of heat waves characteristics in each region (Moron et al., 2016; Ngoungue Langue et al., 2023). The regions are structured as follows :

- Continental region (CO hereafter) englobes the cities of Bamako, Ouagadougou and Niger [Fig.1];

- Coastal atlantic region (AT hereafter) englobes the cities of Dakar, Nouakchott, Monrovia and Conakry [Fig.1];

- Coastal Guinean region (GU hereafter) englobes the cities of Yamoussoukro, Abidjan, Lomé, Abuja, Lagos, Accra, Cotonou and Douala [Fig.1].

### 2.2 Reanalysis products

African cities suffer from a critical lack of weather observation stations; and the few available are not well distributed in the regions. Therefore, reanalysis data appear as good candidates to overcome this issue. Reanalyses provide a numerical description of the recent climate by combining models with observations and are invaluable to numerous users around the

world (Hersbach, 2016). In this work, we use two state-of-the art reanalysis products namely the fifth-generation European Center for Medium-Range Weather Forecasts reanalysis (ERA5, (Hersbach et al., 2020)) and the Modern-Era Retrospective analysis for Research and Applications, version 2 (MERRA-2, (Gelaro et al., 2017)) from the National Oceanic Atmospheric Administration (NOAA); (in the following, we will use "MERRA" to refer to MERRA-2) as our references for the evaluation of the forecast models. ERA5 and MERRA are part of the most reliable reanalyses used in Africa regions especially on the monitoring of heat waves [e.g. Barbier et al. (2018); Ngoungue Langue et al. (2021); Engdaw et al. (2022)]. Since ERA5 is used to initialize the atmospheric component of the ECMWF extended reforecasts, one could argue that the evaluation of the skills of the forecast models using only ERA5 as reference is circular. Therefore, we also included MERRA as a second reference. It also allows for estimating the uncertainties of the reanalyses.

### 2.2.1 ERA5

The ERA5 reanalysis provides hourly estimates of various climate variables for the entire globe using 137 hybrid sigma levels up to 80 km above the surface (Hersbach et al., 2020). The original spatial resolution is 0.28125 degrees, interpolated to a regular 0.25°x 0.25° grid. ERA5 outputs are generated by the CY41r2 model cycle of the Integrated Forecast system (IFS) of ECMWF, which uses a ten-member ensemble of 4D variational data assimilation. The set of atmospheric variables used within the ERA5 database are the hourly 2-meter temperature (T2m) and hourly 2-meter dew point temperature (d2m) covering the period from 1 January 2001 to 9 February 2021. From these variables, daily T2m (max, min, mean) and wet bulb temperature (Tw) are derived. The dataset is accessed through the Climserv database of the Institut Pierre Simon Laplace (IPSL) server or the Copernicus Climate Data Store (CDS). The land-sea mask used in this study is obtained from the ERA5 reanalysis and is available on the CDS.

### 2.2.2 MERRA

MERRA, unlike ERA5, has a spatial resolution of 0.625°x0.5° and provides data on 42 standard pressure levels. It uses an upgraded version of the Goddard Earth Observing System Model, Version 5 (GEOS-5) data assimilation system and the Global Statistical Interpolation (GSI) analysis scheme of Wu et al. (2002). To ensure consistency in our analysis, we converted the MERRA data to a spatial resolution of 0.25°x0.25°, similar to ERA5 using a conservative first-order interpolation. In the MERRA database, we use the hourly T2m, hourly 2-meter specific humidity and the pressure field at the surface from 1 January 2001 to 9 February 2021 to calculate daily T2m(max,min,mean) and Tw. The MERRA dataset was also accessed through the Climserv database on the Institut Pierre Simon Laplace (IPSL) server.

## 2.3 Forecasts products

To bridge the gap between medium range weather forecasts and seasonal forecasts, in 2013 the World Weather Research program (WWRP) and the World Climate Research program (WCRP) jointly launched a 5-year research initiative called the Subseasonal to Seasonal (S2S) project (Vitart et al., 2017) which aims to improve forecast skill and understanding on

the subseasonalto seasonal timescale with special emphasis on high-impact weather events. The S2S project focuses on the risk of extreme weather conditions, including tropical cyclones, droughts, floods, heat waves and monsoonal rainfall (see http://www.s2sprediction.net/). In order to obtain more robust statistical results, we conducted our study over a 20-year period, specifically using hindcast data from 2001 to 2020. Among the twelve models involved in the S2S project, we evaluated the ECMWF and UKMO (United Kingdom Met Office) models because they are both available throughout the study period.

### 2.3.1 ECMWF forecasts

The extended-range ECMWF forecast model runs on the Integrated Forecast System (IFS) cycle CY47R3 released on October 10th, 2021. The native spatial resolution of the ECMWF model is Tco639 L137 (about 16 km) up to day 15 and Tco319 (about 32 km) after day 15, but the downloaded data are interpolated to a regular 0.25°x 0.25° latitude/longitude grid to match the resolution of ERA5 for evaluation. It contains 91 sigma levels from the surface to 80 km. ECMWF provides two types of outputs for the S2S program: real-time forecasts and reforecasts called "hindcasts". Real-time forecasts are forecasts for the coming days. Hindcasts are forecasts produced for past dates using the most recent version of the forecasting system, and allow analysis of how the current system would have performed, alongside a consistent dataset covering a longer time period for evaluation. ECMWF extended-range real-time forecasts are run with 51 ensemble members (50 perturbed and 1 unperturbed), while hindcasts are run with 11 members. In this study, we focus on hindcasts only. ECMWF extended-range hindcasts are produced twice a week, on Monday and Thursday at 00Z. This means that for each week a new set of hindcasts is produced to calibrate the real-time ensemble forecasts for Monday and Thursday of the following week using the latest version of the IFS. We only analyzed the hindcasts produced on Thursday. This is because we firstly want to carry out a multi-model analysis. According to a first investigation on the initialization dates of the hindcasts of different models, we found that most of the models were initialized on the same date as ECMWF (Thursday of each week) but did not cover the study period. The 11-member ensemble hindcasts start on the same day and month as the real-time forecast, but covering the last 20 years. In our case, the forecast year is 2021 and we focus on the previous 20 years from that date, and the hindcasts run from 0-46 days. The variables of interest in the ECMWF S2S are T2m(max,min) over the last 6 hours, daily average T2m and d2m from which the daily average Tw was derived. The data are open access and available on the S2S project website(https://apps.ecmwf.int/datasets/data/s2s-realtime-instantaneous-accum-ecmf/levtype=sfc/type=cf/).

### 2.3.2 UKMO forecasts

The UKMO model runs on the HadGEM3 GC2.0 model which simulates the uncertainties of the initial conditions using a lagged initialisation and the uncertainties of the model using a stochastic scheme. The native spatial resolution of the UKMO model is N216: 0.83°x0.56° (about 60 km at mid-latitudes). It contains 85 vertical levels from the surface to 85 km and 4 soil levels: level 1 (0 - 0.1 m), level 2 (0.1 - 0.35 m), level 3 (0.35 - 1 m) and level 4 (1- 3 m). Similar to ECMWF, UKMO provides to the S2S program real-time forecasts and hindcasts. The UKMO real-time forecast consists of a set of 4 members (3 perturbed members and 1 control member) run daily for a period of 60 days. The UKMO hindcasts are produced 4 times per month, on the $1^{st}$, $9^{th}$, $17^{th}$ and $25^{th}$, and cover a 24-year period from 1993 to 2016. We are aware that these initialization

dates are not the same as those of ECMWF, but we are interested in this work on the predictability of heat waves in a broad perspective, not on specific events. Prior to 2017, specifically on March 25th, the UKMO ensemble hindcasts were composed of 3 members per cycle (2 perturbed and 1 control). Since 2017, the number of members has increased from 3 to 7 (6 perturbed and 1 control). Our target period is going from January 2001 to February 2021, and as mentioned earlier, the UKMO hindcasts are not available after the year 2016. To solve this problem and get more robust statistical results, we recompose the products to obtain a new composite that covers the whole target period. The process applied is described in the following expression:

$$\mathbf{UKMO_{2001-2021} = Concatenation}\left[\mathbf{UKMO_{hindcast2001-2016}, UKMO_{realtimeforecast2017-2021}}\right] \tag{1}$$

The real time forecasts were extracted for the same days as the hindcasts initialization. In order to apply the concatenation over time between the re-forecasts and real time forecasts, the coordinates dimensions of the two datasets must be the same. As shown early, the number of ensemble members in UKMO re-forecasts and real time forecasts are completely different. Therefore, to meet this requirement, we reduced the number of ensemble members from 7 to 4 (1-control member and 3-perturbed members)in the re-forecasts to match the number of ensemble members in the real-time forecasts. We selected the three first perturbed members in the hindcasts over the 7 available. The UKMO forecasts analyzed in this work are launched for a 6-week duration. The variables extracted from the UKMO database are the same as those in ECMWF. A summary of the main differences between the two models (ECMWF, UKMO) is provided in [Table1].

## 2.4   Metrics

### 2.4.1   Estimation of temperatures at the city scale

Weather forecasts provide the evolution of atmospheric variables on a global scale, which implies the need to have data from observation stations to access information on a local scale. This is a major problem in areas where there are not enough weather stations to collect data, as is the case in African cities. Nevertheless, when observation stations are available in the region, access to the data collected remains difficult. To address this issue, downscaling methods can be employed. However, in this study, we study phenomena at the city scale, and the spatial resolution of the reanalyses (ERA5, MERRA) is too coarse for this purpose. Although the reanalysis scale is more representative of the spatial variability of a heat wave occurring in a city than an isolated local station, a validation analysis is needed on test stations in order to determine the best interpolation technique for estimating local temperature from reanalyses. Following the same approach as developed in Ngoungue Langue et al. (2023), local temperatures over the cities were derived from the reanalysis using the reanalysis grid point closest to the station that satisfies a land-sea mask (lsm) of at least 0.5 ([Table2] shows the lsm values of all cities considered in this study). The same technique was applied for the forecast models.

## 2.4.2 Heat wave detection

In the present study, two types of heat waves are investigated : dry and wet heat waves. Dry heat waves are mostly driven by incoming solar radiation and occurred during the day. The detection of dry heat waves is processed using maximum values of T2m (T2m_max) as indicator. The most lethal heat waves are due not only to high temperatures but also to the effect of humidity (Steadman, 1979a, b). Humidity is an important driver of wet heat waves. Wet heat waves is detected using minimum values of T2m (T2m_min) and mean Tw as indicators.

The computation of Tw is given by the following formula:

$$\mathbf{Tw = T * atan\left[A(Rh+B)^{\frac{1}{2}}\right] + atan\left[T+Rh\right] - atan\left[Rh-C\right] + D*(Rh)^{\frac{3}{2}}*\left[atan(E*Rh)\right] - F} \tag{2}$$

(Stull, 2011), (Rh is used in percentage, for example 40 for Rh=40%).

The computation of relative humidity (RH) varies depending on the available variables in the products. The first formula is applied for ERA5, and the second for MERRA reanalyses.

$$\mathbf{Rh = 100 * \frac{\exp\left(\frac{a*T_d}{b+T_d}\right)}{\exp\left(\frac{a*T}{b+T}\right)}} \tag{3}$$

(August, 1828; Magnus, 1844; Alduchov and Eskridge, 1996)

$$\mathbf{Rh = 0.263 * p * q * \left[\exp\left(\frac{17.67*(T-T_0)}{T-29.65}\right)\right]^{-1}} \tag{4}$$

(Ngoungue Langue et al., 2023)

$\mathbf{a} = 17.625, \mathbf{b} = 243.04, \mathbf{A} = 0.151977, \mathbf{B} = 8.313659, \mathbf{C} = 1.676331, \mathbf{D} = 0.00391838, \mathbf{E} = 0.023101, \mathbf{F} = 4.686035, \mathbf{T_0} = 273.16K$

Where $\mathbf{T}$(°C), $\mathbf{T_d}$(°C), $\mathbf{T_0}$(K), $\mathbf{p}$(hPa) and $\mathbf{q}$ are respectively the ambient temperature, dew-point temperature, reference temperature, pressure and specific humidity.

Daily maximum and minimum temperatures are computed respectively from maximum and minimum temperatures in the last 6 hours. This choice of the computation of the extreme daily values is made according to the forecast models outputs. Daily average wet bulb temperature is computed from hourly dew point temperature. This restriction to T2m_min, T2m_max and Tw is related to the atmospheric variables available on S2S outputs preventing us from computing more elaborated indices as in Ngoungue Langue et al. (2023). We considered a heat wave as a consecutive period of at least 3 days during which the daily temperatures exceed the calendar $90^{th}$ percentile threshold computed over the entire period for T2m_min, T2m_max or

240 Tw respectively [Fig.2]. The $90^{th}$ percentile is calculated for each calendar day of the year over the study period independently for the reanalyses and the forecast models. In general, model hindcasts (ECMWF,UKMO) are run at least every week. For each model run, the $90^{th}$ percentile threshold is calculated individually for each lead time on a daily basis. For example, using ECMWF, the daily climatological $90^{th}$ percentile is calculated over the study period separately for hindcasts run every Thursday of the month (see [Fig.S1] in supplement material). Heat wave detection and model evaluation are carried out separately

for the different initialization dates by calculating some statistical metrics (see section 2.4.5). The metrics are then grouped by week (from week 1 to week 6) for each month and the average score per week is calculated. The choice of a relative threshold is more appropriate as it is easily replicable in other regions. When two heat wave events are separated by one day with an indicator value below the daily $90^{th}$ percentile, they are pooled together to form a single event [Fig.2].

### 2.4.3 Heat wave characteristics

After the detection of a heat wave, some important characteristics are deduced, namely the duration and the intensity. They are useful to investigate the severity of an event. The predictability of heat waves is assessed for occurrence, duration and intensity. To determine the occurrence and duration of heat waves, we create individual boolean files from the T2m_min, T2m_max and Tw time series at each grid point, which is equal to 1 if it is a hot day and 0 otherwise. This operation is performed on a daily time scale over the study period. Hot days are days on which the values of T2m_min, T2m_max or Tw are above the daily

$90^{th}$ percentile thresholds. In order to assess the characteristics of heat waves, only hot days belonging to heat wave sequences are considered (Ngoungue Langue et al., 2023). Boolean files are calculated separately for reanalyses and forecasts in order to assess the representation of heat wave occurrence and duration.

The intensity of a heat wave was defined as the sum of the daily exceedances of the indicators values to the climatological threshold during the event (see Ngoungue Langue et al. (2023) for more details). This study is in the framework of the project

Agence National de la Recherche STEWARd (STatistical Early WArning systems of weather-related Risks from probabilistic forecasts, over cities in West Africa) project which focuses on the human impacts of climate extremes. We are therefore interested in heat waves, which can be harmful to human health. To do so, the climatological daily threshold is chosen to be constant over the whole period for the computation of heat waves intensity. It is defined as the minimum of the daily climatology $90^{th}$ percentile over the study period. This approach allows us to properly assess the severity of a heat wave and

its potential human impacts, therefore, most dangerous heat waves will have higher intensity values.

### 2.4.4 From probabilistic to deterministic forecast

Ensemble forecasting is a tool used for making probabilistic weather forecasts. It is both an alternative and an indispensable complement to deterministic weather forecasts. Ensemble forecasting is of major interest because it provides different scenarios of the evolution of the state of the atmosphere. The particularity of ensemble forecasting is that, unlike deterministic forecast-

270 ing, many trajectories are simulated in order to take into account uncertainties in the physical component of the model, the chaotic nature of the atmosphere, the observation network and imperfect initial conditions of the forecasts. Although ensemble

forecasting has many advantages over deterministic forecasting, the evaluation of the performance of an ensemble forecasting model remains quite complex due to the amount of information available. Statistical analyses can be performed by considering only the mean of the members, the median member, the warmest and coldest members, the 1st and 3rd quartiles. In order to extract deterministic information from a probabilistic system for the calculation of prediction scores, we defined an approach based on member thresholds inspired by Lavaysse et al. (2019). Three threshold values were tested (see [Table3]) to optimize the ensemble forecast system based on results found in Lavaysse et al. (2019).

### 2.4.5 Skill scores

The evaluation of the skills of the models to predict heat waves is carried out from January to December to cover the wet and dry seasons in the region. The results were then splitted into four sub-seasons to assess the intra-seasonal variability of the skills of the models. The representation the evolution of T2m, Tw and extreme events in the forecasts is investigated using probabilistic scores such as the Continuous Rank Probability Score (CRPS) and Brier Score (BS).

The CRPS is a quadratic measure of the difference between the forecast and reanalysis cumulative distribution functions (CDFs); it quantifies the relative error between forecasts and observations. The values of the CRPS range from [0 to 1], the closer it is from 0, the better the forecast. The CRPS is calculated using the following formula:

$$\mathbf{CRPS} = \int_{-\infty}^{+\infty} \left( \mathbf{P_f(x)} - \mathbf{P_o} \right)^2 \mathbf{dx} \tag{5}$$

$\mathbf{P_f}, \mathbf{P_o}$ represent the forecast and reanalysis CDFs respectively.

The BS measures the difference between the probability of a forecast and the outcomes; it is a good metric to quantify the accuracy of a forecast system. The BS is given by :

$$\mathbf{BS} = \frac{1}{\mathbf{N}} \sum_{i=1}^{\mathbf{N}} \left( \mathbf{P_i} - \mathbf{O_i} \right)^2 \tag{6}$$

$\mathbf{P_i}, \mathbf{O_i}$ are respectively the forecast probability and observed outcome (1 if the event occurs and 0 if not). N is the length of the forecasts or observations time series.

The predictability of heat waves in the forecast models is evaluated using dichotomous scores based on the coherence between reanalysis and forecasts such as hits, false alarms, misses and correct rejections. Let consider a 2x2 contingency table [Table4] from which 3 metrics have been computed : hit-rate, FAR and GSS. The hit-rate indicates the percentage of observed heat waves that have been correctly forecasted. The False Alarm Ratio (FAR) gives the percentage of forecasted events that did not occur. The Gilbert Skill Score (GSS) measures the fraction of observed events that are correctly predicted, adjusted for hits associated with random chance (close to the climatology). Chance hits (CH) are given as the event frequency multiplied by the number of event forecasts. The GSS takes in account the hits, misses, false alarms and neglects the correct rejections that

would artificially improve the score. The values of the GSS range from $[-\frac{1}{3}$ to 1]; a GSS=0 indicates no skill while a GSS=1 perfect skill. The GSS is calculated using the following formula:

$$\mathbf{GSS} = \frac{\mathbf{hits} - \mathbf{CH}}{\mathbf{hits} + \mathbf{false\_alarms} + \mathbf{misses} - \mathbf{CH}} \tag{7}$$

Where CH is given by:

$$\mathbf{CH} = \frac{(\mathbf{hits} + \mathbf{false\_alarms})(\mathbf{hits} + \mathbf{misses})}{\mathbf{hits} + \mathbf{false\_alarms} + \mathbf{misses} + \mathbf{correct\_rejections}} \tag{8}$$

### 2.4.6 Heat waves forecast strategy

The assessment of heat wave predictability in the forecast models is a very difficult task. To address this concern, previous studies [e.g. Vitart and Robertson (2018); Batté et al. (2018)] have used T2m anomalies and large-scale atmospheric variables such as the geopotential at 500hPa for example, to predict their occurrence a few days in advance. In this study, we adapted the methodology developed by Lavaysse et al. (2019) to evaluate the predictability of heat waves in the forecasts. This assessment is done in two steps:

– The first approach is to determine the predictability of heat wave days in the forecasts with respect to the reanalyses. To do this, dichotomous scores (hits, false alarms, misses and correct rejection) have been calculated for each day at different lead times from which the hit-rate, FAR and GSS have been derived. The scores are calculated independently for each week and each city. This type of information is useful for early warning systems such as the STEWARd project.

– The second approach focuses on the predictability of the whole heat wave rather than hot days in the heat wave. Therefore, a heat wave occurring in the reanalyses is considered as predicted by the forecast models if at least one hot day of the event is correctly predicted. To facilitate the evaluation, dichotomous scores are now calculated for each week, and the hit-rate, FAR, GSS are derived. For example, if a heat wave is detected in the reanalysis during the first week, and a hot day within this heat wave is correctly predicted by the forecast model, then the heat wave is considered as predicted and the hit=1 and miss=0. We repeat the process over all weeks and calculate the hit-rate.

## 3 Results

### 3.1 Climatology evolution of some atmospheric variables (T2m and Tw) in the forecasts and reanalyses

The first assessment focuses on the representation of the climatological evolution of two key variables for this study, T2m and Tw in the forecasts compared to the reanalyses. To do so, the seasonal climatological bias between the forecasts and

the reanalyses is computed over the 20-year period. The bias is computed for each couple of models (ECMWF/UKMO) and reanalysis (ERA5/MERRA) using T2m(min,max) and Tw values [Fig.3] and [Fig.4] respectively). First, we analysed the bias between the forecast models and the ERA5 reanalysis using T2m_min values.The range of biases associated with T2m spans from -4 K to 4 K. Both models show a negative bias over the Sahel region which is more pronounced with UKMO [Fig.3(i)].

We observed in UKMO, a progressive shift of this strong negative bias over the northern Sahel during the season from winter to summer [Fig.3 (i)(e-g)]. UKMO exhibits a positive bias with respect to ERA5 over the Atlantic Ocean that decreases considerably in summer. The negative bias found in ECMWF over the Sahel region is consistent with previous global studies [e.g. Johnson et al. (2019); Haiden et al. (2021)]. The results obtained with T2m_max values are similar to those found with the T2m_min for both models. We notice especially a strong positive bias over the Atlantic coast during the winter and spring

[Fig.3 (ii)(a-b),(e-f)]. The evaluation of the representation of Tw in the forecasts shows strong negative bias with respect to the reanalyses over the whole Sahel region [Fig.4]. Compared to the results found with T2m, this strong negative bias can be related mainly to the underestimation of humidity in the models in this region. The assessment of the bias between the forecast models and MERRA on the evolution of T2m highlights significant discrepancies with the results obtained using ERA5 as a reference. Using T2m_min, we observed a positive bias in ECMWF with respect to MERRA over the Sahel and Guinea region during

winter and autumn which tends to decrease during spring and summer. These results highlight the uncertainties between the two reanalyses already discussed in Ngoungue Langue et al. (2023). It can be deduced from these results using T2m_min that ERA5 has a warmer trend than MERRA over the Sahel region, while being cooler than MERRA over the Atlantic Ocean (see [Fig.S2] in supplement material). The spatial distribution of T2m in the forecast models shows significant biases with respect to the reanalyses. In order to determine whether these biases change over time, we analyze the spatio-temporal evolution of

T2m and Tw. To do this, we calculate the daily climatological biases between forecasts and reanalyses over the CO, AT and GU regions at different lead times from week1 to week6. We observed a high spatial variability of the biases in the three regions. The models show smaller biases in the AT region in spring, of the order of +/- 0.25 K and larger biases, of the order of +/- 2 K are found in the CO region with T2m_min [Fig.S3]. These significant biases observed in the CO region are considerably reduced in summer. The results obtained with T2m_max are quite similar to those obtained with T2m_min (not shown). As

observed in the previous results with Tw [Fig.4], the models show lower negative biases in the CO region in winter ([Fig.S4] in supplement). The range of biases associated with Tw spans from -14 K to 0 K. We do not observe any systematic increase in these biases from Week1 to Week6 ([Fig.S3] and [Fig.S4] in supplement). This first assessment of the evolution of T2m and Tw in the models compared to the reanalyses reveals significant biases in the models which may lead to poor predictive skills.

## 3.2 Assessment of the predictive skills of T2m and Tw forecasts

In this section, we carry out a global assessment of the predictive skills of the models on the representation of T2m and Tw at different time scales. To do this, we analyzed the interannual variability of the CRPS calculated between forecasts and reanalyses over the 3 regions for T2m_min, T2m_max and Tw. The forecasts used in this work are launched for a 6-week duration; for each week, the median CRPS was calculated from 2001-2020 using ERA5 and MERRA. The CRPS results are in the same range for the first two weeks (Week1, Week2), the two intermediate weeks (Week3, Week4) and the last two

360  weeks (Week5, Week6) (see [Fig.S5] in supplement). According to these findings, we have chosen to organize our results into medium-range forecasts (Week2) and long-range forecasts (Week5). Week 5 was chosen for the long-range forecasts instead of week 6 because from week 5 onwards, the models generally reach the predictability horizon and are closer to the climatology. This organization of the results applies to the rest of the study. Firstly, the results of the CRPS obtained using ERA5 as reference are analyzed [Fig.5 (i)]. We have noticed that the skill of the models does not improve necessarily with decreasing lead time.

This could be related to systematic biases (the representation of atmospheric circulation, local scale processes) in the forecast models. The CRPS score shows a high spatial variability in the 3 regions, indicating a high dependance of the skill of the models with the region. The forecast models show better predictive skills in the AT region during all the seasons and lead times with T2m_min except for UKMO during summer. The CRPS associated with T2m_max indicates that ECMWF is more skillful in the GU region (it is also the case with UKMO except during Autumn when the AT region seems to be more predictable).

In general, the models present higher skills for T2m_min than T2m_max over the AT region. ECMWF generally shows more skill than UKMO. The evolution of the CRPS calculated using MERRA as a reference is similar to that found using ERA5 (not shown). As we previously observed a significant negative bias in the Sahel region with Tw, we expect high values of CRPS in the different regions. This is indeed the case, with CRPS values ranging from 4 to 13, i.e. 6 times higher than those obtained with T2_min or T2m_max. The CRPS values are drastically reduced in the CO region in winter, which is consistent with the

previous results in section 3.1 (see [Fig.5 (ii)] in supplement). The models show more predictive skill on the representation of T2m than Tw. This is not surprising, given that Tw combines T2m and humidity, which is very difficult to predict.

### 3.3 Extreme temperature values and heat waves in the models and reanalyses

#### 3.3.1 Extreme temperatures

After the evaluation of the forecast models on the representation of T2m and Tw, we focus now on the extreme heat events

referred here as hot days (see section 2.4.3). This is done through the calculation of the Brier score between the forecasts and reanalyses (ERA5, MERRA). In order to compute the Brier score, the raw forecasts and reanalyses are converted to boolean data, where a value of 1 represents a hot day and a value of 0 indicates no hot days. This approach based on a relative threshold (see section 2.4.3) will contribute to partially correct the biases previously found in the models. By using a percentile-based approach, we are not focusing on the intensity of extreme events but on the number of events above the threshold.

We firstly evaluated the Brier score between the forecasts and ERA5 for T2m and Tw. The range of values for the Brier score is between 0.05 to 0.175, indicating that the models are able to detect hot days in agreement with the reanalyses over the three regions for medium and long range forecasts (Week2,Week5) [Fig.6]. We found with T2m that in spring the models are more skillful and present less spatial variability compared to other seasons [Fig.6 (i)]. In winter and autumn, for the medium range lead time (Week2), the forecast models perform better for T2m_max than T2m_min. Over all lead times, regions and

seasons, ECMWF performs better than UKMO on the detection of hot days except for Tw in the GU region in summer [Fig.6 (ii)]. Using MERRA reanalysis as reference, we also found that ECMWF has higher skills than UKMO, except over the CO region during winter (see [Fig.S6] in supplement material) and the Brier score is in the same range of values as the Brier score

computed using ERA5 for T2m and Tw. The results show a good performance of the models in predicting hot days for medium and long range forecasts.

### 3.3.2 Spatial variability of heat wave characteristics

After assessing the skill of the models in predicting hot days, we study the spatial variability of heat waves in the forecast models versus the reanalyses. Heat waves are defined as a period of at least three consecutive hots days. The frequency and characteristics of heat waves were calculated using T2m and Tw. The mean duration (resp. mean intensity) of heat waves was calculated for each grid point as the sum of heat wave duration (resp. intensity) divided by the number of years affected by a heat wave during the period from 2001 to 2020. A first assessment of the representation of heat waves in the forecast models is made by calculating the bias between the models and the reanalyses for heat wave frequency, duration and intensity. The spatial distribution of the heat wave frequency bias between the forecast models (ECMWF, UKMO) and ERA5 reanalysis shows similar evolution for T2m_min and T2m_max [Fig.7]. The models overestimate the frequency of heat waves in spring and summer over the Sahel for ECMWF, and from the Sahel to the Guinean region for UKMO. This overestimation in heat wave frequency is well marked in UKMO, indicating the inaccurate representation of the daily variability of T2m in the model. The Sahel and Guinean regions exhibit a strong convective activity during spring and summer which is very complex to take into account in the models. The representation of convective processes in the two models is assessed in the discussion at section 4. Some discrepancies are observed in the spatial evolution of heat waves frequency when using Tw over the Guinea coast [Fig.8]; we noticed an underestimation in heat waves frequency over the Sahel and Guinea region in autumn. In a second step, we assess the evolution of the number of heat wave days in the models versus ERA5 reanalysis. We observed a north-south gradient well established over West Africa and an overestimation of the number of heat wave days over the Guinea region for both T2m_min and T2m_max [Fig.9]. The north-south gradient is well marked in UKMO. This north-south gradient observed in UKMO with T2m is also found in Tw and tends to strengthen from spring to autumn [Fig.10]. The main differences between T2m and Tw on the representaion of heat wave days are found with ECMWF. In a third step, the spatial evolution of heat waves intensity is evaluated. ECMWF generally tends to underestimate the intensity of heat waves over the Sahel region, while UKMO overestimates the intensity of events from the Sahel to the Guinean coast for T2m_min (see [Fig.S7] in supplement). This overestimation of heat wave intensity in UKMO with T2m_min is considerably reduced when using T2m_max. The evolution of heat wave intensity with Tw is similar to that observed with T2m_min, except in summer for the ECMWF (see [Fig.S8] in supplement). We found very similar patterns when calculating the bias of intensity using MERRA reanalysis as reference for T2m_min, T2m_max and Tw (not shown). The bias of heat wave duration using MERRA as reference shows some differences with the results obtained with ERA5 reanalysis for T2m_min and T2m_max. ECMWF shows mostly an underestimation of heat wave days from the Sahel to the Guinean region in spring and summer for T2m_min and T2m_max (see [Fig.S9] in supplemental material). In summer, UKMO shows a large negative bias over Senegal, Guinea, Mali and Cameroon which tends to extend over the east of Sahel in autumn for T2m_min (see [Fig.S9 (i)] in supplement). We do not find significant differences when using MERRA as reference for the evaluation of heat waves duration with Tw (not shown).

## 3.4 Predictability of the heat waves in extended forecasts

In the previous section, we analyzed the spatial variability of heat wave characteristics in the forecast models versus the reanalyses. We are now assessing the skills of the models to detect heat waves in the three regions. This is done by calculating the following probabilistic scores described in Section 2: the hit-rate, GSS and FAR (see Section 2). The scores are calculated for T2m_min, T2m_max and Tw at daily and weekly time scales using the optimized forecasts (see Section 2) and ERA5 reanalysis chosen as reference. The evaluation of the forecast models at daily and weekly time scales provides useful insights for policy makers and climate services. Daily information is relevant for the accurate prediction of heat waves and early warning alert systems. The calculated metrics are then compared to a baseline climatology defined as the probability of having a heat wave in the ERA5 reanalysis over the period 2001-2020. The results presented below are obtained using a 20% threshold value to optimize the ensemble forecast system (see Section 2.4.4) [Fig.11]. The hit-rate and GSS values of the optimized forecasts using 40% and 60% percentile threshold values are lower than those obtained with the 20% threshold. The hit-rate values show a weak spatial variability in the 3 regions, which indicates that the skills of the models are not too sensitive to the geographical and climatic characteristics of the three regions. We also noticed a gradual loss of predictability in the models from winter to autumn, with high hit-rate values in winter. The forecast models show skills above the reference both for medium- and long-range forecasts (Week2, Week5) but the hit-rate values are below 0.5 indicating misses in the forecasts. The hit-rate values are slightly better for medium-range forecasts. ECMWF presents higher skills than UKMO for medium-range forecasts in winter. UKMO, for instance, is better for long-range forecasts, mainly for T2m_min [Fig.11 (i)(a-d)]. We also noticed that the predictability of heat waves is slightly better for heat waves associated with T2m_min than for T2m_max. We can infer from this result that wet heat waves are more predictable than dry heat waves. The models show higher skills in the AT and CO regions in winter and spring for T2m_min values for medium-range forecasts. The analysis of the inter-day variability of T2m is assessed by calculating the standard deviation (std) for each region using ERA5 reanalysis. We found small std values in the AT and GU regions, indicating low variability of daily T2m in these regions [Table5]. Conversely, high variability of daily T2m is observed in the CO region. The low inter-day variability of T2m in the AT region indicates a more stable signal which will lead to favorable conditions for heat wave detection in the models based on a statistical perspective. Statistically, this will contribute to the occurrence of heat waves, as the probability of having consecutive days above the threshold is higher in a stable signal than in one with high daily variability. The skills of the models previously highlighted in the AT region could be partly explained by this low variability of daily T2m (min, max) in the region. The hit-rate values obtained with Tw are very close to those associated with T2m in the different regions and seasons; but we also noticed with Tw, that UKMO is more skillful than ECMWF for medium- and long-range forecasts (see [Fig.S10(a-d)] in supplement material).

The second metric calculated for the evaluation of the models is the GSS. The GSS follows the same evolution as the hit-rate with much lower values between 0 and 0.2 both for T2m_min, T2m_max and Tw ([Fig.11 (i)(i-l)], [Fig.11 (ii)(i-l)] and [Fig.S10(i-l)] respectively). This suggests that the skills of the models to predict heat waves are considerably reduced when random hits (chance hits) are removed from the total hits. We found that the GSS values are low overall lead time and season; the highest values are observed in winter. In winter, the atmospheric circulation in West Africa regions is mainly governed

by the harmattan flow which results in low convective activity in the regions and therefore an improvement of the predictive skills of the models compared to summer. The GSS values are greater than 0, indicating that the forecast models perform better than a random forecast mainly for medium-range forecasts (Week2) [Fig.11(i-l)]. Similar results of the GSS are found with Tw (see [Fig.S10(i-l)] in supplement). An important parameter of a forecast system is its reliability in predicting events. This property is assessed using the False Alarm ratio: Do the events predicted by the models always occur in the reanalysis? This is done by the calculation of the FAR between the forecast models and ERA5 reanalysis for T2m_min [Fig.11 (i)(e-h)], Tw [Fig.S10(e-h)] and T2m_max [Fig.11 (ii)(e-h)]. The FAR values are too high on average, about 0.7 and 0.85 for the medium and long range forecasts, respectively for all the three indicators. This suggests that the models tend to overestimate the number of heat days with respect to ERA5. The FAR increases with the lead time; ECMWF makes fewer false alarms than UKMO for medium-range forecasts. The FAR is considerably reduced when we increase the threshold values used to optimize the ensemble forecasting systems (see [Fig.S11] in supplement for T2m_min). This is highlighted more clearly in ECMWF than UKMO over all seasons and regions. This result is consistent with the fact that the prediction of an event will be more robust as a large number of model members have predicted the event. However, we found that this is not the case for the hit-rate and GSS for which high values are obtained with lowest threshold (20%) (see [Fig.S12 and Fig.S13] in supplemental material). Thus, according to the context of the study, a compromise must be found in order to obtain a good balance between hit-rate and false alarms. The skill of the models in detecting heat waves is also assessed on a weekly time scale. As expected, we found an overall increase in the values of hit-rate and GSS for T2m_min, T2m_max and Tw (see [Fig.S14] for T2m_min in supplement). The forecast models show skills at weekly time scale compared to the baseline climatology. To be consistent with the previous analyses, the predictability of heat wave intensity in the models is also assessed for medium- and long-range forecasts (Week2 and Week5) using ERA5 reanalysis. We found a large spatial variability in heat wave intensity across regions. The strongest heat waves are found in the CO region (as shown in Ngoungue Langue et al. (2023)) and low intensity heat waves in the GU region except for Tw in summer for all the three indicators. Although the models demonstrate consistency in forecasting the spatial distribution of heatwave intensity, they fail in accurately predicting the specific intensity values. The models underestimate the intensity of heat waves (see [Fig.12 (i)] for T2m_min, the results are very similar for T2m_max [Fig.12 (ii)] and Tw (not shown)). The forecast of heat wave intensity in the models remains a difficult task, even though the models show skills in heat wave detection.

## 4 Discussion

The behavior of the forecast models varies across different climatic regions. The question that arises is : Why do these differences exist?

In order to tackle this question, we analyze some key factors in the models such as the physical parameterizations, the data assimilation technique and the spatial resolution of the atmospheric component. First of all, the physical parameterizations used to simulate atmospheric processes such as convection, turbulence, interactions surface-ocean, surface-radiation and cloud microphysics are different for the two models. For example, for the representation of the convective activity, ECMWF is

using the Tiedtke scheme (Tiedtke, 1989) and UKMO, the Met Office convective scheme (Hagelin et al., 2017). The Tiedtke convection scheme is one of the first mass-flow convection schemes, which aims to parameterise the effects of deep convection in numerical weather models. It simulates the vertical transport of heat, moisture and momentum associated with convective updrafts and downdrafts. The system takes into account various factors, including atmospheric instability, moisture content and boundary layer conditions to estimate convective processes. UKMO also uses a mass flux convection scheme, but different from the Tiedtke scheme, which takes into account atmospheric instability and moisture content to determine convective activity. The difference between the two convective schemes could lead to a wrong representation of convective activity in the region, and thus limit the predictive skills of the models mostly for wet heat waves. The models used the same data assimilation methods (4D-Var) for control analyses but the data and initial conditions are completely different. ECMWF assimilates a wide range of global and regional observational data, including satellite, radar and ground-based measurements. The UKMO focuses on observation data relevant to the United Kingdom and surrounding regions. These differences may influence the representation of surface-ocean interactions in the models. The differences observed in the representation of T2m_min over the Atlantic ocean [Fig.3] can result from these types configurations. Another important factor is the spatial resolution of the atmospheric component of the two models : ECMWF has a higher spatial resolution than UKMO (0.16°x0.16° Vs 0.83°x0.56°), which means that it can capture local-scale variability or atmospheric processes and provide more accurate forecasts for specific regions. Even if we transform the native resolution of the two models into a regular 0.25°x0.25° grid, some local-scale patterns will be found in the new grid. This may explain why the ECMWF performs better than the UKMO in some regions, for example in the case of extreme temperatures over AT and GU regions [Fig.6]. A more detailed analysis of the influence of each factor on the results is beyond the scope of this paper.

Another point to highlight in this work is the use of reanalyses to evaluate heatwave forecasting models. We are aware that reanalysis data have a high resolution compared to observations from local stations. As a result, they are unable to represent the urban heat island effect which exacerbates heat stress during heat waves. The resolution of the reanalyses makes it impossible to detect the highest temperatures at specific locations. Heat waves in the Sahel region occur mainly in spring due to the high temperatures in the region at that time (Barbier et al., 2018; Guigma et al., 2020). In this study, the region of interest was extended to the Guinean region in which heat waves are mainly driven by humidity. Heat wave detection was then carried out using the $90^{th}$ percentile as a threshold over the January to December season. The $90^{th}$ percentile appears to be a sufficient threshold for monitoring heat waves affecting human health. Nevertheless, it is useful to calculate the intensity of events in order to determine a classification according to their severity (intensity), from "harmless" to "extremely dangerous", for example. This is what the STEWARD project is doing by developing a database on heat waves and their potential impact on human health. This study showed that the forecast models were actually able to predict heatwaves occurrence up to two weeks in advance in the different regions. On the other hand, we found that the models overestimated the frequency and duration of events, whatever the lead time. Consequently, it will be necessary to find a good balance between hits and false alarms in order to develop a robust early warning system to prevent populations from heatwaves.

We noticed that in some regions, the CRPS score is slightly better during week 5 than in week 2; which is rather surprising. This behavior is more apparent with UKMO; it is indeed the case in the Guinea region in January and February (see [Fig.S15] in supplement material). One first hypothesis on this behavior in the UKMO model could be the strong seasonality in the region. Following this hypothesis, we investigate the seasonal evolution of the spread of temperatures in the Guinea region over the lead times. This investigation is very interesting because the variability of the spread can be linked to the skills of a forecasting model. We noticed higher spread values in Winter compared to the rest of the seasons (see [Fig.S16] in supplement material). However, when we looked at week 2 and week 5, we did not find a specific evolution of the spread which can explain the behavior observed in UKMO. The second hypothesis is the presence of a bias in UKMO which decreases over the lead times. Therefore, we analyzed the evolution of the bias of temperatures over lead times. We found an intense cold bias of UKMO with respect to ERA5 over the Guinea region during winter. This intense cold bias could lead to high CRPS values in the region during winter (see [Fig.S17] in supplement material). The evolution of the bias during week 2 and week 5 shows similar patterns in January and February. This hypothesis on the bias evolution is not supported by these findings. The investigation in more detail of the origins of this behavior in the UKMO model is very complex, and outside the scope of the present study.

## 5 Conclusions

This study is a first assessment of the predictability of heat waves in West African cities on a subseasonal time scale. Two models that are part of the S2S prediction project, namely: ECMWF and UKMO, were evaluated using two state-of-the-art reanalysis data : ERA5 and MERRA over the period 2001-2020. To carry out this study, we first analyzed the representation of T2m and Tw in the forecast models with respect to the reanalyses. We found that the models are cooler than ERA5 in the Sahel region, and hotter than MERRA in the Sahel and Guinea region. These uncertainties between the 2 reanalyses have also been highlighted in Ngoungue Langue et al. (2023). Secondly, we investigate the representation of hot days and heat wave characteristics in the models with respect to the reanalyses. The Brier score is very good for T2m and Tw, indicating that the models are able to detect single extreme events up to 5 weeks lead time. Over all lead times, seasons and regions, ECMWF performs better than UKMO in detecting single extreme events when compared to both ERA5 and MERRA. But when it comes to heat waves, which are rarer and extremely difficult to predict compared to a single extreme event because the persistence factor comes into play, the skill of the models decreases. Furthermore, we also know that the forecasting models underperform in tropical regions due to a poor representation of convective processes in their physical parameterisation. Consequently, these scores, which are low but greater than the climatology, are significant for assessing the skill of the models in predicting heat waves in tropical regions which remains a complex task. The hit-rate values are slightly better for medium-range forecasts (up to two weeks lead time). We found that wet heat waves are more predictable than dry heat waves. This result is consistent with Guigma et al. (2021). On average, the False Alarm Ratio (FAR) is excessively high and tends to increase with the lead time. Only approximately 15% to 30% of the predicted heat wave days are actually observed for Week 5 and Week 2, respectively. This suggests that the models overestimate the duration of the heat waves with respect to ERA5. ECMWF issues fewer false alarms than UKMO for medium-range forecasts. The detection of dry heat waves is slightly better with ECMWF for medium-

range forecasts, while it is better with UKMO for long-range forecasts. For wet heat waves, UKMO outperforms ECMWF for both medium- and long-range forecasts. Although the models demonstrate skills on heat wave detection compared to a baseline climatology, they underestimate the intensity of heat waves with respect to ERA5 at short, medium and long range forecasts. Regarding these results, we can recommend the use of subseasonal forecasts to predict the occurrence of heat waves up to two weeks in advance, but as far as their intensity is concerned, it is still challenging.

In future work, we will investigate in more detail the origins of the differences observed in the two forecast models over the different regions. It has been shown recently in some studies that machine learning techniques can be useful to extend the predictability range of weather forecasts [e.g. Salcedo-Sanz et al. (2016); Anjali et al. (2019); Azari et al. (2022); van Straaten et al. (2023)]. Based on these findings, it would be interesting to investigate the potential of machine learning algorithms on heat waves forecasting.

*Competing interests.* The contact author has declared that none of the authors has any competing interests

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

**Tables**

Table 1. Differences between the two forecasts models

| Models | Hindcasts | | | | Real time forecasts | | | |
|---|---|---|---|---|---|---|---|---|
| | *dates* | *size* | *range* | *period* | *dates* | *size* | *range* | *Model version* |
| **ECMWF** | 2/week, on Monday and Thursday | 11 | 0-46 days | past 20 years | 2/week, on Monday and Thursday | 51 | 0-46 days | CY48R1 |
| **UKMO** | 4/month on the 1st, 9th, 17th, 25th | 3 prior 2016 7 from 25/03/2017 | 0-60 days | 1993-2016 | 4/month 1st, 9th, 17th, 25th | 4 | 0-60 days | GloSea5-GC2-LI |

Table 2. Land sea mask (lsm) of west African towns used in this study

| Towns | latitude | longitude | lsm |
|---|---|---|---|
| DAKAR | 14.75 | -17.25 | 0.6 |
| ABIDJAN | 5.25 | -3.75 | 0.5 |
| NOUAKCHOTT | 18 | -16 | continent |
| CONAKRY | 9.5 | -13.5 | 0.5 |
| MONROVIA | 6.25 | -10.75 | 0.6 |
| BAMAKO | 12.5 | -8 | continent |
| YAMOUSSOUKRO | 6.75 | -5.25 | continent |
| OUAGADOUGOU | 12.25 | -1.5 | continent |
| ACCRA | 5.5 | -0.5 | 0.8 |
| LOMÉ | 6 | 1 | 0.5 |
| NIAMEY | 13.5 | 2 | continent |
| COTONOU | 6.5 | 2.5 | 0.7 |
| LAGOS | 6.5 | 3.5 | 0.5 |
| ABUJA | 9 | 7.5 | continent |
| DOUALA | 4 | 9.75 | 0.9 |

Table 3. Description of the threshold values.

| Threshold values | Description |
|---|---|
| 20% | 20% of the ensemble members are associated to a hot day |
| 40% | 40% of the ensemble members are associated to a hot day |
| 60% | 60% of the ensemble members are associated to a hot day |

**Table 4.** Contingency table.

| 2X2 Contingency table | | Event Observed | |
|---|---|---|---|
| | | YES | N0 |
| **Event forecast** | YES | hits | false alarms |
| | NO | misses | correct rejections |

**Table 5.** Inter-daily variability of 2-meter temperature over the period 2001-2020 using ERA5 reanalysis during the seasons for T2m_min and T2m_max.

| | AT | | | | GU | | | | CO | | | |
|---|---|---|---|---|---|---|---|---|---|---|---|---|
| | Win | Spri | Sum | Aut | Win | Spri | Sum | Aut | Win | Spri | Sum | Aut |
| T2m_min | 0.5 | 0.5 | 0.43 | 0.77 | 0.57 | 0.36 | 0.44 | 0.44 | 1.64 | 1.74 | 1.08 | 1.69 |
| T2m_max | 0.38 | 0.35 | 0.43 | 0.58 | 0.39 | 0.69 | 0.76 | 1 | 1.71 | 0.81 | 2.04 | 1.66 |

**Figures**

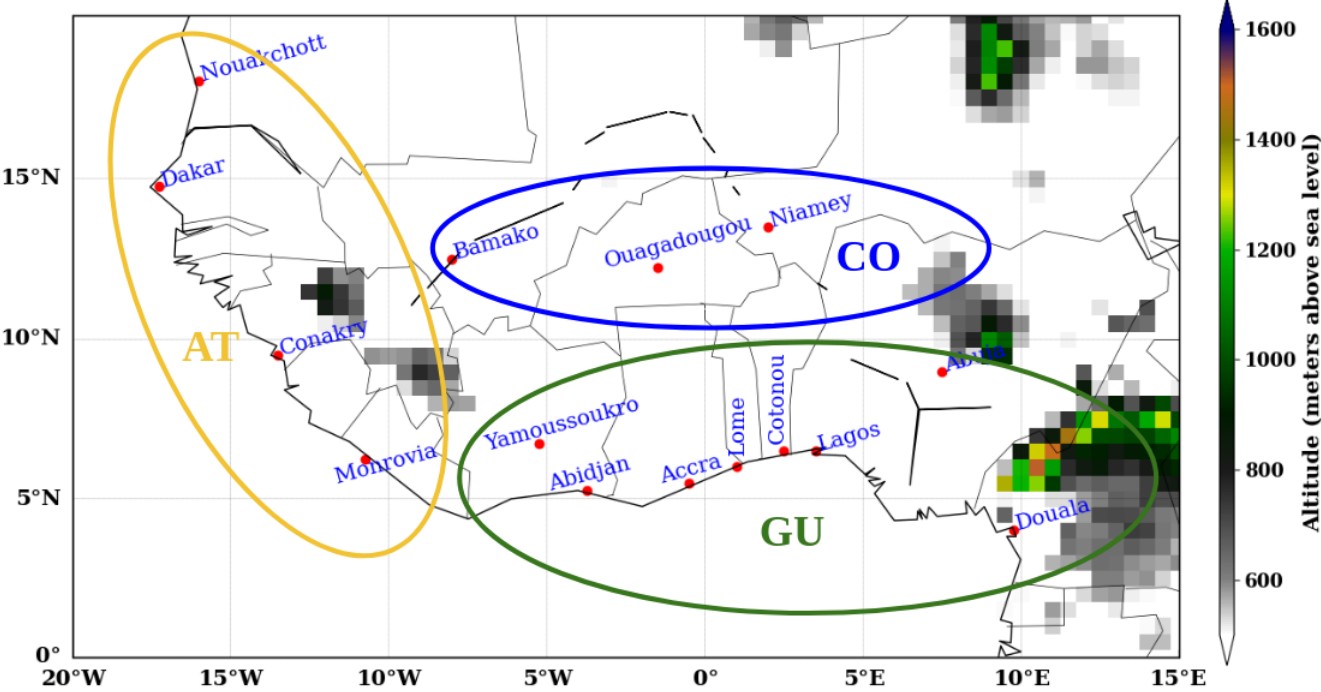

**Figure 1.** Topographic map of West Africa using ERA5 elevation data. The circles on the map represent the different climatic zones: AT (Coastal atlantic zone), CO (Continental zone) and GU (Coastal Guinean zone). The y and x axes represent the latitude and longitude respectively. The color bar shows the elevation in meters over the region.

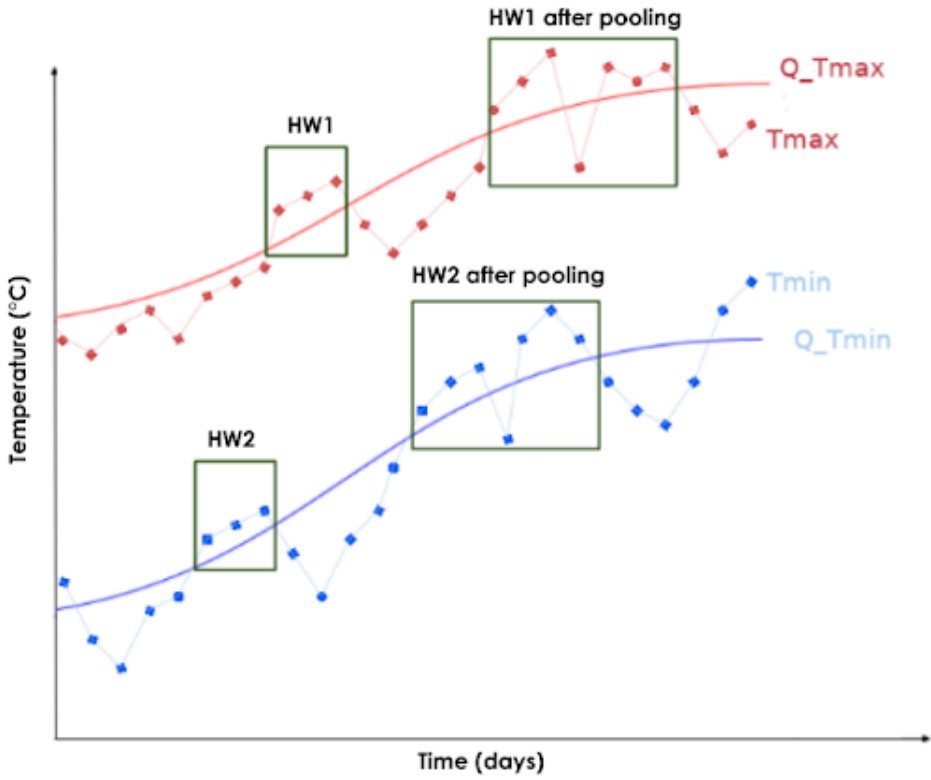

**Figure 2.** Detection process of heat wave: HW1/HW2 represent events associated respectively to maxima/minima temperature. The red/blue lines with circles are max/min daily temperatures. Red/blue solid lines are respectively max/min thresholds. X- and Y- axis represent the time in days and the temperature in degrees celsius. 'With pool' refers to the pooling of two (or more) events separated by a day characterized by the value of a given indicator below the daily $XX^{th}$ percentile. This figure is a 'theoretical/schematic' illustration of the different types of heat waves investigated in this work.

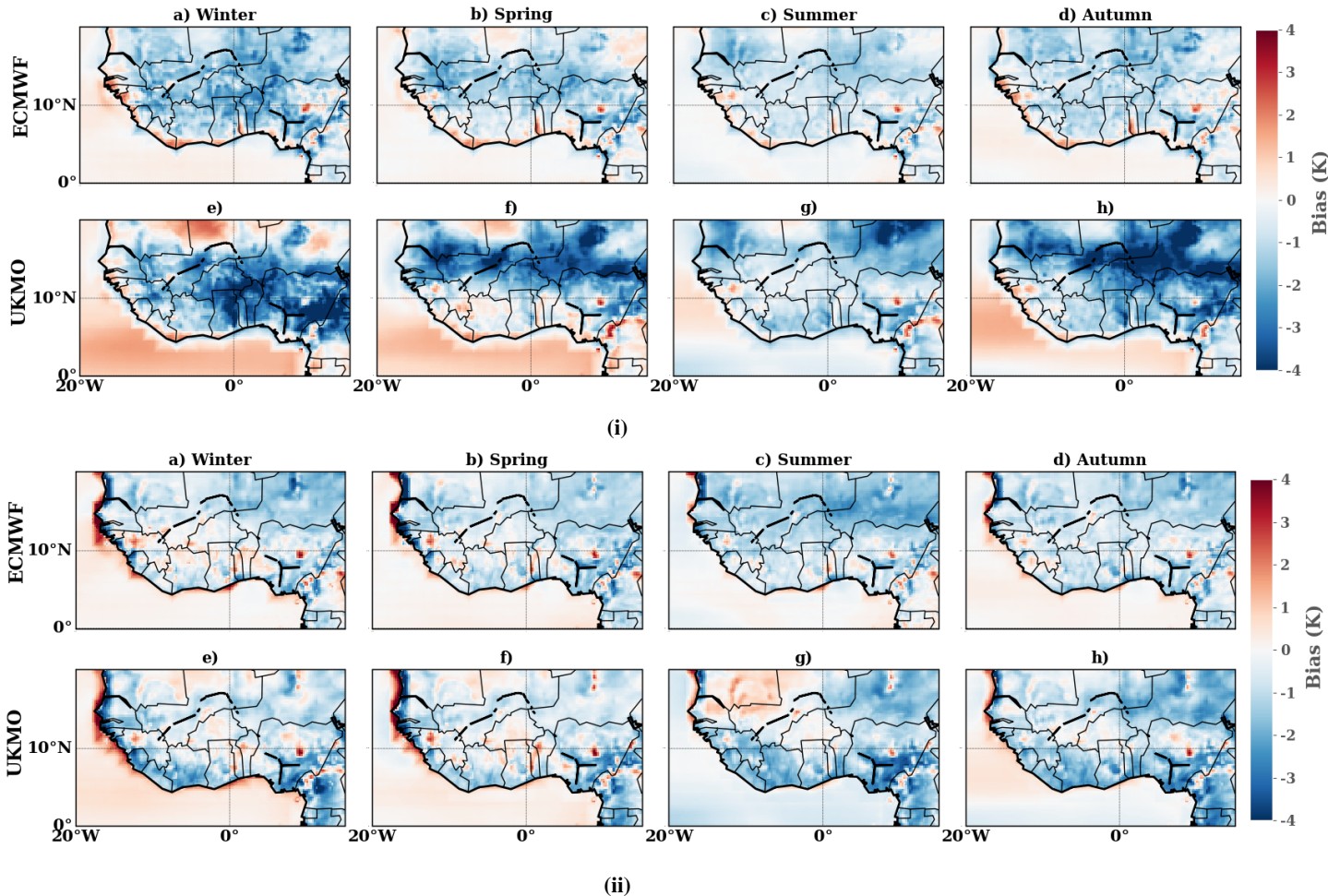

**Figure 3.** Spatial variability of the climatological bias between the forecast models ensemble mean and ERA5 reanalysis over the period 2001-2020 for : **(i)** T2m_min and **(ii)** T2m_max, during the seasons : (a,e) winter; (b,f) spring; (c,g) summer and (d,h) autumn. The bias is computed as the difference between the forecast models and ERA5 considering all the lead times. The color indicates the bias values in degrees Kelvin. The X and Y axes represent the longitude and latitude respectively.

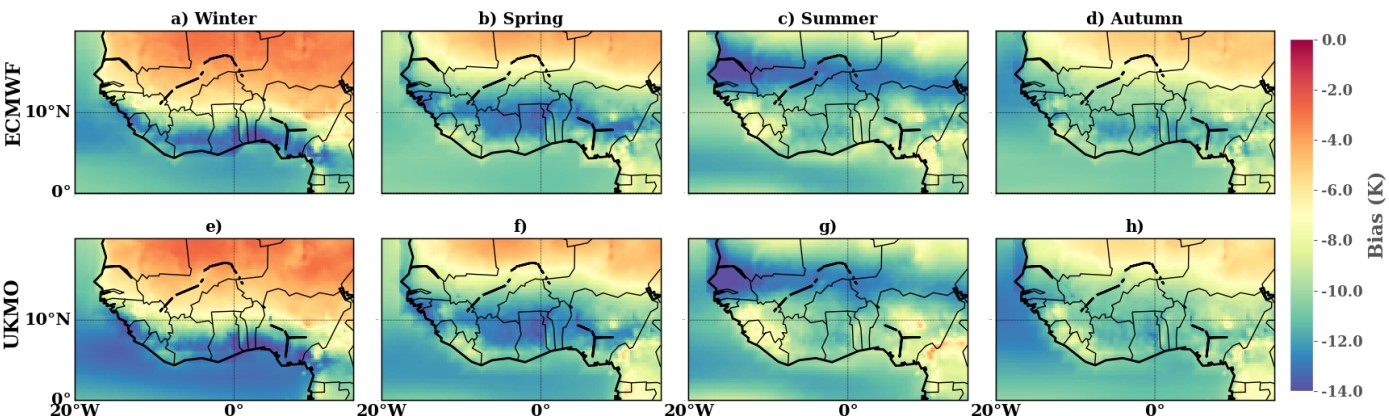

**Figure 4.** Spatial variability of the climatological bias between the forecast models ensemble mean and ERA5 reanalysis over the period 2001-2020 for Tw during the seasons : (a,e) winter; (b,f) spring; (c,g) summer and (d,h) autumn. The bias is computed as the difference between the forecast models and ERA5 considering all the lead times. The color indicates the bias values in degrees Kelvin. The X and Y axes represent the longitude and latitude respectively.

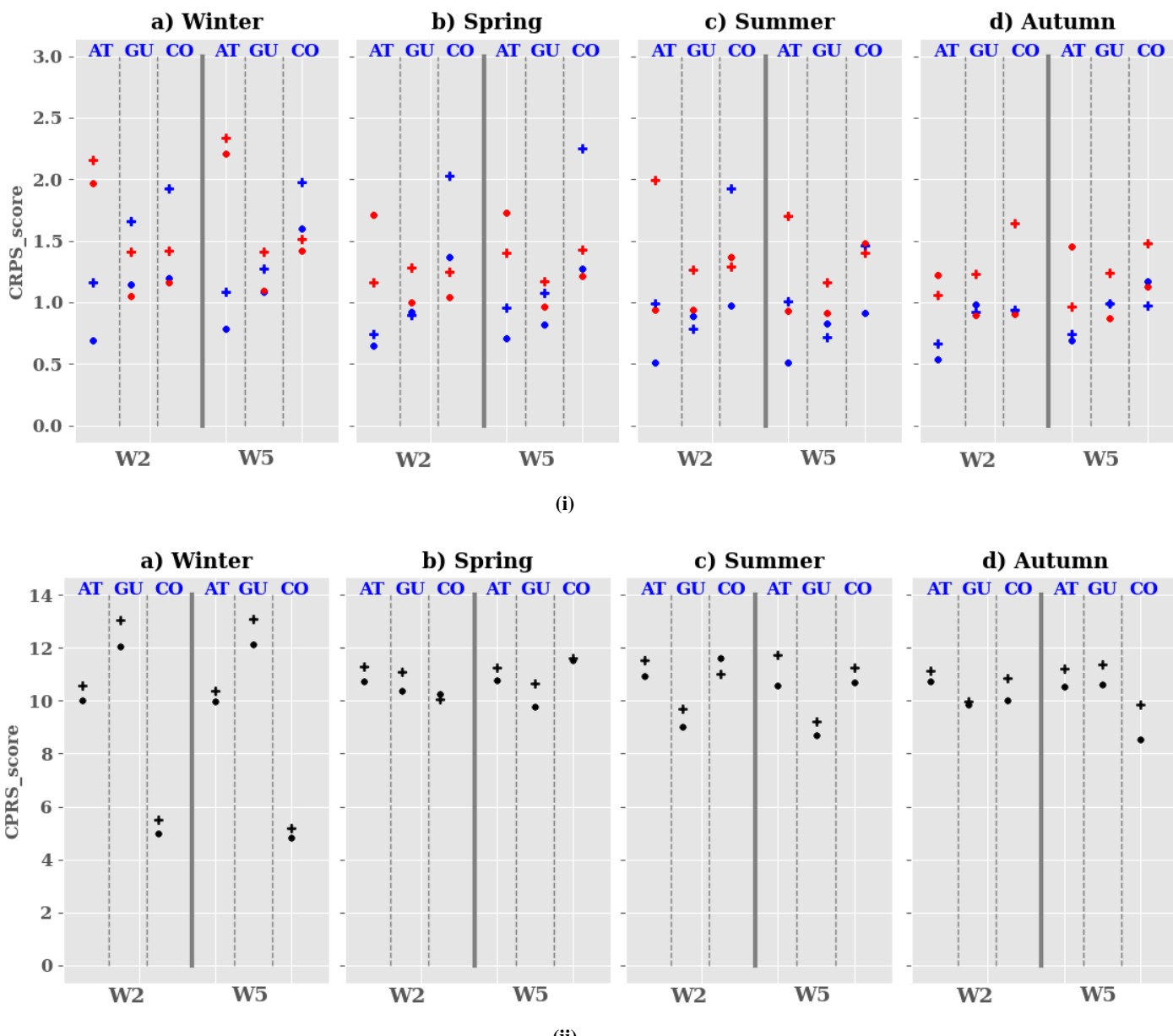

**Figure 5.** Evolution of the CRPS score between the forecast models and ERA5 reanalysis using T2m **(i)** and Tw **(ii)** over the period 2001-2020 during the seasons : (a) winter, (b) spring, (c) summer and (d) autumn. The blue, red and black colors represent the CRPS score calculated using T2m_min, T2m_max and Tw mean respectively. The dot and cross symbols indicate the CRPS score obtained with ECMWF and UKMO respectively. The Y and X axes show the CRPS values and the lead times ( W2: week2 and W5: week5) respectively.

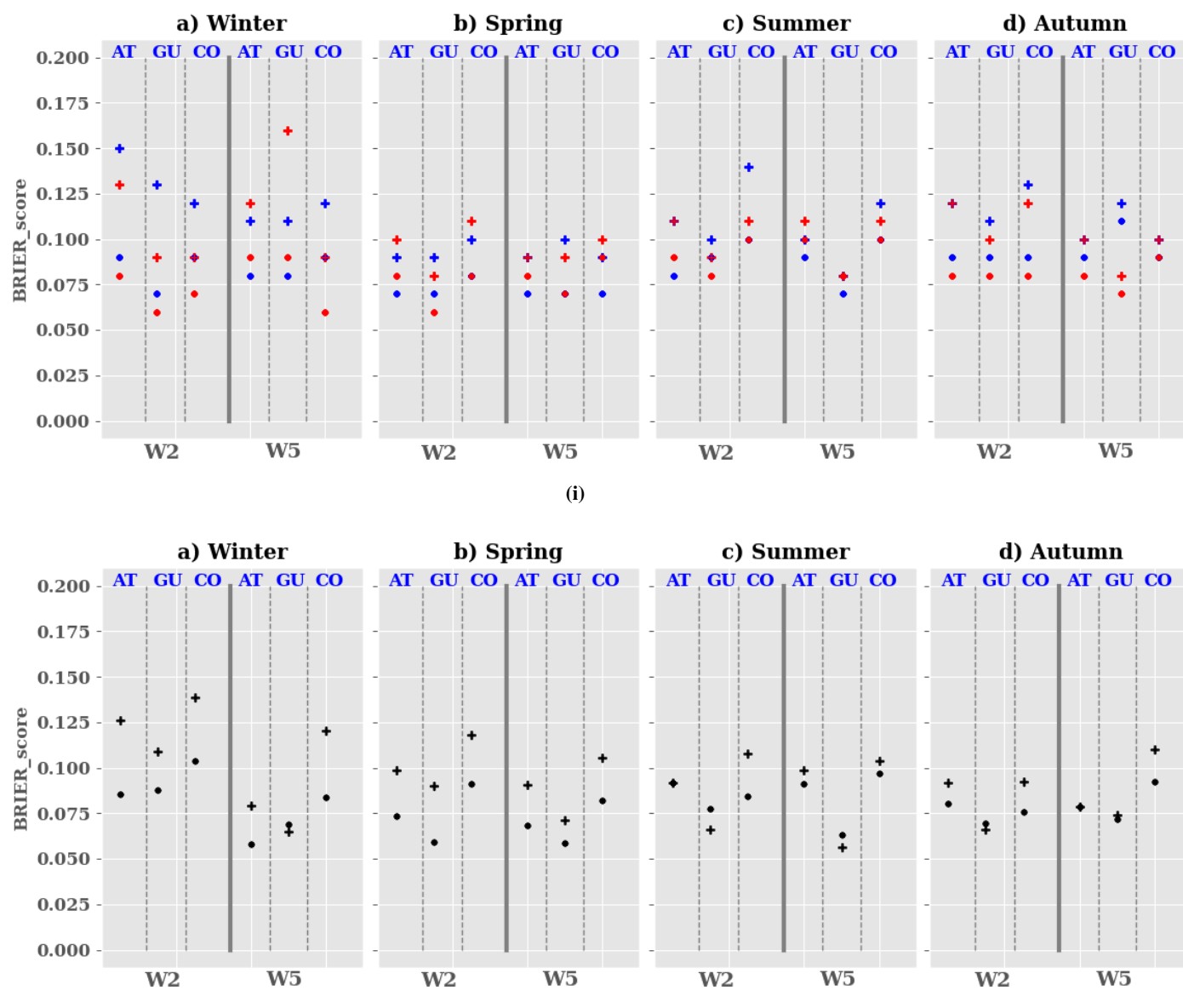

**Figure 6.** Evolution of the Brier score between the models and ERA5 reanalysis using T2m **(i)** and Tw **(ii)** over the period 2001-2020 during the seasons : (a) winter, (b) spring, (c) summer and (d) autumn. The blue, red and black colors represent the CRPS score calculated using T2m_min, T2m_max and Tw mean respectively. The dot and cross symbols indicate the Brier score obtained with ECMWF and UKMO respectively. The Y and X axes show the CRPS values and the lead times ( W2: week2 and W5: week5) respectively.

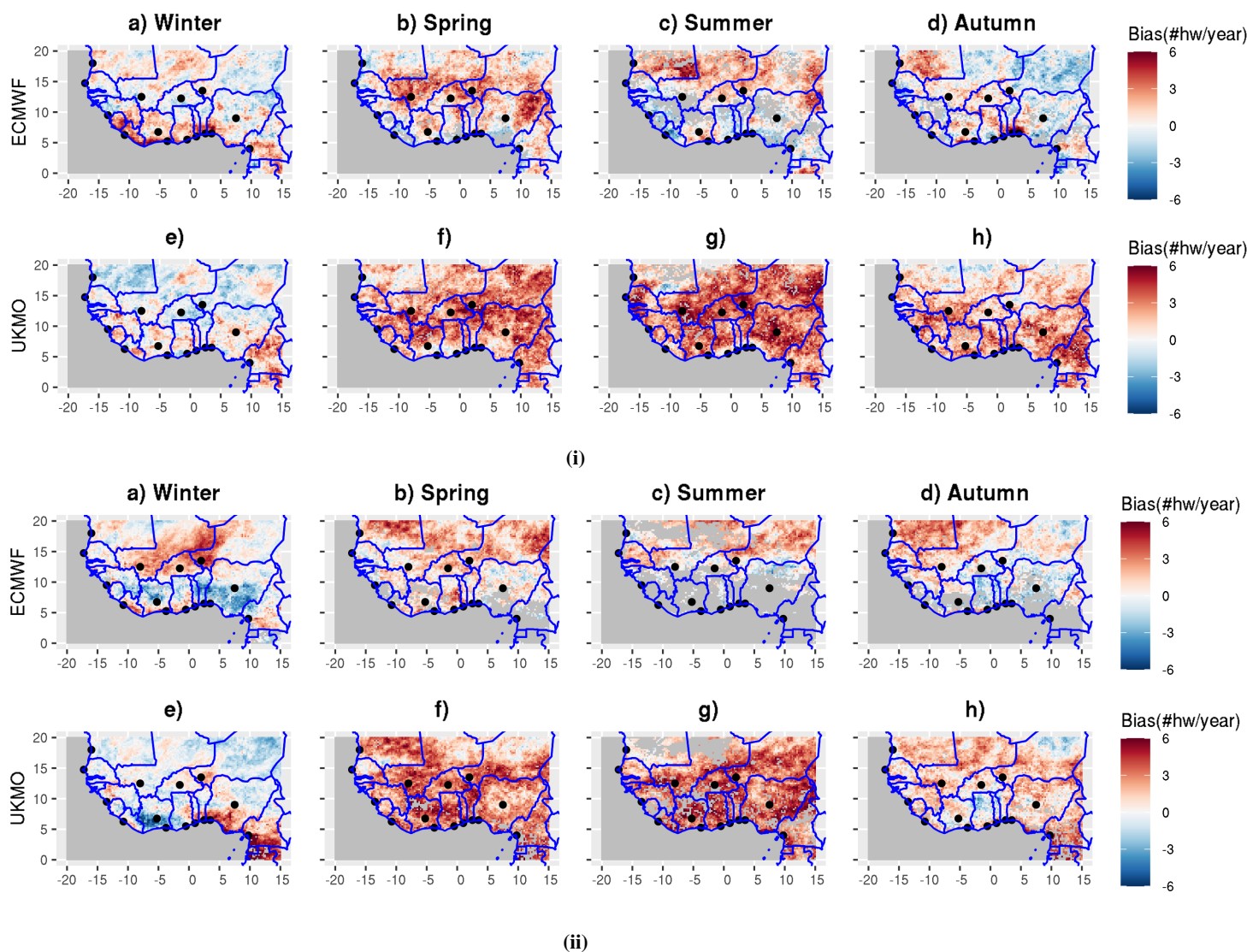

**Figure 7.** Spatial variability of heat wave frequency bias between forecast models and ERA5 over West Africa from 2001 to 2020 for: **(i)** T2m_min values and **(ii)** T2m_max values, during: (a,e) winter; (b,f) spring; (c,g) summer and (d,h) autumn. The bias is calculated as the difference in heat wave frequency between the forecast models and ERA5. This analysis is performed using the unperturbed member of the models. The color bar indicates the bias values without units. The grey color represents missing values. The X and Y axes represent longitude and latitude respectively. The solid blue lines indicate the borders between countries; the black dots represent the cities of interest for this study (this applies to the rest of the paper).

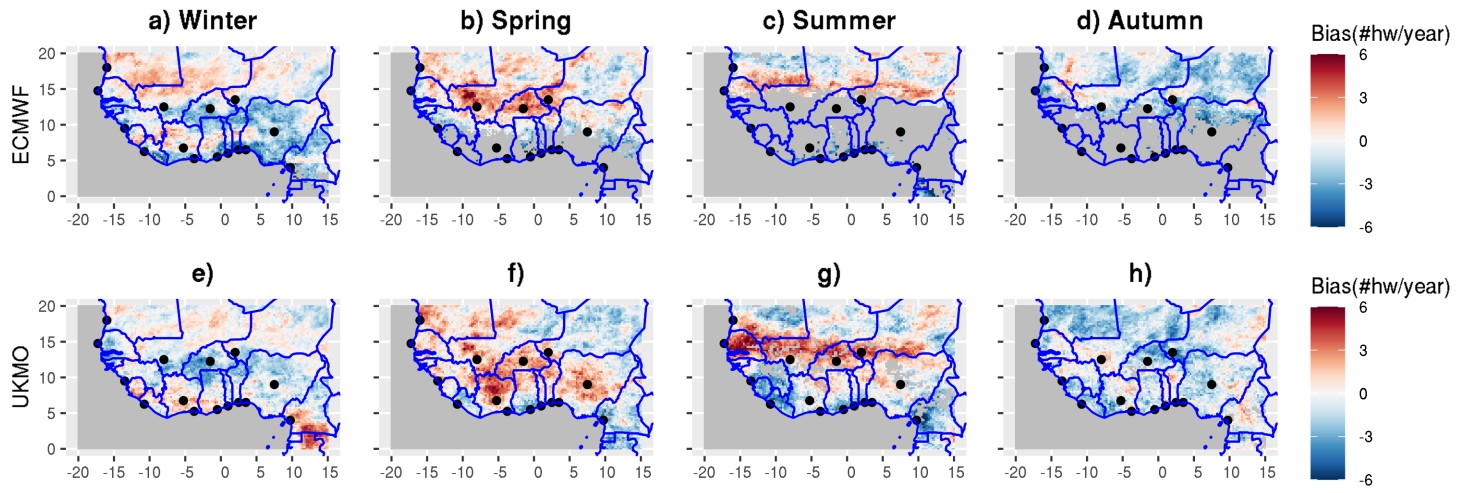

**Figure 8.** Spatial variability of heat wave frequency bias between forecast models and ERA5 over West Africa from 2001 to 2020 using Tw during: (a,e) winter; (b,f) spring; (c,g) summer and (d,h) autumn. The bias is calculated as the difference in heat wave frequency between the forecast models and ERA5. This analysis is performed using the unperturbed member of the models. The color bar indicates the bias values without units. The grey color represent missing values. The X and Y axes represent longitude and latitude respectively.

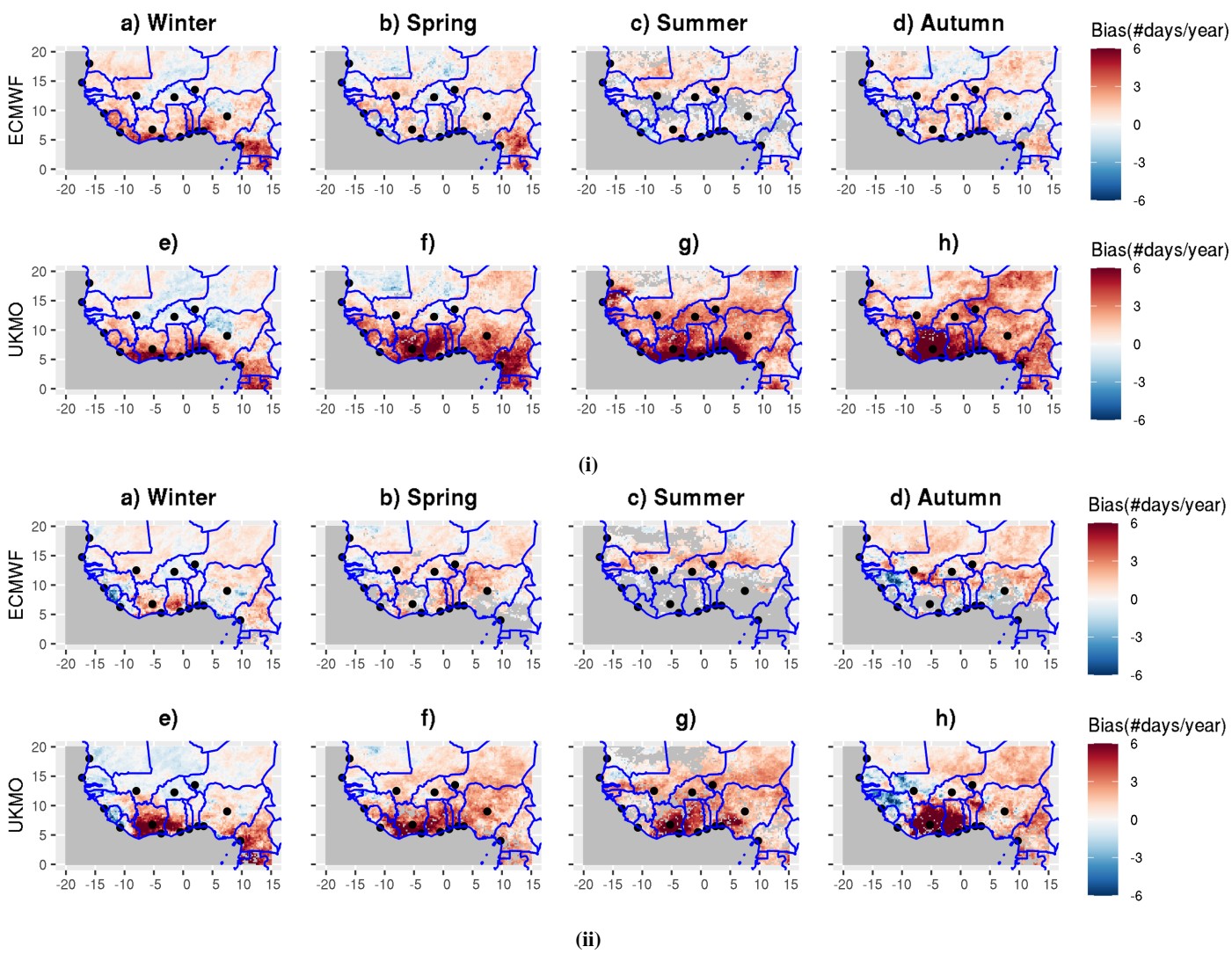

**Figure 9.** Spatial variability of heat wave duration bias between the forecast models and ERA5 over West Africa from 2001 to 2020 for: **(i)** T2m_min values and **(ii)** T2m_max, during: (a,e) winter; (b,f) spring; (c,g) summer and (d,h) autumn. The bias is calculated as the difference in heat wave duration between the forecast models and ERA5. This analysis is performed using the unperturbed member of the models over years where heat waves were detected. The color bar indicates the bias values without units. The X and Y axes represent longitude and latitude respectively.

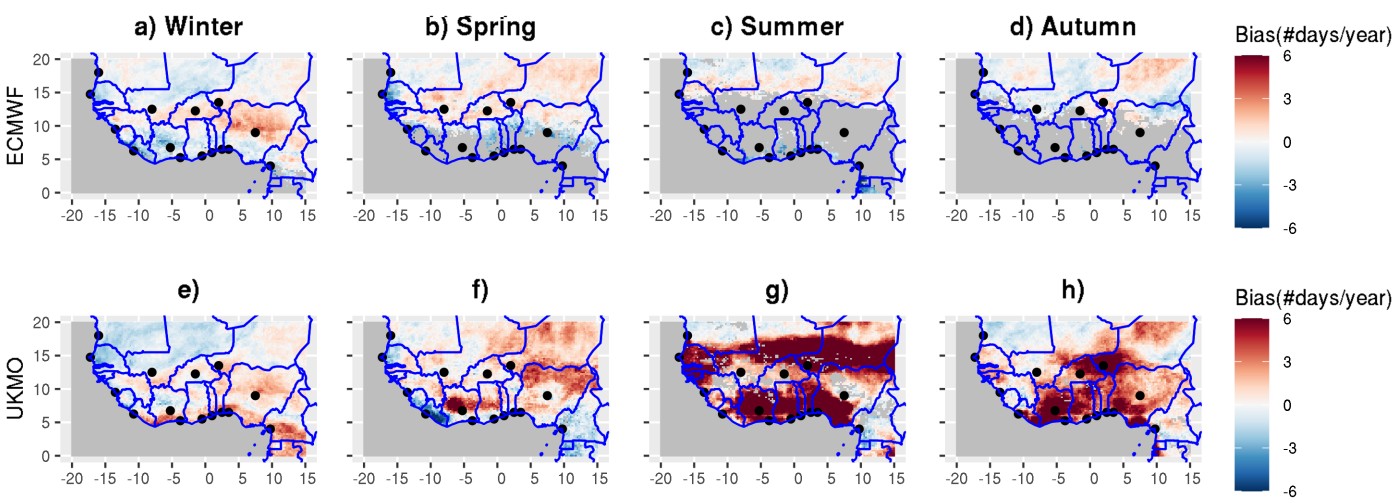

**Figure 10.** Spatial variability of heat wave duration bias between forecast models and ERA5 over West Africa from 2001 to 2020 using Tw during: (a,e) winter; (b,f) spring; (c,g) summer and (d,h) autumn. The bias is calculated as the difference in heat wave duration between the forecast models and ERA5. This analysis is performed using the unperturbed member of the models. The color bar indicates the bias values without units. The X and Y axes represent longitude and latitude respectively.

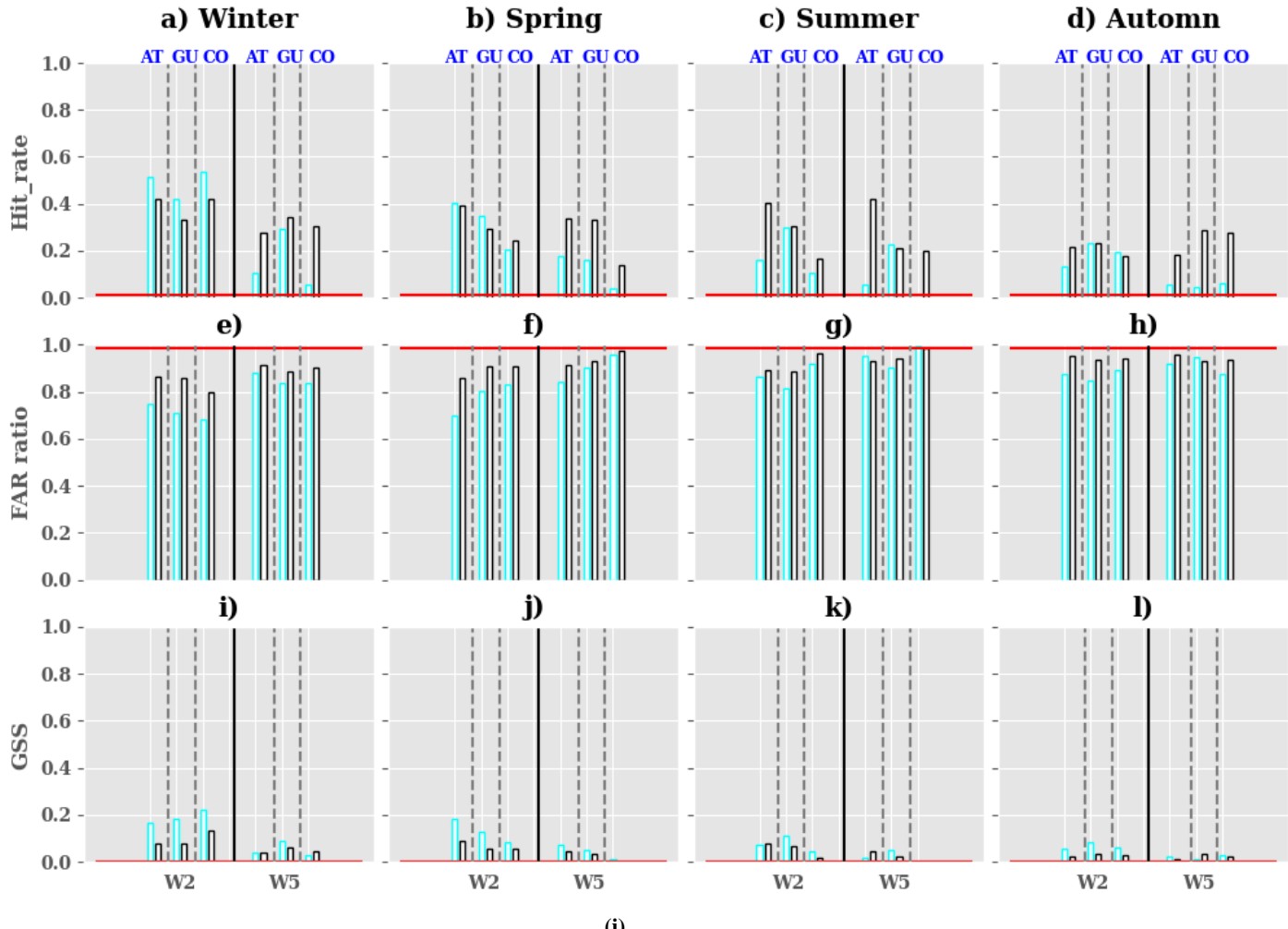

**(i)**

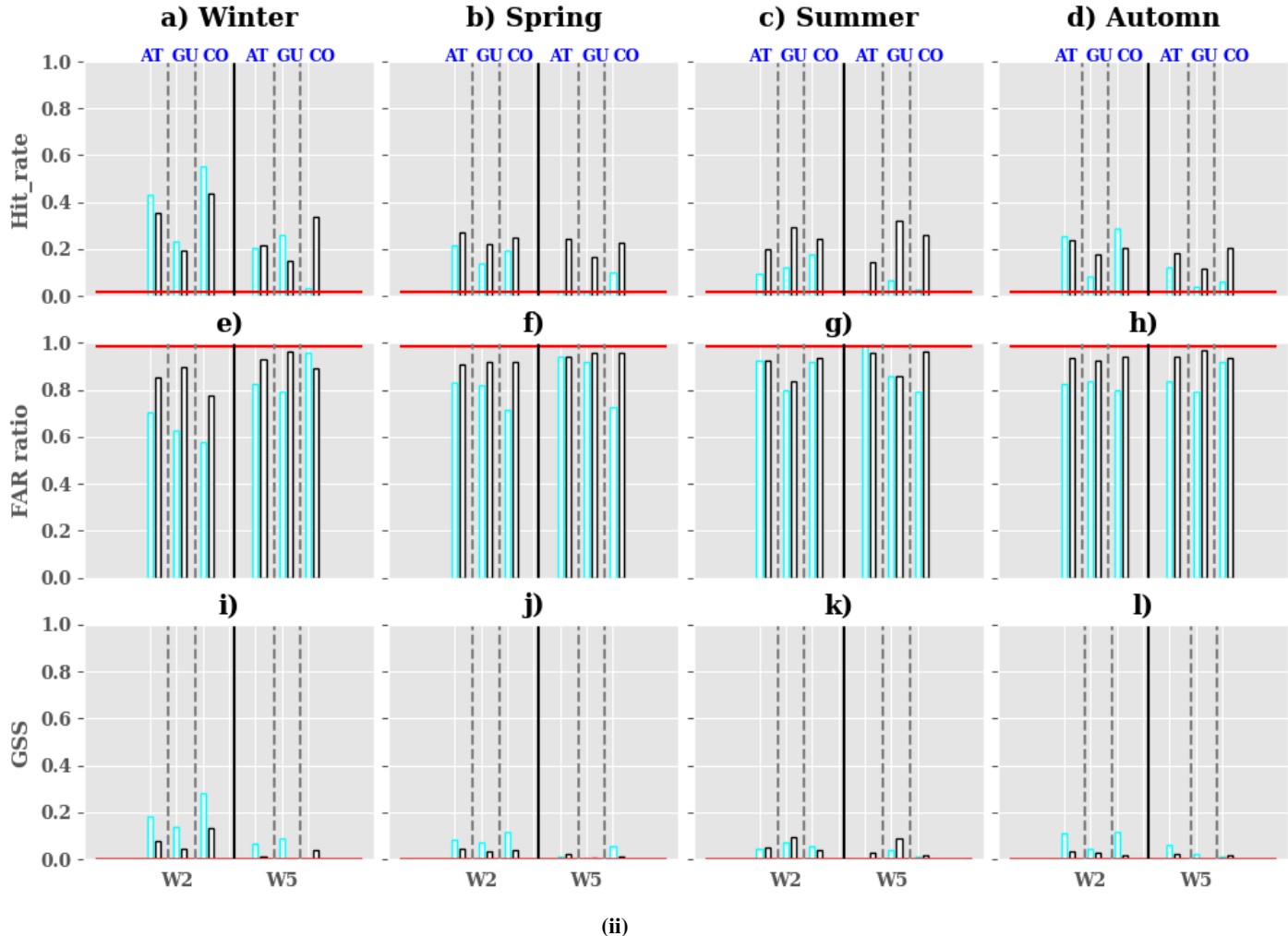

**Figure 11.** Evaluation of heat waves detection in the forecast models with respect to ERA5 at daily time scale over the period 2001-2020 using T2m_min **(i)** and T2m_max **(ii)** values for : (a-d) hit-rate, (e-h) FAR and (i-l) GSS. The metrics were computed using the optimized forecasts with the 20% threshold (see section Methods for the optimisation of the ensemble forecasts). The metrics were calculated during the seasons : (a,e,i) winter; (b,f,j) spring; (c,g,k) summer and (d,h,l) autumn. The cyan and black borders of bar plots indicate the metrics obtained when using ECMWF and UKMO respectively. The Y and X axes show the metrics values and the lead times (W2: week2 and W5: week5) respectively. The horizontal red line represents the baseline climatology.

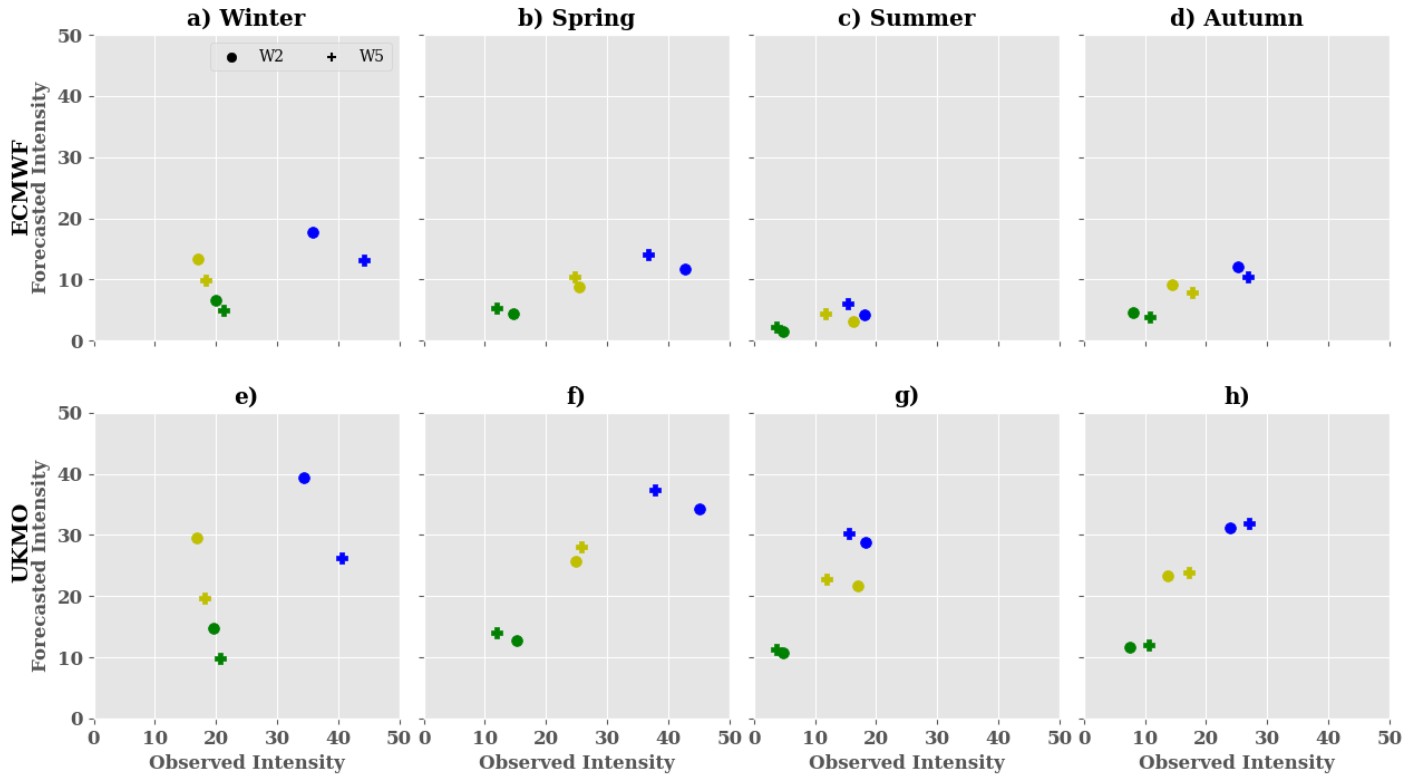

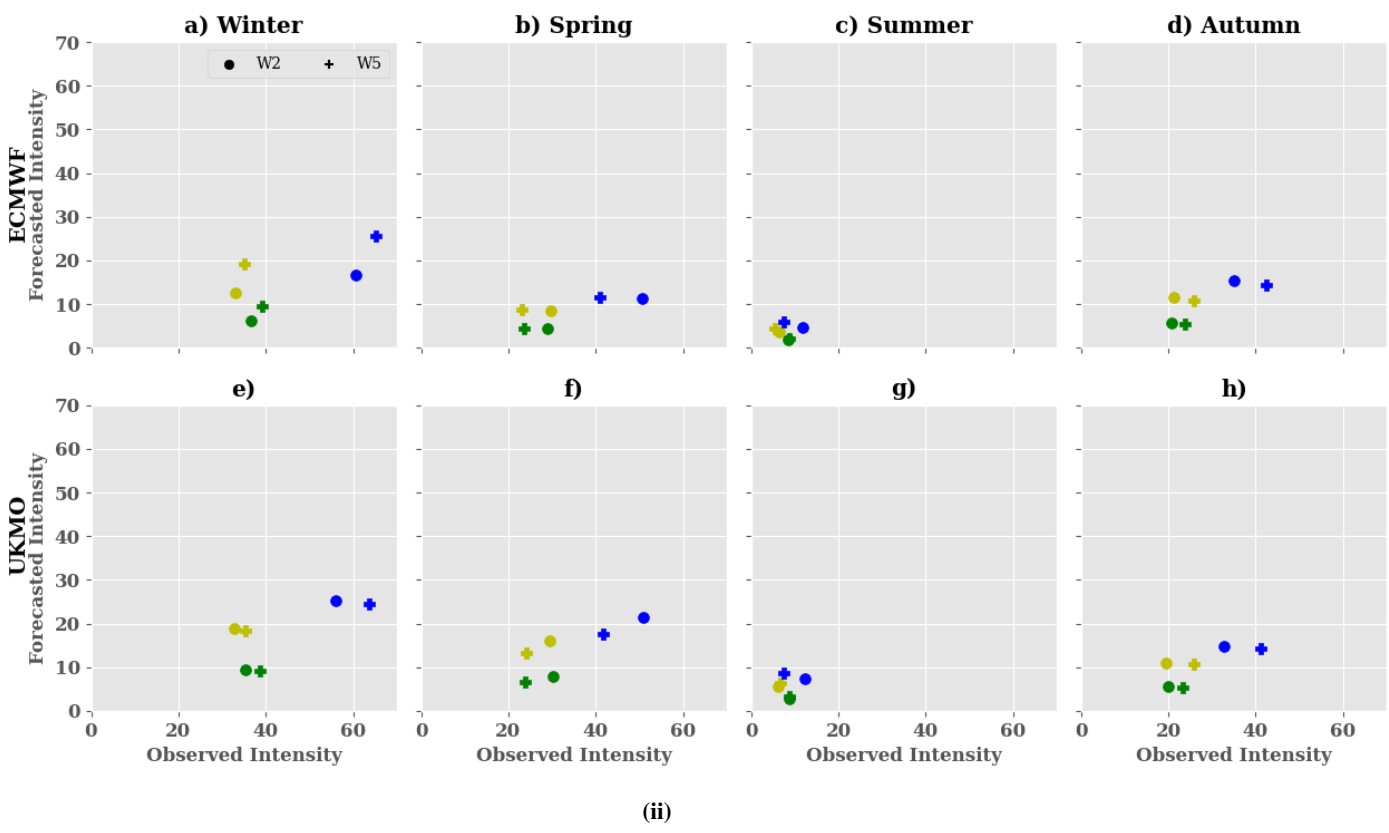

**Figure 12.** Evaluation of the intensity of heat waves in the forecast models and ERA5 over the period 2001-2020 during the seasons : (a,e) winter; (b,f) spring; (c,g) summer; (d,h) autumn using T2m_min **(i)** and T2m_max **(ii)**. Yellow, green, blue colors represent the values of intensity in the AT, GU, CO regions respectively. The dot and cross symbols represent the intensity of heat waves during week2 (W2) and week5 (W5) respectively. The Y and X axes represent the forecasted and observed intensities in ERA5 reanalysis respectively.