# Peer review of "Subseasonal forecasts of Heat waves in West African cities"

_Natural Hazards and Earth System Sciences, 2023_

## Referee Comment (RC1)

**Review of:**

**Subseasonal-to-seasonal forecasts of Heat waves in West African cities, NHESS (Langue et al., 2023)**

The present study addresses the predictability of different heat wave types at the sub-seasonal time scale in West African cities over the period 2001-2020. The authors evaluate heatwave predictability in 2 state of the art sub-seasonal forecast models (46 forecast days), using two of the best reanalysis data sets available.

Overall, I find that the authors demonstrate their findings with a very nice sequence of Figures. However, sometimes important conclusions are derived about Tmax without demonstrating the Tmax plots, even though they can easily be added together with the respective Tmin in the Figures that exist in the main manuscript, or simply shown in the appendix.

Methodologically, I find very appropriate that the authors use several variables and several skill metrics in their analysis. Specifically, the authors proceeded with the evaluation of 3 variables of high interest and usability for heatwave definitions and heatwave research (Tmin, Tmax, Twet_bulb). The skill of the models to detect heat extremes is evaluated using the Brier score and the CRPS, while the predictability of heat waves in the forecast models is assessed by calculating categorical metrics such as the hit-rate, the Gilbert score, and the false alarm ratio (FAR). The results of this study (after Figure 4) are based on the percentile threshold selected for detecting extremes, as the authors create a 0-1 vector containing ones for days indicating extremes.

Unfortunately, I find that the percentile threshold selected for the detection of extremes is subjective and does not consider the model drift increase with lead time. The model drift is investigated in the following study and nicely shown in Figure 1: https://doi.org/10.1029/2019MS001751

The 90$^{th}$ percentile threshold the authors select is calculated over every calendar date and then the minimum percentile out of those is considered as a threshold for the full 46-day ECMWF forecast. The authors claim that they keep the selection of threshold constant, due to the relation of the study to a project investigating human impacts of climate extremes. However, this choice is still not justified, since this is a percentile threshold that the authors calculated. Normally human health impact studies do not use percentile thresholds, but actual temperature in degrees, e.g., 28 C. Moreover, this study is still a model evaluation study and, as the authors mention in the conclusions, the results will be used to investigate in detail the origins of the differences observed in the two forecast models over the different regions.

Normally, in sub-seasonal time scales, another 90$^{th}$ percentile threshold will be calculated for, e.g., August 1$^{st}$ for an initialization on July 31$^{st}$ and another 90$^{th}$ percentile threshold will be calculated for August 1$^{st}$ for an initialization on July 15$^{th}$. However, a technique like that is not followed, so the threshold selection of the authors leads in many cases to better model skill in terms of CRPS and Brier score at lead week-5 instead of lead week-2. Moreover, the authors claim that the results of lead week 1 and lead week 2 are similar, which is a discrepancy to other studies. The reason for this discrepancy might again be the threshold selected. Moreover, if the authors want to support such argument, they could at least show some figures in the appendix.

This study presents important research in the field of sub-seasonal prediction for a region that lacks evaluation studies. However, the authors' conclusions cannot be supported by the current analysis. My recommendation is to reconsider the manuscript after major revisions focused on a correct estimation of thresholds. A more detailed review per section is provided bellow.

**Abstract:**

1. The conclusions given in lines 10-15 are not supported by the current results. Even though the brier score is lower than 0.1 in many cases, from the metrics shown Figure 10 we can deduce that the model shows no skill in detecting heat waves. A general comment to that is the distinction made by the authors between heatwave detection and heatwave prediction. Isn't for a forecast model the heatwave prediction a synonym to a heatwave "detection"? All metrics used are valuable for the evaluation of the model, as this study shows that an apparently low CRPS and Brier score is not a synonym to the model's ability to separate extreme heat from non-extreme heat.

2. Please replace the expression "the model shows skills" throughout the manuscript.

**Introduction:**

1. I was very confused by the fact that the title and the introduction have a lot of material about the seasonal time scale and that the authors claim that they will evaluate this time scale as well. However, the authors use sub-seasonal forecasts going maximum to 6 weeks lead time, so 1.5 months. The authors should better define time scales in the introduction. The sub-seasonal time scale covers 2 weeks to 2 months. The definition can be found in the S2S project here:

https://public.wmo.int/en/resources/bulletin/subseasonal-seasonal-prediction-project-bridging-gap-between-weather-and-climate

Following this comment, I recommend that the authors should not refer on their intro or anywhere else in the manuscript to the seasonal time scale.

2. I really liked the part where the evaluation is done at the city scale. We do not normally see that is sub-seasonal prediction studies and it adds novelty to this study.

**Methodology:**

Line 214: Do the authors mean that they calculate the daily climatological 90th percentile threshold? So, this threshold should vary depending on the time of the year, right? Is the 90-percentile calculated separately for forecast model, reanalysis data, and station data?

The authors return to the explanation of the 90-percentile definition in the line 233 and state "…daily exceedances of daily values of indicators to the climatological daily threshold " which kind of agrees with the statement above but then in lines 236-238 they state: "Therefore, the climatological daily threshold is chosen to be constant over the whole period; and it is defined as the minimum of the daily climatology thresholds over the study period. This approach allows us to properly assess the severity of a heat wave and its potential human impacts." At the end what is it exactly that the authors do?

Also, as previously mentioned, a lead time dependent percentile should be considered.

Line 226: I thought the authors mentioned before in their manuscript that they assess separately wet and dry heatwaves. Why then your binary vector contains data from extreme values of all temperature variables?

Major comment for methodology: The predictability of a model should not be assessed only by comparing to a random chance, as this would not make a strong argument into using this forecast model. The authors should assess predictability by comparing to a reference forecast. For example, the Brier skill score could be calculated separately for two common reference forecasts, being the climatological forecast and the persistence forecast.

**Results:**

**Section 3.1:** Calculating forecast climatologies for a sub-seasonal forecasting system that provides forecast over sub-seasonal lead times (maximum 6 weeks) does not mean that the evaluation done here is an evaluation of seasonal forecasts (also the title of the manuscript states that).

The authors do not evaluate the seasonal predictability of the forecast system, as this model cannot provide a seasonal prediction. The authors basically provide the climatological biases of the sub-seasonal forecasting systems over the different seasons.

Seasonal forecasting means that the forecasting system provides at lead-zero more than 8 weeks forecast, which is completely different with calculating climatologies. Other than that, I find this section very interesting, as we can conclude during which seasons the sub-seasonal forecasts have the largest biases over west Africa.

**Section 3.2:** As seen in the supplementary material of the previous section, the climatology is lead time dependent and therefore crucial for understanding whether the model predicts an extreme or not. I think that this would significantly change the results on predictability of this study and the authors would also see important differences between lead week 1 and lead week 2. For example, the outcome stated in lines 335-336 ("We have noticed that the skill of the models does not improve necessarily with decreasing lead time.") is related to the choice of climatological distribution.

Here are some presentations from the ECMWF where they explain the calculation of the lead time dependent climatology:

Example for seasonal forecasts:

https://confluence.ecmwf.int/pages/viewpage.action?pageId=174864039

Model climate calculation in page 33:
https://resources.eumetrain.org/data/7/711/high_latitudes_ew_2023_s1c.pdf

Demonstration on how lead time affects EFI verification in page 34. Have a look on known issues on page 35, that is also why in the current study the Thursday initializations should not have been removed as it drastically affects sample size:
https://www.google.com/url?sa=t&rct=j&q=&esrc=s&source=web&cd=&ved=2ahUKEwiT2_GgwKeB AxWUh_0HHZtzD-oQFnoECA4QAQ&url=https%3A%2F%2Fconfluence.ecmwf.int%2Fdownload%2Fattachments%2F709 51731%2FForecasting_Extremes_Oct2017.pdf%3Fapi%3Dv2&usg=AOvVaw3YVlL5ZgSZsAzDf-LKfIzI&opi=89978449

Line 338: Could the authors explain why the forecast models show better predictive skill in the AT region?

**Section 3.3.1:**

Lines 354-355: The authors state "This approach based on a relative threshold (see section 2.4.3) will contribute to partially correct the biases previously found in the models." How does a variable daily threshold correct bias? A daily threshold simply accounts for differences between seasons.

408-409: The authors state "The forecast models show skills above the reference both for short- and long- term forecasts (Week2, Week5)"

I am wondering which figure shows that there is skill above the reference. Moreover, when the authors say skill, they should specify the metric to which they refer to. Taking into account the pairs of FAR and Hit_rate, Figure 10 shows no skill in detecting/predicting heat waves for none of the seasons/ models/ lead weeks.

410: Do the authors here mean instead of "more skills", "higher skill scores"?

Figure 10: Why is only the evaluation for T_min shown? The authors could add Tmax evaluation in Figures 10 and 11.

The authors state "We can infer from this result that nighttime heat waves are more predictable than daytime heat waves." Is there a Tmax figure I missed from which the authors infer this? Moreover, if that is indeed shown on a figure, is the reason for the higher predictability the actual $90^{th}$ percentile metric used which could be more variable in Tmax and that is why it appears less predictable.

Also, why do the authors not provide the evolution of the mean climatological biases between the forecast models and reanalyses for Tmax in the Appendix as provided for Tmin and Tw in S2 and S3?

The authors should avoid the expression "the models show more skills". It should be rephrased to "the models show higher Brier skill in …

415: What is an inter-day variability? Is it the std calculated over all your samples per region?

416-417 Can the authors explain more this this sentence: The low inter-day variability of T2m in the AT region indicates a more stable signal which will lead to favorable conditions for heat wave detection in the models based on a statistical perspective.

Wouldn't a stable signal lead to lower probability of extremes, so an even harder prediction of heatwaves?

Line 426: By "significantly decreasing" do the authors mean that they have calculated a level of significance? Do maybe the authors mean that the values are strongly decreasing? Or maybe they mean that the values are strongly decreased. In any case, GSS values that go from 0.2 to 0.1 are not strongly decreased, are low overall.

432: The use of the word "ability" in this sentence is misleading. Is it an ability to predict events that did not occur?

The authors provide the T_min variable in Figure 10 and in Figure S15. What is the difference in the 2 plots? Why there are no Tw and Tmax plots provided in the appendix? In Figure 10 which percentile threshold of ensemble members is used? The authors should add this in the figure caption.

445: How did the authors get to this conclusion: "The forecast models show skills at weekly time scale compared to the baseline climatology." Calculating the Brier skill score using the climatological forecast as reference could support (or not) this statement.

Figure 10 shows that, according to the evaluation done here, the model is over-forecasting heat waves. This can be even more explicitly shown if the authors plot hit rate against FAR to create the Roc curve. In this curve, the pair of 0.55 hit rate – 0.75 FAR will be below the diagonal. Being bellow the diagonal indicates no skill to discriminate between events and no-events, with the diagonal indicating random value/no-skill. This plot would disagree with the conclusion drawn by GSS.

http://www0.cs.ucl.ac.uk/staff/w.langdon/roc/

In the following ROC curve, I indicate with a yellow dot where your 0.55 hit rate – 0.75 FAR pair would be.

[Figure]

FIGURE 11: The markers have very small size, and so are hard to see. Here the authors could easily plot Tmax values as well, which is a very valuable parameter for heatwaves and its important to see its model biases here as well. I would suggest that the authors create one row for each variable (Tmin, Tmax, Tw) and maybe show both models in every sub-panel.

**Minor comments:**

Line 24 needs some references.

Line 37: That is a very big list of references, just to reference daily raw temperature as a variable relative for heatwaves. In the case of simple definitions, it would be helpful to keep the refence lists shorter and target to show the refences that are the most relevant.

Line 45-46 "This is usually done using seasonal weather forecast models." This sentence needs references. Also, since the study is also for sub-seasonal time scales it would be great if the authors motivate the sub-seasonal time scale and add references with studies connecting early warnings with the sub-seasonal time scale.
One reference I have in mind is:
Osman et al., 2023: Sub-seasonal to decadal predictions in support of climate services
DOI: 10.1016/j.cliser.2023.100397

Line 51: There is a published study on the predictability of extreme events across the globe:
Advances in the Sub-seasonal Prediction of Extreme Events: Relevant Case Studies across the Globe
Domeisen et al., 2022, DOI: https://doi.org/10.1175/BAMS-D-20-0221.1
Also, overall, for the onset/intensity/duration of European heatwaves at sub-seasonal time scales, where the calculation of lead-time dependent climatology is also explained:
Subseasonal predictability of onset, duration, and intensity of European heat extremes
Pyrina et al., 2022, https://doi.org/10.1002/qj.4394

Line 66: "Expert Team on Climate Change Detection and Indices (ETCCDI) database," also needs a reference.

Line 67: What is apparent temperature?

65-71: I find this paragraph very long and confusing especially because it is not going to be related to the method that will be used in the current paper. I would either remove it or keep a few sentences about it. Also, the paragraph goes on about what other studies did but there is no connection to what will be done here. The method of Lavaysse et al., 2019 is mentioned later, but it would be nice to also mention a few sentences about their method and why is good and you have followed it here.

Also, the Lavaysse et al., 2019 study could be mentioned before, when talking about the heatwave studies, so that the reader connects the study already with heatwave evaluation.

Line144: Please provide a citation from a publication or a book. Citations of webpages are not proper for research papers.

Lines 167-169: I do not understand the meaning of this sentence, was there a problem with the ECMWF output? Why would you choose to not evaluate all available initializations and reduce so much your sample size? A more accurate approach would be to evaluate the hindcasts for another operational model version, such as the model version of 2021, 2022, or even 2019.

Line 183: Here it is stated that the authors are "interested in the predictability of heat waves in a global perspective", which is confusing as it may be understood that the evaluation will be done for the whole globe. Also, what is stated in the sentence is not a good argument on why using different weekly initializations are comparable. The authors could just say that using ECMWF initializations on Thursdays leads to 4 initializations per month, making the sample size comparable to the UKMO model.

Figure 3: The authors cannot name a) for T2m_min and a) as well for winter. You could do i) t2m_min, ii) tm_max.

Figures 3,4: It should be mentioned in the figure that the authors consider all available lead times for this figure.

Figures 5,6:

1. For each of the variables investigated (Tmin, Tmax, Tw) when are there the stronger climatological biases? How much do they change if we consider a lead time dependent percentile threshold?

2. In many cases the skill increases with lead time, which is not common at all especially comparing forecast weeks 2 and 5. Is the change in skill driven by some particularly well

predicted period of extremes at lead week 5? Or maybe it comes by the fact that the authors define a common 90 percentile threshold for all lead times?

According to the error metrics of figures 5c and 6c, a conclusion would be that the users should trust the summer prediction over the CO region at lead week 5 more than at lead week 2!? This result is even more striking when looking at the winter season and Tw in figure S6.

Figure 7,8: The authors should explain what the grey values represent in the plots.

Line 322: Actually, there are systematic decreases and increases in biases with lead time in several subplots. For example, the bias is especially pronounced for ERA5 in the region CO (S2-a,f,h,e and in S3-everywhere). Why do some of the biases decrease with lead time? Can the authors explain some of these results or at least mention them?

Figures 7, 8 Why would the authors indicate a colorbar without units? Here it should be Bias (%)

Figure 9 Again here why having a colorbar without units? Especially with your duration definition that is very important. Please change to: Bias (days per year)

--The long text in each section is hard to read and makes it hard to return to a specific point when needed. Please separate the long text of each section in paragraphs. See here some tips: https://www.uvic.ca/learningandteaching/assets/docs/instructors/for-review/Information%20for%20Students/science%20paragraphs.DVG.FINAL.pdf

--Some figures have: "FAR ratio", but ratio is inside the word FAR anyway. Change to: "FAR".

479: How do the authors now that the models have issues with the "spatial evolution of heat waves"? Do they mean spatial variability?

**Conclusion section:**

This section should be rewritten after the revision of this study.

**Typos:**

Line 99: The references need brackets: Moron et al. (2016); Ngoungue Langue et al. (2023)

Line 182: Change "init dates" to "initialization dates"

Line 251: Change in the "The skill of the probabilistic models … are assessed..." to "…is assessed…"

Line 236: remove ;

---

## Author Comment (AC1)

Review of nhess-2023-144: **Subseasonal-to-seasonal forecasts of Heat waves in West African cities**

**Review overview**

The concept of this study is certainly of interest, and evaluation of extreme events such as heatwaves, which are increasing in frequency, intensity and duration, is important at a range of timescales. I read this paper with interest. This study examines the extended-range/subseasonal timescale, which can be useful for providing early indications of potentially hazardous events, ahead of more detailed forecasts that shorter-range forecasting systems are capable of predicting.

While I find the concept useful and interesting, and I like the range of skill scores used, I did find that several aspects of the methodology are not described clearly and there are some questions around the datasets and methodology used. Much more clarification is required around the forecasts used for this evaluation, and discussion of the potential drawbacks. The authors consistently refer to 'subseasonal to seasonal', which, while catchy, is not completely covered here, as the seasonal time frame is not included. The descriptions should probably be changed to subseasonal throughout. I also had some questions around the identification of heatwaves and the thresholds used, and the method of dealing with ensemble members. The abstract and introduction mention both wet and dry heatwaves, and daytime and nighttime, but these distinctions are not clearly defined and discussed, and appear to be mostly lacking from the rest of the paper and the results and conclusions.

While the research questions and results are interesting, I feel that the structure of the writing could be significantly improved throughout the paper, as it is currently challenging in places to follow the work, and to fully understand the somewhat contrasting conclusions. For a decision-maker, what are the takeaways to help understand how these forecasts could/could not be used in heatwave forecasting and anticipatory action?

Thank you to the reviewer for taking the time to revise this document. The comments made by the reviewer are very insightful. The predictability of heat waves is assessed using two sub-seasonal forecasting models involved in the S2S project, namely "ECMWF" and "UKMO". We totally agree with the reviewer that the forecast products do not cover the seasonal range, and we have replaced the term "sub-seasonal to seasonal" with "sub-seasonal" throughout the manuscript. Heat waves are evaluated in the models using dichotomous measures such as success rate, false alarm ratio and Gilbert skill score. Two types of heat waves are studied: wet heat waves are those resulting from a combination of humidity and air temperature, while dry heat waves are associated with maximum air temperatures. On the basis of the reviewer's comments, we have clarified some points in the main document and improved its quality. The results of the present study show predictive skills in sub-seasonal forecasting models up to two weeks in most cities of the region of study, however they overestimated the occurrence of heat waves. The results are useful for policy makers to develop early warning systems to prevent the population from potential heat waves. We added this aspect of the results to the main document.

I have provided some more specific comments on the text and some of the figures below. I hope these can be useful as the authors consider the revisions and next stages of the manuscript.

**Detailed comments**

**Abstract**

- Line 13: Short-term forecasts typically refers to those of <4 days – 2 weeks lead time would typically be classed as medium-range forecasting

  We replaced "short-term forecasts" in the manuscript by "medium range forecasts"

- Line 15: Fail is a strong word, without context?

We replaced "they fail in predicting the intensity of heat waves." by " the accurate forecasts of the intensity of heat waves remains challenging by the models"

**Introduction**

- Line 30-35: It could be worth mentioning that often, national meteorological services have a definition of a heatwave used to provide warnings? (unless it is not the case in the study region, but otherwise, there is also a WMO recommended heatwave definition (https://www.un-spider.org/category/disaster-type/extreme-temperatu re).

  We added the definition of heatwave given by WMO in the introduction "A period of marked unusual hot weather (maximum, minimum and daily average temperature) over a region persisting at least three consecutive days during the warm period of the year based on local (station-based) climatological conditions, with thermal conditions recorded above given thresholds."

- Same comment at line 41, research paper authors are not the only ones / the authoritative ones to define heatwaves, particularly in a forecasting perspective. Is there a definition used most often by the forecasting services based in the study region?

  We did not find a definition of heat wave provided by the forecasting services in the study area. They used the same definition provided by the literature review.

- Line 39: 'min, min or max' – is there a typo here? Min seems repeated

  That was a mistake. We replaced 'min, min or max' by " min, mean or max"

- Line 39: heat stress indices are mentioned, but not really defined anywhere? (check and come back to) – it may be useful to define here what a heat stress index is and how it differs from the other metrics listed

We added "The heat stress indices refers to indices resulting from a combination of some atmospheric variables useful to assess the human body comfort (wind speed, relative humidity, and incoming solar radiation) such as apparent temperature, Universal Thermal Comfort index, excess heat factor (EHF) and excess heat index (EHI) (McGregor et al., 2015)".

- Line 45: It is of course of crucial importance for early warning systems to provide information on the occurrence of heatwaves. However, early warning systems are not usually done using seasonal weather forecast models, which often lack the skill and resolution to accurately predict individual extreme events. Typically, an early warning system would refer to a shorter/medium-range lead time, supplemented with advanced information on the potential for hazardous weather using S2S forecasts. The authors go on to make this point about seasonal forecasts providing early indications, which I completely agree with, but early warning systems require a range of lead times, including shorter timescales to account for the fact that forecasts get much more accurate at shorter lead times.

Thanks to the reviewer for this clarification. We changed: " This is usually done using seasonal weather forecast models" by :

" In general, early warning systems integrated shorter and medium-range forecasts of potential weather hazards. This type of forecast window refers to sub-seasonal time scale from 2 up to 6 weeks. The sub-seasonal range is highly relevant for actions aimed at mitigating the human and health consequences of extreme heat [e.g. \citet{white2017potential,moron2018sub,tompkins2019predicting,osman2023sub}]. Sub-seasonal forecasts are used to monitor the evolution of specific weather patterns that have been identified in advance with seasonal forecasts. "

- Line 49-50: citation?

We added the following references "Bazo et al., 2019; Lala et al., 2020"

- Line 51-xx: I saw that Vitart (et al) also studied the Pacific Northwest heatwave of 2021, considering the ECMWF subseasonal forecasts and a more recent version of the ECMWF model, and 9 other S2S models. This may be of interest for the authors to include, as it uses a more recent model version than the Russian heatwave studies. https://agupubs.onlinelibrary.wiley.com/doi/10.1029/2021GL097036 Other authors also examined other time ranges of the forecasts for this heatwave.

  Thanks to the reviewer, we added this work in the manuscript.

- Line 114: ERA5 is used to initialise the extended-range **reforecasts**, but not all of the different aspects of ECMWF's IFS

  This is a good remark.

  We replaced "Since ERA5 is used to initialize the atmospheric component of the ECMWF model which is one of the forecast models" by

  " Since ERA5 is used to initialize the atmospheric component of the ECMWF extended reforecasts (ENS-ext) "

**Section 2**

- Section 2.1: it could be interesting in this section to include some description of heatwaves and their impacts in this region – what have been the impacts of significant past heat waves? What kinds of temperatures are reached? During what season do impactful heatwaves occur?

  We added this information to the manuscript :

  "In April 2010, North Africa was affected by a severe heatwave, with daily maximum and minimum temperatures exceeding 40°C and 27°C respectively over a period of 5 days. This event was disastrous for the population and caused significant material damage. In May 2013, the Senegalese city of Matam, experienced an intense heat wave with temperatures sometimes reaching 50°C in the shade. The event was

persistent both during the day and night, and it caused 18 deaths among the elderly people in 10 days. Mauritania also experienced a devastating heatwave in May 2013, with maximum temperatures exceeding 46°C, causing the death of more than 25 elderly people and children."

- Section 2.1: this section refers to the region having a short wet season followed by a long dry season. But the results later are split into winter/spring/summer/autumn – this should be further explained and justified.

We added this information in the document:

"The evaluation of the skills of the models to predict heat waves is carried out from January to December to cover the wet and dry seasons in the region. The results were then splitted into four sub-seasons to assess the intra-seasonal variability of the skills of the models."

- Section 2.2: A brief discussion on the potential disadvantages of reanalysis datasets might be warranted, if not included later (e.g. they may not always have the resolution to be able to pick up the very highest temperatures during heatwaves)

Thanks to the reviewer for this suggestion. We added the following to the manuscript: " Another point to highlight in this work is the use of reanalyses to evaluate heatwave forecasting models. We are aware that reanalysis data have a high resolution compared to observations from local stations. As a result, they are unable to represent the urban heat island effect which exacerbates heat stress during heat waves. The resolution of the reanalyses makes it impossible to detect the highest temperatures at specific locations.

Section 2.2.2: The assessment of dry and wet heat waves seems like it should be a separate section, as at the moment it seems to be that it is only applied to MERRA, as it sits under that subtitle. It is also not clear

how the heatwave identification described later takes into account both wet and dry heatwaves, nor is this clear in the results. Wet and dry heatwaves should also be defined (do they refer to the afore mentioned wet and dry seasons? to the humidity experienced during a heatwave? Or otherwise?)

We clarified this clarification in subsection 2.1 Region of interest

" In this study, two types of heat waves are analyzed :  dry and wet heat waves. Dry heat waves are mostly driven by incoming solar radiation and occuring during the day. The detection of dry heat waves is processed using maximum values of T2m as indicator. Wet heat waves are the most lethal for human health. Humidity is an important driver of wet heat waves. The detection of wet heat waves is done by using minimum values of T2m and mean wet bulb temperature as indicators. "

- ○ Section 2.3.1 ECMWF forecasts: there are some errors in this section, and I find some of the description unclear. This isn't helped by the fact that ECMWF very recently upgraded their models so some of this information no longer stands. It may be useful to revisit the description in this section for clarity.

    Thanks to the reviewer for this remark

    More details below:

- ○ The ECMWF IFS has several separate forecasting systems (medium-range (now high-resolution), extended-range, seasonal), and it would be useful to specify which is being used and described here, as other parts of the system have different resolutions, lead times and ensemble members.
- ○ ECMWF provides both extended-range (up to 46 days) and seasonal (up to 6 months) forecasts to the S2S programme. I understand that the authors are using the extended-range forecasts, and it may be useful to refer to the forecasts as this throughout.

- ○ 'ECMWF ENS' is often used to refer to the medium-range (up to 15 days and now high-resolution) ensemble, and so could cause confusion – these are not the same exact forecasting system as the extended-range (at least not any more).
- ○ The authors may wish to specify that they are interpolating to a 0.25° grid to match the resolution of ERA5 for evaluation (I presume), with the caveat that this does reduce the resolution of the native forecasts, and that higher resolutions can be beneficial for capturing extremes.
- ○ The IFS is no longer running at CY41r2. If the authors downloaded hindcast data from forecast dates in 2021, the cycles could have been 47r1 (implemented 30 June 2020), 47r2 (implemented 11 May 2021), or 47r3 (implemented 12 October 2021). The authors should confirm which cycle(s) was used. These cycles indeed had 51 ensemble members at 18 and 36km resolution depending on the lead time, as the authors describe. The latest version of the extended-range forecasts (48r1) has 101 members, run at 36km for the full forecast range (days 0 to 46), and is run daily rather than twice a week. (https://www.ecmwf.int/en/about/media-centre/news/2023/model-upgrade-increases-skill-and-unifies-medium-range-resolution)
- ○ A useful description of hindcasts/reforecasts may be 'Hindcasts are forecasts produced for past dates using the most recent version of the forecasting system, and allow analysis of how the current system would have performed, alongside a consistent dataset covering a longer time period for evaluation', as a useful use of these data for the authors' purposes?
- ○ The authors also use 'hindcasts' and 'reforecasts' interchangeably. ECMWF typically call them reforecasts, and the authors should be clearer if it is themselves calling them 'hindcasts' throughout the study.
- ○ I didn't understand the explanation for not using the Thursday hindcasts, sorry

We modified this section according to the reviewer's remarks. The new section is the following :

" 2.3.1 ECMWF forecasts

The extended-range ECMWF forecast model runs on the Integrated Forecast System (IFS) cycle CY47R3 released on October 10[th], 2021. The native spatial resolution of the ECMWF model is Tco639 L137 (about 16 km) up to day 15 and Tco319 (about 32 km) after day 15, but the downloaded data are interpolated to a regular 0.25°x 0.25° latitude/longitude grid to match the resolution of ERA5 for evaluation. It contains 91 sigma levels from the surface to 80 km. ECMWF provides two types of outputs for the S2S program: real-time forecasts and reforecasts called "hindcasts". Real-time forecasts are forecasts for the coming days. Hindcasts are forecasts produced for past dates using the most recent version of the forecasting system, and allow analysis of how the current system would have performed, alongside a consistent dataset covering a longer time period for evaluation. ECMWF extended-range real-time forecasts are run with 51 ensemble members (50 perturbed and 1 unperturbed), while hindcasts are run with 11 members. In this study, we focus on hindcasts only. ECMWF extended-range hindcasts are produced twice a week, on Monday and Thursday at 00Z. This means that for each week a new set of hindcasts is produced to calibrate the real-time ensemble forecasts for Monday and Thursday of the following week using the latest version of the IFS. We only analyzed the hindcasts produced on Thursday. This is because we firstly want to carry out a multi-model analysis. According to a first investigation on the initialization dates of the hindcasts of different models, we found that most of the models were initialized on the same date as ECMWF (Thursday of each week) but did not cover the study period. The 11-member ensemble hindcasts start on the same day and month as the real-time forecast, but covering the last 20 years. In our case, the forecast year is 2021 and we focus on the previous 20 years from

that date, and the hindcasts run from 0-46 days. The variables of interest in the ECMWF S2S are T2m(max,min) over the last 6 hours, daily average T2m and d2m from which the daily average Tw was derived. The data are open access and available on the S2S project website (https://apps.ecmwf.int/datasets/data/s2s-realtime-instantaneous-accum-ecmf/levtype=sfc/type=cf/)."

- ○ Line 83: the authors may wish to acknowledge that when dealing with extreme events, including different extreme events in the analysis may well result in different conclusions regarding the skill.

  Yes, we agree with the reviewer, but we didn't catch the link to the present.

- Section 2.3.2: Parts of the UKMO forecast description are also confusing, for example the transition from discussing 4 members to 7 members. Perhaps a table outlining key aspects of both forecasting systems, and the timeframes to which they apply, would be helpful to provide an overview of the system characteristics?

We clarified this point in the manuscript by replacing :

" The UKMO real-time forecast consists of a set of 4 members run daily for a period of 60 days (3 perturbed members and 1 control member). The UKMO hindcasts are produced 4 times per month, on the 1st, 9th, 17th and 25th, and cover a 24-year period from 1993 to 2016. We are aware that these initialization dates are not the same as those of ECMWF, but we are interested in this work on the predictability of heat waves in a broad perspective, not on specific events. The ensemble hindcasts are composed of 7 members per cycle (from the 25 March 2017 hindcasts, prior to that 3 members per cycle)." by

"The UKMO real-time forecast consists of a set of 4 members (3 perturbed members and 1 control member) run daily for a period of 60 days. The UKMO hindcasts are produced 4 times per month, on the 1st,

9th, 17th and 25th, and cover a 24-year period from 1993 to 2016. We are aware that these initialization dates are not the same as those of ECMWF, but we are interested in this work on the predictability of heat waves in a broad perspective, not on specific events. Prior to 2017, specifically on March 25$^{th}$, the UKMO ensemble hindcasts were composed of 3 members per cycle (2 perturbed and 1 control). Since 2017, the number of members has increased from 3 to 7 (6 perturbed and 1 control)."

- Section 2.3.2: I believe the description of the concatenation could be simpler. Is an equation necessary, or is it enough to simply state that prior to 2016, hindcasts are used, and after that, the real-time forecasts are used, followed by the details from line 190?

We agree with the reviewer that the description of the concatenation applied here can be simplified, but through this equation we want to highlight the complexity behind this data processing task. We will keep the equation in the document.

- I am not completely convinced of the decision to reduce the number of ensemble members in this methodology, thus reducing the uncertainty representation of the forecast.

We added this explanation to the main document :

"In order to apply the concatenation over time between the re-forecasts and real time forecasts, the coordinates dimensions of the two datasets must be the same. As shown early, the number of ensemble members in UKMO re-forecasts and real time forecasts are completely different. Therefore, to meet this requirement, we reduced the number of ensemble members from 7 to 4 (1-control member and 3-perturbed members)in the re-forecasts to match the number of ensemble members in the real-time forecasts."

It would also be useful to provide an overview of how other characteristics of the model have changed between the hindcast version and the potentially multiple operational versions used during

the period of the real-time forecasts? This could be covered in the aforementioned table.

We added the following table in the document.

| Models | Hindcasts | | | | Real time forecasts | | | |
|--------|-----------|------|-------|--------|-----------|------|-------|---------------|
| | *dates* | *size* | *range* | *period* | *dates* | *size* | *range* | *Model version* |
| **ECMWF** | 2/week, on Monday and Thursday | 11 | 0-46 days | past 20 years | 2/week, on Monday and Thursday | 51 | 0-46 days | CY48R1 |
| **UKMO** | 4/month on the 1st, 9th, 17th, 25th | 3 prior 2016 7 from 25/03/2017 | 0-60 days | 1993-2016 | 4/month 1st, 9th, 17th, 25th | 4 | 0-60 days | GloSea5-GC2-LI |

- Line 197: I believe here the authors are referring to a lack of data from local stations to evaluate the forecasts again. The sentence implies that no data is available from this region for weather forecasts to assimilate in their production – are the authors sure this is the case? Particularly since weather forecasts also use various other sources of observations beyond station data.

We clarified this point in the manuscript as follow:

We replaced: "Weather forecasts provide the evolution of atmospheric variables on a global scale, which implies the need to have data from local stations to access information on a local scale. This is a major problem in areas where there is a lack of weather stations to collect data, as is the case in African cities." by

"Weather forecasts provide the evolution of atmospheric variables on a global scale, which implies the need to have data from observation stations to access information on a local scale. This is a major problem in areas where there are not enough weather stations to collect data, as is the case in African cities. Nevertheless, when observation stations are available in the region, access to the data collected remains difficult."

- Section 2.4.2:
  - Are the daily maximum, daily minimum and wet bulb computed from the hourly data? Or otherwise?

    We added this information to text :

    "Daily maximum and minimum temperatures are computed respectively from maximum and minimum temperatures in the last 6 hours. This choice of the computation of the extreme daily values is made according to the forecast models outputs. Daily average wet bulb temperature is computed from hourly dew point temperature."

  - Are nighttime and daytime heatwaves considered separately, or as one continuous heatwave that does/does not provide relief overnight? This can have implications for heat stress and health, but it is not clear how it is factored into the authors' definition of a heatwave. I think it is hinted at, but was not entirely clear to me in the definition.

    We clarified this point in the manuscript:

    "Nighttime and daytime heat waves are considered separately in the study. Nighttime heat waves are detected using minimum values of indicators, while for daytime heat waves, maximum values of the indicators are used. "

  - Is the 90$^{th}$ percentile representative of the health impact of heatwaves on humans / ecosystems?

    Thanks to the reviewer for this interesting remark.

    According to previous studies on heat waves and their impacts on human health, the 90th percentile appears to be a sufficient threshold for heat waves detection (e.g., Fischer and Schär,

2010; Perkins et al., 2012; Perkins and Alexander, 2013; Fontaine et al., 2013; Russo et al., 2016; Lavaysse et al., 2018; Ngoungue et al., 2023).

○   What if the 90th percentile does not reach a temperature likely to cause heat stress? Why not use a temperature or wet bulb threshold known to cause health impacts in this region?

We added more explanations about this point in the manuscript :

"Heat waves in the Sahel region occur mainly in spring due to the high temperatures in the region at that time \citep{barbier_detection_2018,guigma_characteristics_2020}. In this study, the region of interest was extended to the Guinean region in which heat waves are mainly driven by humidity. Heat wave detection was then carried out using the $90^{th}$ percentile as a threshold over the January to December season. The $90^{th}$ percentile appears to be a sufficient threshold for monitoring heat waves affecting human health. Nevertheless, it is useful to calculate the intensity of events in order to determine a classification according to their severity (intensity), from "harmless" to "extremely dangerous", for example. This is what the STEWARD project is doing by developing a database on heat waves and their potential impact on human health."

○   Line 213-214 states the 90th percentile is computed over the entire period, and then line 215 says it's calculated for each day of the year. I am left unsure as to which of these is used (or which is used for which analysis, if both are used at different points), and this could be quite impactful for the results.

We clarified this point by replacing :

"We defined a heat wave as  a consecutive period of at least 3 days during which the daily temperatures exceed the calendar 90th percentile threshold computed over the entire period for

T2m_min, T2m_max or Tw respectively. The 90th percentile is calculated for each calendar day of the year." by :

" We defined a heat wave as a consecutive period of at least 3 days during which the daily temperatures exceed the calendar 90th percentile threshold computed over the entire period for T2m_min, T2m_max or Tw respectively."

- Section 2.4.3; the description of these steps could be simplified and clarified further. The two first points may not really be necessary to spell out, and the third could perhaps be simplified, but it is also not clear over which timeframe this is done. Is it done for each day of the time series?

We clarified this point by replacing :

"To determine the occurrence and duration of heat waves, we create boolean files from T2m_min, T2m_max and Tw time series at each grid point following the steps below :

for days in T2m_min, T2m_max or Tw time series, if days are hot days, we replace in our zero vector the values corresponding to those days by 1. Hot days are days with T2m_min, T2m_max or Tw above the 90th percentile daily thresholds. In order to assess the characteristics of heat waves, only hot days belonging to heat wave sequences are kept. This is applied for all grid points and we obtain boolean files containing 0 or 1 (Ngoungue et al. 2023). These boolean files will be processed both for the reanalyses and the forecasts to assess the representation of heat waves occurrence and duration." by

"To determine the occurrence and duration of heat waves, we create individual boolean files from the T2m_min, T2m_max and Tw time series at each grid point, containing 1 if hot days and 0 otherwise. This operation is performed on a daily time scale over the study period. Hot days are days on which the values of T2m_min, T2m_max or Tw are above the daily 90th percentile thresholds. In order to assess the characteristics of heat waves, only hot days belonging to heat wave

sequences are considered (Ngoungue et al. 2023). Boolean files are calculated separately for reanalyses and forecasts in order to assess the representation of heat wave occurrence and duration."

- And then if the number of hot days in a row is not >=3, the value is returned to 1?

  No, as we define a heat wave as at least 3 consecutive hot days, if the number of hot days in a row is not >=3, the value is returned to 0.

- Line 232-233 isn't clear to me, apologies.

  We clarified this point in the manuscript, by replacing:

  "The intensity of heat waves was defined as the cumulative sum of the daily exceedances of daily values of indicators to the climatological daily threshold in a sequence of hot days " by

  "The intensity of a heat wave was defined as the sum of the daily exceedances of the indicators values to the climatological threshold during the event."

- Line 236-238: the reasoning behind this, and how this is applied in the methodology, isn't clear to me. Why the minimum of the daily thresholds? Does this correspond to a value that is certain to have an impact on human health? How does this allow proper assessment of the severity? Please expand on this. This relates to a previous point about using percentile thresholds, when using set values corresponding to heat stress may be both simpler and more effective.

  We clarified this point by adding this information in the document :

  " This study is in the framework of the project Agence National de la Recherche STEWARd (STatistical Early WArning systems of weather-related Risks from probabilistic forecasts, over cities in West Africa) project which focuses on the human impacts of climate extremes. We are therefore interested in heat waves, which can be harmful to human health. To do so, the climatological daily threshold is chosen to be constant over the whole period for the computation of

heat waves intensity. It is defined as the minimum of the daily climatology 90th percentile over the study period. This approach allows us to properly assess the severity of a heat wave and its potential human impacts, therefore, most dangerous heat waves will have higher intensity values."

- Line 243: ensemble forecasting does not only account for uncertainties in the physical component of the model, but also uncertainty arising from the chaotic nature of the atmosphere, and from an imperfect observation network and therefore imperfect initial conditions of the forecast.

  Thanks to the reviewer for the suggestion, we added this information to the manuscript.

- Line 247-249: By considering the mean, medium, warmest, coolest, $1^{st}$ and $3^{rd}$ ensemble members, you have identified 6 'members'. The Met Office forecasts only have 2 or 7 members, and the ECMWF forecasts 11 members, so I am unsure as to why it is less challenging to use these 6 'members' chosen by the authors, rather than more usefully examining the entire ensemble and therefore the full range of uncertainty represented by the ensemble? It should also be considered that the mean (and quartiles, depending on how these are produced) do not represent an actual forecast scenario or physically likely state of the atmosphere, produced by the model, and so caution is required in assessing this both as a forecast and in evaluating it.

  Here, we don't compute any of these statistics, we just want to show how it is difficult to evaluate ensemble forecast systems based on the amount of information they provide (mean, median, warmest, coolest, 1st and 3rd quartiles of the ensemble members, ensemble members).

**Section 3**

- some paragraphs would be helpful for readability in sections 3,4,5

  We added some text to facilitate the comprehension of sections 3,4 and 5

- Section 3.1: the use of 'hot bias' and 'cold bias' is quite strong wording, as opposed to positive and negative. How large are the biases? It is not mentioned in the text,but some of these 'hot' biases may only be a small fraction of a degree, so hot might not be the most appropriate choice of wording?

  The bias found in the study varies with the variables, with T2m the bias is around -4 and 4 K, while it is more important with Tw between -12 and 0 K. We replaced the terms 'hot bias' and 'cold bias' in the manuscript by 'positive bias' and 'negative bias' respectively.

- Given that the authors state that the results comparing to MERRA are significantly different to those using ERA5, I am surprised not to see some figures included in the main text. How does this discrepancy impact the evaluation results, if the two verification datasets are so different?

  We have noticed some differences between MERRA and ERA5 when carrying out some analyses on the forecasting models, for example the estimation of the bias in the evolution of T2m and Tw, and the spatial variability of heat wave duration. However, the assessment of the CRPS and the Brier remains similar for both reanalyses. Therefore, for the evaluation of the predictability of heat waves in the models, we used ERA5 as a reference. We agree that the heat wave predictability results for MERRA will not be exactly the same as those obtained with ERA5, but the evaluation metrics will be of the same order. Some figures with MERRA are added to the manuscript.

  Line 321: the plots are shown in °C, but the text uses K – why refer to it differently between the texts and figures?

  That's a good point. It's a mistake that we've corrected.

- Section 3.2: Why are the Tw results only shown in supplementary material if they make up an important part of the research question and results?

That's a good point. We added some results on the Brier score, spatial variability of heat waves duration and the metrics evaluation with Tw in the main text.

- Section 3.3.2: Why is the mean duration the sum divided by the number of affected years, rather than divided by the number of heatwaves? (what if there is more than one heatwave per year?)

  We are interested here in the average characteristics of heat waves over the period in which they occurred. As a result, the average duration of heat waves corresponds to the sum of heatwave days divided by the number of years concerned.

- Line 375: can the authors comment on the representation of convection in both models?

  "The representation of the convective activity in ECMWF, is done using the Tiedtke scheme (Tiedtke, 1989) and UKMO, the Met Office convective scheme (Hagelin et al., 2017)."

- Section 3.4: could the authors explain further the reasoning behind the 20%, 40% and 60% percentile thresholds? I did not follow the aim and reasoning here. The text and Figure 10 seem to refer to a section of the methods that I was unable to find. Perhaps it refers to the last sentences of section 2.4.4, but I did not follow the link, and further explanation may be required.

  We clarified this point in the manuscript by added :

  "The results presented below are obtained using a 20\% threshold value to optimize the ensemble forecast system (see Section 2.4.4) ."

  An initial evaluation of the forecast models was performed using all ensemble members, and the metrics (hit_rate, FAR, GSS) were calculated for each member. Then, the evaluation metrics are calculated as the average of the metrics for all members. This approach is not suitable for model evaluation. In order to optimize the forecast systems, we transformed the probabilistic forecasts into

deterministic forecasts using threshold values, following the methodology proposed by Lavaysse et al. 2019.

- Regarding seasons, are there seasons where there may technically be heatwaves as the temperature exceeds the 90$^{th}$ percentile for the time of year, but they would not cause heat stress or health impacts? Should these be considered in the same way as those during other seasons? Why are winter/spring/summer/autumn used if the region experiences two seasons (dry/wet) – how do these correspond?

These questions have been clarified previously ( see the second part of the discussion in the manuscript).

Some context regarding heatwaves themselves and the temperatures reached and impacts in this region could provide interesting further insight (for example in section 2.1 this could be added).

This has been done previously in section 2.

- From a decision-making perspective, it would be interesting to understand how far in advance these forecasting systems may be able to provide a useful prediction/indication of a heatwave. The results are interesting from a modeling perspective, but I finish reading the results section feeling that I would not really have a confident answer to this question. Could the discussion be expanded to consider the results in this context?

We added this part to the discussion :

" This study showed that the forecast models were actually able to predict heat waves occurrence up to two weeks in advance in the different regions. On the other hand, we found that the models overestimated the frequency and duration of events, whatever the lead time. Consequently, it will be necessary to find a good balance between hits and false alarms in order to develop a robust early warning system to prevent populations from heatwaves. "

**Section 4**

- Line 456-460: the names of the convection schemes unfortunately do not mean much to me – what are the key differences and the implications?

We added this information to the main document :

"The Tiedtke convection scheme is one of the first mass-flow convection schemes, which aims to parameterise the effects of deep convection in numerical weather models. It simulates the vertical transport of heat, moisture and momentum associated with convective updrafts and downdrafts. The system takes into account various factors, including atmospheric instability, moisture content and boundary layer conditions to estimate convective processes. UKMO also uses a mass flux convection scheme, but different from the Tiedtke scheme, which takes into account atmospheric instability and moisture content to determine convective activity. The difference between the two convective schemes could lead to a wrong representation of convective activity in the region, and thus limit the predictive skills of the models mostly for wet heat waves."

- Line 461: could the authors expand on 'the data and initial conditions are completely different' ?

We added this information to the main document :

"ECMWF assimilates a wide range of global and regional observational data, including satellite, radar and ground-based measurements. The UKMO focuses on observation data relevant to the United Kingdom and surrounding regions. ECMWF uses a 4D-Var assimilation which considers the temporal dimension (four dimensions) in addition to the three spatial dimensions to generate the initial condition. The UKMO

employs two data assimilation techniques : the 4D-Var and Ensemble Variational (En-Var) to estimate the initial state of the atmosphere."

- Line 465: did the authors not reduce the resolution of both forecasts? What impact could this have? Particularly on the discussion of all results relating to the spatial variability and the intensity

  We added this information to the main document :

  "The native resolution of the models has been transformed into a regular 0.25*0.25 grid. Even if we transform the native resolution of the two models into a regular 0.25°x0.25° grid, some local-scale patterns will be found in the new grid. However the impact of the resolution on the skills of the models is not assessed."

- Overall, I find the discussion section raises some interesting points, but does not really expand on why or how they influence the results

  In fact, in this study we have identified some differences between the two models that may explain the differences in predictions in the regions. A more detailed analysis of the influence of each factor on the results is beyond the scope of this paper.

**Section 5**

- Line 484-485: it was not clear where the key results were that make any distinction between daytime and nighttime heatwaves and how this was handled in the methodology. An interesting aspect of heatwaves is the drop in temperature overnight, and whether this provides any relief from the daytime heat stress, but this doesn't factor into the discussion at all.

  This has been clarified in the previous comment on the definition of daytime and nighttime heat waves(see Section 2.1).

  We added this to the manuscript :

  "The prediction of dry heat waves is slightly better with ECMWF for medium range forecasts, while it is better with UKMO for long-range

forecasts. For wet heat waves, UKMO outperforms ECMWF for both medium- and long-range forecasts."

- What do the authors consider as a nighttime heatwave, one that only occurs at night and not also in the day? It is a little confusing, and more context and insights could probably be included.

  We clarified this point in the previous comment. Yes, nighttime heat waves are those occurring during the night. In the document, we decided to replace "nighttime" and "daytime" heat waves by "wet" and "dry" heat waves to avoid some confusions.

- On a similar note, it is not clearly defined the difference between a wet and a dry heatwave, other than the use of different variables. These terms are only really use in the introduction and conclusions, but the link to the results is missing and the methods are not entirely clear.

  We clarified this point in the previous comment.

- Line 491: what is counted as a failure to predict the intensity? At what lead time? This is a very broad statement.

  We changed " they fail in predicting the intensity of heat waves; the accurate forecast of heat waves intensity remains a challenging task for the models." by :

  "They underestimate the intensity of heat waves with respect to ERA5 at short, medium and long range forecasts. "

  It implies the forecasts are not useful at all – do the authors conclude in this paper that extended-range forecasts are not useful for predicting heat waves? Can they be used or interpreted at all to complement short-range forecasts? Can some information be provided on the skill of short-range forecasts (could be from other studies), to provide context?

  This does not mean that forecasts are not useful at all; they show skill in capturing some heatwave events at medium and long range. We added this information to the main document :

*"Regarding these results, we can recommend the use of subseasonal forecasts to predict the occurrence of heat waves up to two weeks in advance, but as far as their intensity is concerned, it is still challenging."*

How do the authors tie in these results, with the earlier statements that based on some skill scores, the models can detect extreme events up to 5 weeks ahead? Detect in what sense?

We have clarified this point in the manuscript, the term "detect" used here is a synonym for "forecast". As heat waves are defined as persistent extreme events, we first assessed the representation of single extreme events in the forecasts. To do this, we calculated the Brier score, and this first assessment does not take into account the persistence of the events. The second assessment concerns the predictability of heat waves in the models, by calculating the hit-rate, FAR and GSS.

**Figures**

Fig 3: The use of (a) and (b) for both the upper and lower panels and the individual panels is a little confusing at first. Perhaps consider (i) and (ii) for the panels? (or just upper and lower?), or split this into two figures.

We replaced (a) and (b) by (i) and (ii) according to the reviewer comment

Fig. 4: The colour scale here is misleading – it should be adjusted so that the white colour falls at 0, with positive and red and negative in blue, otherwise it is very challenge to properly assess where there is a warm/cold bias, particular with a gradient rather than discrete colour bar. The colour scale should reach the same value at the positive and negative ends.

This has been done according to the reviewer comment and the new figure is the following.

[Figure]

Figure4: Spatial variability of the climatological bias between the forecast models ensemble mean and ERA5 reanalysis over the period 2001-2020 for Tw during the seasons : (a,e) winter; (b,f) spring; (c,g) summer and (d,h) autumn. The bias is computed as the difference between the forecast models and ERA5. The color indicates the bias values in degrees Celsius. The X and Y axes represent the longitude and latitude respectively.

Figures 7, 8: Again, it appears that the colour scales are not covering the same range for the positive and negative ends, and therefore the white colour doesn't represent 0. This can be misleading for the interpretation and should be fixed so that the scale is the same at each end.

We changed figures 7 and 8 according to the reviewer comments.

[Figure]

(i)

[Figure]

(ii)

Figure 7 : Spatial variability of heat wave frequency bias between forecast models and ERA5 over West Africa from 2001 to 2020 for:(i) T2m_min values and (ii) T2m_max values, during: (a,e) winter; (b,f) spring; (c,g) summer and (d,h) autumn. The bias is calculated as the difference in heat wave frequency between the forecast models and ERA5. This analysis is performed using the unperturbed member of the models. The color bar indicates the bias values without units. The X and Y axes represent longitude and latitude respectively. The solid blue lines indicate the borders between countries; the black dots represent the cities of interest for this study (this applies to the rest of the paper).

[Figure]

Figure 8 : Spatial variability of heat wave frequency bias between forecast models and ERA5 over West Africa from 2001 to 2020 using Tw during: (a,e) winter; (b,f) spring; (c,g) summer and (d,h) autumn. The bias is calculated as the difference in heat wave frequency between the forecast models and ERA5. This analysis is performed using the unperturbed member of the models. The color bar indicates the bias values without units. The X and Y axes represent longitude and latitude respectively.

---

## Author Comment (AC2)

**Review of:**

**Subseasonal-to-seasonal forecasts of Heat waves in West African cities, NHESS (Langue et al., 2023)**

The present study addresses the predictability of different heat wave types at the sub-seasonal timescale in West African cities over the period 2001-2020. The authors evaluate heatwave predictability in 2 state of the art sub-seasonal forecast models (46 forecast days), using two of the best reanalysis data sets available.

Overall, I find that the authors demonstrate their findings with a very nice sequence of Figures. However, sometimes important conclusions are derived about Tmax without demonstrating the Tmax plots, even though they can easily be added together with the respective Tmin in the Figures that exist in the main manuscript, or simply shown in the appendix.

Methodologically, I find very appropriate that the authors use several variables and several skill metrics in their analysis. Specifically, the authors proceeded with the evaluation of 3 variables of high interest and usability for heatwave definitions and heatwave research (Tmin, Tmax, Twet_bulb). The skill of the models to detect heat extremes is evaluated using the Brier score and the CRPS, while the predictability of heat waves in the forecast models is assessed by calculating categorical metrics such as the hit-rate, the Gilbert score, and the false alarm ratio (FAR). The results of this study (after Figure4) are based on the percentile threshold selected for detecting extremes, as the authors create a 0-1 vector containing ones for days indicating extremes.

Unfortunately, I find that the percentile threshold selected for the detection of extremes is subjective and does not consider the model drift increase with lead time. The model drift is investigated in the following study and nicely shown in Figure 1: https://doi.org/10.1029/2019MS001751.

We agree with the reviewer that in general, forecast models present a drift which increases with the lead time. The selection of 90th percentile threshold is supported by previous studies on heat waves and their impacts on human

health (e.g., Fischer and Schär, 2010; Perkins et al., 2012; Perkins and Alexander, 2013; Fontaine et al., 2013; Russo et al., 2016; Lavaysse et al., 2018; Ngoungue et al., 2023).

The 90th percentile threshold the authors select is calculated over every calendar date and then the minimum percentile out of those is considered as a threshold for the full 46-day ECMWF forecast. The authors claim that they keep the selection of threshold constant, due to the relation of the study to a project investigating human impacts of climate extremes. However, this choice is still not justified, since this is a percentile threshold that the authors calculated. Normally human health impact studies do not use percentile thresholds, but actual temperature in degrees, e.g., 28 C. Moreover, this study is still a model evaluation study and, as the authors mention in the conclusions, the results will be used to investigate in detail the origins of the differences observed in the two forecast models over the different regions.

We want to clarify this point. The detection of heat waves is processed using the daily climatological 90th percentile computed over the study period as threshold. After the detection of heat waves is addressed, we compute their intensity using a constant threshold defined as the minimum daily climatological  90th percentile. The intensity is defined as the sum of the daily exceedance of the indicators from this constant threshold.

Normally, in sub-seasonal time scales, another 90th percentile threshold will be calculated for, e.g., August 1st for an initialization on July 31st and another 90th percentile threshold will be calculated for August 1st for an initialization on July 15th . However, a technique like that is not followed, so the threshold selection of the authors leads in many cases to better model skill in terms of CRPS and Brier score at lead week-5 instead of lead week-2.

That's a good point. In fact, that's what we've done: the 90th percentile threshold is calculated separately for each initialisation of the models. For example, when using UKMO, the threshold is calculated four times a month according to the different initialisation dates (1st, 9th, 17th and 25th). Heatwave detection is also processed at different initialisation dates. The

results were quite surprising, but this behavior in the models can be explained by the fact the predictability limit is reached too early after week 2.

Moreover, the authors claim that the results of lead week 1 and lead week 2 are similar, which is a discrepancy to other studies. The reason for this discrepancy might again be the threshold selected. Moreover, if the authors want to support such argument, they could at least show some figures in the appendix.

We added some figures on the evolution of evaluation metrics for different weeks.

This study presents important research in the field of sub-seasonal prediction for a region that lacks evaluation studies. However, the authors' conclusions cannot be supported by the current analysis. My recommendation is to reconsider the manuscript after major revisions focused on a correct estimation of thresholds. A more detailed review per section is provided below.

Thanks to the reviewer for taking the time to revise this paper.

**Abstract:**

1. The conclusions given in lines 10-15 are not supported by the current results. Even though the Brier score is lower than 0.1 in many cases, from the metrics shown Figure 10 we can deduce that the model shows no skill in detecting heat waves.

We do not agree with the reviewer on this point. The conclusions given here are supported by the results obtained with the Gilbert skill score in Figure 10, which in many cases shows values greater than zero.

A general comment to that is the distinction made by the authors between heatwave detection and heatwave prediction. Isn't for a forecast model the heatwave prediction a synonym to a heatwave "detection"?

We agree with the reviewer that for a forecast model, the prediction of heat waves is synonymous to the detection of heat waves. We replaced "prediction" by "detection" in the manuscript.

All metrics used are valuable for the evaluation of the model, as this study shows that an apparently low CRPS and Brier score is not a synonym to the model's ability to separate extreme heat from non-extreme heat.

In this study, we found high CRPS values, greater than 1, which indicate biases in the prediction of the T2m and Tw variables. These biases are considerably reduced when the predictability of extreme heat days is assessed using the Brier score (days with T2m or Tw above the 90th percentile of T2m or Tw respectively).

2. Please replace the expression "the model shows skills" throughout the manuscript.

This expression is supported by the results found with the Gilbert Skill score and hit-rate in Figure 10.

**Introduction:**

1. I was very confused by the fact that the title and the introduction have a lot of material about the seasonal time scale and that the authors claim that they will evaluate this time scale as well. However, the authors use sub-seasonal forecasts going maximum to 6 weeks lead time, so 1.5 months. The authors should better define time scales in the introduction. The sub-seasonal time scale covers 2 weeks to 2 months. The definition can be found in the S2S project here:

https://public.wmo.int/en/resources/bulletin/subseasonal-seasonal-prediction-project-bridging-gap-between-weather-and-climate

Following this comment, I recommend that the authors should not refer on their intro or anywhere else in the manuscript to the seasonal time scale.

This is a good point, we kept only the subseasonal time scale in the manuscript.

2. I really liked the part where the evaluation is done at the city scale. We do not normally see that in sub-seasonal prediction studies and it adds novelty to this study.

Thanks to the reviewer for this appreciation.

**Methodology:**

Line 214: Do the authors mean that they calculate the daily climatological 90th percentile threshold? So, this threshold should vary depending on the time of the year, right? Is the 90-percentile calculated separately for forecast model, reanalysis data, and station data?

Yes, we computed the daily climatological 90th percentile and we agree with the reviewer that the threshold varies depending on the time of year. The 90-percentile is computed separately for the forecast model and the reanalysis data. We clarified it in the manuscript :

"The 90th percentile is calculated for each calendar day of the year and separately for the forecast models and the reanalysis data."

The authors return to the explanation of the 90-percentile definition in the line 233 and state "...daily exceedances of daily values of indicators to the climatological daily threshold " which kind of agrees with the statement above but then in lines 236-238 they state: "Therefore, the climatological daily threshold is chosen to be constant over the whole period; and it is defined as the minimum of the daily climatology thresholds over the study period. This approach allows us to properly assess the severity of a heat wave and its potential human impacts." At the end what is it exactly that the authors do?

After processing to heat wave detection using the 90th percentile climatology threshold, we computed their intensity using this time the minimum of the daily climatological 90 th percentile over the study period.

We clarified this point in the manuscript, as follow:

"The intensity of a heat wave was defined as the sum of the daily exceedances of the indicators values to the climatological threshold during the event. This study is in the framework of the project Agence National de la Recherche STEWARd (STatistical Early WArning systems of weather-related Risks from probabilistic forecasts, over cities in West Africa) project which focuses on the human impacts of climate extremes. Therefore, the climatological daily threshold for the computation of heat wave intensity is chosen to be constant over the whole period. It is defined as the minimum of

the daily climatology thresholds over the study period. This approach allows us to properly assess the severity of a heat wave and its potential human impacts."

Also, as previously mentioned, a lead time dependent percentile should be considered.
This point has been clarified in a previous comment.

Line 226: I thought the authors mentioned before in their manuscript that they assess separately wet and dry heatwaves. Why then your binary vector contains data from extreme values of all temperature variables?
Yes that is actually what we did, the boolean files were created separately for each variable T2m_min, T2m_max and Tw. We clarified it in the manuscript:

"To determine the occurrence and duration of heat waves, we create individual boolean files from the T2m_min, T2m_max and Tw time series at each grid point, containing 1 if hot days and 0 otherwise. This operation is performed on a daily time scale over the period studied. Hot days are days on which the values of T2m_min, T2m_max or Tw are above the daily 90th percentile thresholds. In order to assess the characteristics of heat waves, only hot days belonging to heat wave sequences are considered (Ngoungue et al. 2023). Boolean files are calculated separately for reanalyses and forecasts in order to assess the representation of heat wave occurrence and duration."

Major comment for methodology: The predictability of a model should not be assessed only by comparing to a random chance, as this would not make a strong argument into using this forecast model. The authors should assess predictability by comparing to a reference forecast. For example, the Brier skill score could be calculated separately for two common reference forecasts, being the climatological forecast and the persistence forecast.

In this study, the detection of heat waves in the forecast models is done using two types of metrics : general score (CRPS, Brier) and skill scores (GSS,Hit,FAR). The CRPS and Brier Score are used to have an overview  of the forecast skills,

and the GSS,Hit,FAR are used to assess the skills of the models with respect to climatology.

**Results:**

**Section 3.1:** Calculating forecast climatologies for a sub-seasonal forecasting system that provides forecast over sub-seasonal lead times (maximum 6 weeks) does not mean that the evaluation done here is an evaluation of seasonal forecasts (also the title of the manuscript states that). The authors do not evaluate the seasonal predictability of the forecast system, as this model cannot provide a seasonal prediction. The authors basically provide the climatological biases of the sub-seasonal forecasting systems over the different seasons. Seasonal forecasting means that the forecasting system provides at lead-zero more than 8 weeks forecast, which is completely different with calculating climatologies. Other than that, I find this section very interesting, as we can conclude during which seasons the sub-seasonal forecasts have the largest biases over west Africa.

We changed the time scale according to the reviewer's comment.

**Section 3.2:** As seen in the supplementary material of the previous section, the climatology is lead time dependent and therefore crucial for understanding whether the model predicts an extreme or not. I think that this would significantly change the results on predictability of this study and the authors would also see important differences between lead week 1 and lead week 2. For example, the outcome stated in lines 335-336 ("We have noticed that the skill of the models does not improve necessarily with decreasing lead time.") is related to the choice of climatological distribution.

Here are some presentations from the ECMWF where they explain the calculation of the lead time dependent climatology:
Example for seasonal forecasts:
https://confluence.ecmwf.int/pages/viewpage.action?pageId=174864039
Model climate calculation in page 33:
https://resources.eumetrain.org/data/7/711/high_latitudes_ew_2023_s1c.pdf

Demonstration on how lead time affects EFI verification in page 34. Have a look on known issues on page 35, that is also why in the current study the Thursday initializations should not have been removed as it drastically affects sample size:

https://www.google.com/url?sa=t&rct=j&q=&esrc=s&source=web&cd=&ved=2ahUKEwiT2_GgwKeBAxWUh_0HHZtzD-oQFnoECA4QAQ&url=https%3A%2F%2Fconfluence.ecmwf.int%2Fdownload%2Fattachments%2F70951731%2FForecasting_Extremes_Oct2017.pdf%3Fapi%3Dv2&usg=AOvVaw3YVlL5ZgSZsAzDf-LKflzl&opi=89978449

Thanks to the reviewer for these references.

Line 338: Could the authors explain why the forecast models show better predictive skill in the AT region?

This result can be explained by the fact that T2m signals (min, max) show lower daily variability in the AT region.

**Section 3.3.1:**

Lines 354-355: The authors state "This approach based on a relative threshold (see section 2.4.3) will contribute to partially correct the biases previously found in the models." How does a variable daily threshold correct bias? A daily threshold simply accounts for differences between seasons.

This is a good point. The first evaluation of the forecast models was carried out by calculating the bias and the CRPS score using the raw forecasts. Next, we focused on extreme events (days with a temperature above the 90th percentile), so the raw forecasts were transformed into a Boolean vector containing 1 if the days are warm and 0 if they are not.

We added this information to the manuscript :

"By using a percentile-based approach, we are not focusing on the intensity of extreme events but on the number of events above the threshold. This explains why the Brier score values are quite good compared to the CRPS score."

408-409: The authors state "The forecast models show skills above the reference both for short- and long- term forecasts (Week2, Week5)"

I am wondering which figure shows that there is skill above the reference. Moreover, when the authors say skill, they should specify the metric to which they refer to. Taking into account the pairs of FAR and Hit_rate, Figure 10 shows no skill in detecting/predicting heat waves for none of the seasons/ models/ lead weeks.

In Figure 10, we evaluated the skills of the models on heat wave forecasting with respect to ERA5 reanalysis. As mentioned in the main text, the climatology reference is defined as the probability of having a heat wave in the ERA5 reanalysis over the period 2001-2020. We found higher values of hit-rate and GSS with respect to the reference both at medium and long range forecasts, which indicates that the models show skills above climatology.

410: Do the authors here mean instead of "more skills", "higher skill scores"?

We rephrased this sentence in the manuscript:

"ECMWF presents higher skills than UKMO for short-term forecasts in winter."

Figure 10: Why is only the evaluation for T_min shown? The authors could add Tmax evaluation in Figures 10 and 11.

We added plots showing the evaluation for T_max in Figures 10 and 11.

The authors state "We can infer from this result that nighttime heat waves are more predictable than daytime heat waves." Is there a Tmax figure I missed from which the authors infer this? Moreover, if that is indeed shown on a figure, is the reason for the higher predictability the actual 90th percentile metric used which could be more variable in Tmax and that is why it appears less predictable.

We added a plot showing the evaluation for T_max in Figures 10.

Also, why do the authors not provide the evolution of the mean climatological biases between the forecast models and reanalyses for Tmax in the Appendix as provided for Tmin and Tw in S2 and S3?

We added a plot on the evolution of the mean climatological biases using Tmax.

The authors should avoid the expression "the models show more skills". It should be rephrased to "the models show higher Brier skill in …
Thanks to the reviewer for this suggestion.

415: What is an inter-day variability? Is it the std calculated over all your samples per region?
The inter-day variability represents the daily variability of the indicators (T2m_min, T2m_min and Tw) over the season. Yes, the Std is computed for each indicator over the different regions.
We clarified it in the manuscript :
"The daily variability of T2m is assessed by calculating the standard deviation (std) for each region using ERA5 reanalysis."

416-417 Can the authors explain more this sentence: The low inter-day variability of T2m in the AT region indicates a more stable signal which will lead to favorable conditions for heat wave detection in the models based on a statistical perspective. Wouldn't a stable signal lead to lower probability of extremes, so an even harder prediction of heatwaves?
The low daily variability of T2m in the AT region indicates a more stable signal.
We added this information to the manuscript :
"From a statistical point of view, this will contribute to the occurrence of heat waves, as the probability of having consecutive days above the threshold is higher in a stable signal than in one with high daily variability."

Line 426: By "significantly decreasing" do the authors mean that they have calculated a level of significance? Do maybe the authors mean that the values are strongly decreasing? Or maybe they mean that the values are strongly decreased. In any case, GSS values that go from 0.2 to 0.1 are not strongly decreased, are low overall.
We clarified it in the manuscript:
"We found that the GSS values are low overall lead time and season; the highest values are observed in winter."

432: The use of the word "ability" in this sentence is misleading. Is it an ability to predict events that did not occur?

We replaced the word "ability" in the manuscript:

"An important parameter of a forecast system is its reliability in predicting events. This property is assessed using the False Alarm Ratio: Do the events predicted by the models always occur in the reanalysis?"

The authors provide the T_min variable in Figure 10 and in Figure S15. What is the difference in the 2 plots?

In Figure 10, the evaluation metrics were computed at daily time scale using T2m_min, while in Figure S15, the metrics are computed at weekly time scale.

Why there are no Tw and Tmax plots provided in the appendix?

We added plots on Tw and Tmax in the appendix

In Figure 10 which percentile threshold of ensemble members is used? The authors should add this in the figure caption.

We added the information on the caption.

"Figure 10 : Evaluation of heat waves detection in the forecast models with respect to ERA5 at daily time scale over the period 2001-2020 using T2m_min values for : (a-d) hit-rate, (e-h) FAR ratio and (i-l) GSS. The metrics were computed using the optimized forecasts with the 20% threshold (see section Methods for the optimisation of the ensemble forecasts). The metrics were calculated during the seasons : (a,e,i) winter; (b,f,j) spring; (c,g,k) summer and (d,h,l) autumn. The cyan and black borders of bar plots indicate the metrics obtained when using ECMWF and UKMO respectively. The Y and X axes show the metrics values and the lead times (W2: week2 and W5: week5) respectively. The horizontal red line represents the baseline climatology."

445: How did the authors get to this conclusion: "The forecast models show skills at weekly time scale compared to the baseline climatology." Calculating the Brier skill score using the climatological forecast as reference could support (or not) this statement.

As mentioned in the methods section, heat wave predictability is carried out on daily and weekly time scales. For each time scale, the Hit-rate, FAR and GSS are calculated. The results in Figure S15 show higher values of Hit-rate,

FAR and GSS at weekly time scale compared to the climatology of heat waves in ERA5.

Figure 10 shows that, according to the evaluation done here, the model is over-forecasting heatwaves. This can be even more explicitly shown if the authors plot hit rate against FAR to create the Roc curve. In this curve, the pair of 0.55 hit rate – 0.75 FAR will be below the diagonal. Being below the diagonal indicates no skill to discriminate between events and no-events, with the diagonal indicating random value/no-skill. This plot would disagree with the conclusion drawn by GSS.

This is actually a good point, but the choice of measurements for heatwave forecasting depends on the applications we want to achieve. For example, political decision-makers will be interested in a reliable system that issues correct warnings, and therefore in FAR. Weather center forecasters, on the other hand, will want to predict events correctly and will therefore be interested in the hit rate. Consequently, the use of the Roc curve can be confusing in some applications. A more complete score, commonly used, is the Gilbert skill score, which takes into account hits, false alarms, misses and correct rejections.

FIGURE 11: The markers have very small size, and so are hard to see. Here the authors could easily plot Tmax values as well, which is a very valuable parameter for heatwaves and its important to see its model biases here as well. I would suggest that the authors create one row for each variable (Tmin,Tmax, Tw) and maybe show both models in every sub-panel.

We followed the suggestion of the reviewer.

**Minor comments:**

Line 24 needs some references.

We added the following reference:

"Russo et al., 2017: Humid heat waves at different warming levels, Nature Publishing Group UK London"

Line 37: That is a very big list of references, just to reference daily raw temperature as a variable relative for heatwaves. In the case of simple definitions, it would be helpful to keep the reference lists shorter and target to show the references that are the most relevant.

We followed the suggestion of the reviewer and reduced the list of references.

Line 45-46 "This is usually done using seasonal weather forecast models." This sentence needs references.

We added some references according to the reviewer:

Connor et al., 2008: Integration of seasonal forecasts into early warning systems for climate-sensitive diseases such as malaria and dengue. *Seasonal Forecasts, Climatic Change and Human Health*: Health and Climate, 71-84.

Shukla et al., 2019: Assessing North American multimodel ensemble (NMME) seasonal forecast skill to assist in the early warning of anomalous hydrometeorological events over East Africa. *Clim Dyn* **53**, 7411–7427 (2019).

Montes et al., 2022: Developing a framework for an early warning system of seasonal temperature and rainfall tailored to aquaculture in Bangladesh. *Climate Services*, 26, 100292.

Also, since the study is also for sub-seasonal time scales it would be great if the authors motivate the sub-seasonal time scale and add references with studies connecting early warnings with the sub-seasonal time scale.

One reference I have in mind is:

Osman et al., 2023: Sub-seasonal to decadal predictions in support of climate services DOI: 10.1016/j.cliser.2023.100397

Thanks to the reviewer for this suggestion, we added some references.

Line 51: There is a published study on the predictability of extreme events across the globe: Advances in the Sub-seasonal Prediction of Extreme Events: Relevant Case Studies across the Globe

Domeisen et al., 2022, DOI: https://doi.org/10.1175/BAMS-D-20-0221.1

We added this reference to the manuscript.

Also, overall, for the onset/intensity/duration of European heatwaves at sub-seasonal time scales, where the calculation of lead-time dependent climatology is also explained:
Subseasonal predictability of onset, duration, and intensity of European heat extremes Pyrina et al., 2022, https://doi.org/10.1002/qj.4394
Thanks to the reviewer for this suggestion.

Line 66: "Expert Team on Climate Change Detection and Indices (ETCCDI) database," also needs a reference.
We added a reference to the manuscript.

Line 67: What is the apparent temperature?
We added this information in the manuscript.
"Apparent temperature represents the temperature actually felt by humans, caused by the combined effects of air temperature, relative humidity and wind speed."

65-71: I find this paragraph very long and confusing especially because it is not going to be related to the method that will be used in the current paper. I would either remove it or keep a few sentences about it. Also, the paragraph goes on about what other studies did but there is no connection to what will be done here.
We reduced the paragraph :
"To assess the skills of the models, they used indices from the Expert Team on Climate Change Detection and Indices (ETCCDI, \citep{omondi2014changes}) database based on the apparent temperature and 2-meter temperature. Apparent temperature represents the temperature actually felt by humans, caused by the combined effects of air temperature, relative humidity and wind speed. They found that at the seasonal time scale, the skills of MF5 to reproduce inter-annual anomalies of heat wave duration is limited at the grid point level because of the high spatial variability in the region. At sub-seasonal time, they showed that the skills of the model decrease beyond one week."

The method of Lavaysse et al., 2019 is mentioned later, but it would be nice to also mention a few sentences about their method and why is good and you have followed it here. Also, the Lavaysse et al., 2019 study could be mentioned before, when talking about the heatwave studies, so that the reader connects the study already with heatwave evaluation.

We added this information to the manuscript :

"This method is more robust because the occurrence and duration of heatwaves are assessed directly using daily minimum or maximum temperatures. This involves the computation of evaluation metrics to assess the skills of the forecasts."

Line144: Please provide a citation from a publication or a book. Citations of web pages are not proper for research papers.

We added the following reference:

"Ngoungue et al., 2023 : heat waves monitoring over West Africa cities : uncertainties, characterization and recent trends."

Lines 167-169: I do not understand the meaning of this sentence, was there a problem with the ECMWF output? Why would you choose to not evaluate all available initializations and reduce so much your sample size? A more accurate approach would be to evaluate the hindcasts for another operational model version, such as the model version of 2021, 2022, or even 2019.

We clarified this point in the manuscript :

"We only analyzed the hindcasts produced on Thursday. This is because we firstly want to carry out a multi-model analysis. According to a first investigation on the initialization dates of the hindcasts of different models, we found that most of the models were initialized on the same date as ECMWF (Thursday of each week)."

Line 183: Here it is stated that the authors are "interested in the predictability of heat waves in a global perspective", which is confusing as it may be understood that the evaluation will be done for the whole globe.

We changed this sentence in the manuscript :
"We are aware that these initialization dates are not the same as those of ECMWF, but we are interested in this work on the predictability of heat waves in a broad perspective, not on specific events."

Also, what is stated in the sentence is not a good argument on why using different weekly initializations are comparable. The authors could just say that using ECMWF initializations on Thursdays leads to 4 initializations per month, making the sample size comparable to the UKMO model.

We don't completely agree with the reviewer on this point, because sometimes, when we use ECMWF initialisations on Thursdays, we have 5 initialisations per month (e.g., April, July, September, December).

Figure 3: The authors cannot name a) for T2m_min and a) as well for winter. You could do i) t2m_min, ii) tm_max.

We followed the suggestion of the reviewer.

Figures 3,4: It should be mentioned in the figure that the authors consider all available lead times for this figure.

We followed the suggestion of the reviewer.

Figures 5,6:

1. For each of the variables investigated (Tmin, Tmax, Tw) when are there the stronger climatological biases? How much do they change if we consider a lead time dependent percentile threshold?

Strong climatological biases with Tmin, Tmax and Tw are found during Winter. The computation of the percentile threshold is done separately for each lead time.

2. In many cases the skill increases with lead time, which is not common at all especially comparing forecast weeks 2 and 5. Is the change in skill driven by some particularly well predicted period of extremes at lead week 5? Or maybe it comes by the fact that the authors define a common 90 percentile threshold for all lead times?

This is actually a good point, we thought that this behavior in the models is driven by the atmospheric conditions occurring during week 5 are more stable than week 2 leading to good predictions.

According to the error metrics of figures 5c and 6c, a conclusion would be that the users should trust the summer prediction over the CO region at lead week 5 more than at lead week 2!? This result is even more striking when looking at the winter season and Tw in figure S6.

We do not totally agree with the review, we can conclude based on ERA5 reanalysis that UKMO model is slightly better at Week 5 than Week 2 for T2m_min in the CONT region during summer. We added this information in the manuscript.

Figure 7,8: The authors should explain what the grey values represent in the plots.

We added this information in the captions

Line 322: Actually, there are systematic decreases and increases in biases with lead time in several subplots. For example, the bias is especially pronounced for ERA5 in the region CO (S2-a,f,h,e and in S3-everywhere). Why do some of the biases decrease with lead time? Can the authors explain some of these results or at least mention them?

This is actually a good point. In this study, we do not focus on specific events during the season, but on the evolution of T2m and Tw at daily time scale in the models. The decrease in bias with the lead time from week 1 to week 6 can be explained by the fact that the models reach the predictability horizon too early and also by the presence of more complex atmospheric conditions at short lead time.

Figures 7, 8 Why would the authors indicate a colorbar without units? Here it should be Bias (%)

We added units in the colorbar.

Figure 9 Again here why having a colorbar without units? Especially with your duration definition that is very important. Please change to: Bias (days per year)

We added units in the colorbar.

--The long text in each section is hard to read and makes it hard to return to a specific point when needed. Please separate the long text of each section in paragraphs. See here some tips:

https://www.uvic.ca/learningandteaching/assets/docs/instructors/for-review/Information%20for%20Students/science%20paragraphs.DVG.FINAL.pdf

Thanks to the reviewer for this suggestion.

--Some figures have: "FAR ratio", but ratio is inside the word FAR anyway. Change to: "FAR".

We changed it in the document.

479: How do the authors know that the models have issues with the "spatial evolution of heat waves"? Do they mean spatial variability?

We replaced the expression "spatial evolution of heat waves" by "spatial variability of heat waves".

**Conclusion section:**

This section should be rewritten after the revision of this study.

The section is rewritten according to the reviewer comments and the overall modifications (see new conclusion).

Typos:

Line 99: The references need brackets: Moron et al. (2016); Ngoungue Langue et al. (2023)

We corrected it according to the reviewer.

Line 182: Change "init dates" to "initialization dates"

We corrected it according to the reviewer.

Line 251: Change in the "The skill of the probabilistic models ... are assessed..." to "...is assessed..."

We corrected it according to the reviewer.

Line 236: remove ;

We corrected it according to the reviewer.

---

## Referee Report (RR1)

The authors followed several of the recommendations of the first comments provided for their manuscript. However, still, the manuscript is not in a form for publication as the results should be presented alongside accurate conclusions and in a well-written manner.

Some simple examples include the fact that the introduction does not provide a proper introduction on the topic (a matter raised in the previous review). The authors go on with unnecessary information leaving the reader confused and having to deduct alone what will be relevant for the study. For example, lines 58-66. Why are they relevant to the study? Why do we need to know what Katsafados 2014 found on seasonal time scales and related to blocking over Russia? The current study is on heatwave predictability for the subseasonal timescale over Africa and related to extreme temperature skill assessment. Later there is again an expansion on the method by Omondi et al., 2014 without any relevance. Also, even though there was a previous comment and links provided, still the whole manuscript is full of huge blobs of text without being separated into paragraphs. Finally, the title still contains the word "seasonal" forecasts, which are not included in the current study.

Some major concerns were also raised in the previous review, but I guess they were not convincing. The authors state as their main finding in the abstract, text, and conclusions that the model shows a very good Brier Score, it is therefore able to detect heatwaves for lead weeks 2 to 5. This is completely wrong. The Brier score is biased when the categories evaluated are unbalanced, which is the case here with 10% of your sample size being heatwaves. The rarer an event, the easier it is for the Brier Score to get low values, simply because the forecast model predicts well the majority category ----and not the heatwave category---. See here a screenshot of a presentation by a verification workshop by the ECMWF where they explicitly state that the rarer an event the better can the BS get… which is not surprising, as if you look at the equation it contains no sensitivity regarding the climatological frequency of the event.

[Figure]

Here is the link to the presentation:

https://www.ecmwf.int/sites/default/files/elibrary/2007/15489-verification-probability-forecasts.pdf

I also run a very simple code and I get the same values as your study's Brier score. The model in this code just predicts no heatwaves. Does this mean that my model is able to detect heatwaves?!

```python
import numpy as np
from sklearn.metrics import brier_score_loss

**Create an array that has the ground truth, so 10% is heatwaves which is denoted with 1**
**Create an array with 10 zeros**
truth_array_prob = np.zeros(10)
**Modify one element to be 1**
truth_array_prob[0] = 1
print(truth_array_prob)

**Create an array that represents the values predicted by a model whith no ensemble members predicting the HW**
predicted_array_prob = np.zeros(10)
print(predicted_array_prob)

**Calculate Brier skill**
bs = brier_score_loss(truth_array_prob, predicted_array_prob)
print('Brier score', bs)
print('Low Brier score means nothing for unbalanced categories')
```

```
[1. 0. 0. 0. 0. 0. 0. 0. 0. 0.]
[0. 0. 0. 0. 0. 0. 0. 0. 0. 0.]
Brier score 0.1
Low Brier score means nothing for unbalanced categories
```

I will not comment again on the conclusions related to FAR, GSS, and hit rate.

Nevertheless, the addition of figures and discussion about the CRPS score being similar between lead week 2 and 5 adds nicely to the manuscript. However, I am not sure whether the strange CRPS scores have to do with the way it is calculated. It would be good to add which exact initialisations you consider for the calculation of CRPS, for example in January, and add this info in the Appendix. The fact that I am wondering about this, means that the readers will wonder as well. Therefore, being clear with the calculation will add validity to the results.

Regarding the calculation of the 90$^{th}$ percentile, it was great that the authors added an Appendix Figure. However, the procedure of calculations still needs to be clarified. The authors should keep in mind that the readers and reviewers are not supposed to guess the steps followed for such a calculation. Specifically:

Line 243 states: "For example, using ECMWF, the daily climatological 90th percentile is calculated over the study period separately for hindcasts run every Thursday of the month (see [Fig.S1] in supplement material)."

What do the authors mean by separately? Does the word separately refer to the separation between Monday and Thursday runs?

Then we go to the supplementary material and read the caption of Fig S1: "Fig.S1: Evolution of the 90th daily climatological percentile over AT region using T2m_min ECMWF hindcasts for: (i) the first and (ii) the second hindcast initialization dates from January to December."

1. Do the authors mean by the first and second hindcast initialization dates of each month the first Thursday initialization and the second Thursday initialization of the month? Because one of the 2 first initializations is a Monday and the authors stated in the methods that they do not use Monday runs at all. Here it would help to clarify and to add an example, i.e., for January subpanel (i) refers to e.g. Thursday 03.January and subpanel (ii) refers to e.g. Thursday 10.January.

2. After the authors calculate for each initialization (that is for each model run with 46 days) the DAILY $90^{th}$ percentile of that run considering all years, then how do they apply the $90^{th}$ percentile threshold to assess heatwave occurrence? Do they apply the $90^{th}$ percentile of each day of the 03.January initialization on each corresponding day of the run 03.January.2000, 03.January.2001 etc?

Or do they pool together all Thursday runs initialized in January then calculate a lead time depend climatology based on all daily thresholds and of course separately for lead week 1, lead week 2, etc.? The calculation of percentiles is not straight forward in S2S therefore it would be crucial to provide a schematic explaining clearly and in detail the steps 1 and 2 of the above.

Some typos and other comments:

Line 31: typo in "A Heat wave"
Line 42: I think that here you should replace "min, mean
or max" with the full words, being "minimum, mean, maximum"
Line 4:it should be refer "Heat stress indices refers"
Line 101: "This work is carried out in West Africa" The authors should change "in" to "for", otherwise I think that the current sentence means that they carried out the study while being physically in West Africa.

Line 120: Why is there ";" after stations? I do not think this is correct.
126: Replace this: (NOAA); (in the following, we will use "MERRA" to refer to MERRA-2) as our references for the evaluation
of the forecast models. With this: (NOAA). In the following, we will use "MERRA" to refer to MERRA-2 as our reference for the evaluation of the forecast models.
165-167: grammar is not correct
Line 87: I do not think that the word robust here is correct. The difference is that the Lavaysse method evaluates all heatwave characteristics whereas the other methods are focused on the evaluation of intensity. That does not make the Lavaysse method more robust. I propose to the authors to rephrase into: "This method offers a complete evaluation of heatwave characteristics including not only the evaluation of heatwave intensity, but also of heatwave onset and duration. "

Line 160 is missing a verb.

171: I do not think that this is a valid reason for what the authors chose: "We only analyzed the hindcasts produced on Thursday. This is because we firstly want to carry out a multi-model analysis." The authors analyse 2 models… is this a multi-model analysis?

211: Do the authors here mean by "developed approach" the sampling of the closest grid point to a station? If yes, then this is not a developed approach rather a method followed.

217: should be "occur" not occurred
219: are detected and not is detected
Lines: 307-309 do not need to be repeated. There are already in the introduction.

Legend in Figure 2 does not read nicely. The term maxima/minima temperature shoulb be changed to maximum/minimum temperature. Also, this is not grammatically correct: 'With pool' refers to the pooling of two (or more) … Maybe change into: The term "pooling" refers to …

 Mixing tences: line 251 "The predictability of heat waves is assessed .." versus line 258: "The intensity of a heat wave was defined" That may not seem important, but line 258 could mean that the intensity was defined like that in the cited study and not in the current study.

The naming of section 2.4 as "Metrics" is not proper,as more than half of this section does not desribe metrics but: 2.4.1 Estimation of temperatures at the city scale , 2.4.2 Heat wave detection etc..

---

## Author Response (AR2)

The authors followed several of the recommendations of the first comments provided for their manuscript. However, still, the manuscript is not in a form for publication as the results should be presented alongside accurate conclusions and in a well-written manner.

We improved the quality of the document by taking into account the reviewer's comments. Below are some changes to the introduction and other parts of the document.

Some simple examples include the fact that the introduction does not provide a proper introduction on the topic (a matter raised in the previous review). The authors go on with unnecessary information leaving the reader confused and having to deduct alone what will be relevant for the study. For example, lines 58-66. Why are they relevant to the study? Why do we need to know what Katsafados 2014 found on seasonal time scales and related to blocking over Russia? The current study is on heatwave predictability for the subseasonal timescale over Africa and related to extreme temperature skill assessment. Later there is again an expansion on the method by Omondi et al., 2014 without any relevance.

We changed the introduction according to the reviewer's comments. The new introduction is the following :

[revised manuscript text omitted]

\citet{batte_forecasting_2018} assessed heat waves predictability using T2m and AT anomalies. While this approach provides information about the weather situation for the future days, it cannot provide useful information about the onset and duration of heat waves. In this study, we will adapt the methodology proposed by \citet{lavaysse2019predictability} when assessing the predictability of heat waves over Europe. \textcolor{red}{This method offers a complete evaluation of heatwave characteristics including not only the evaluation of heatwave intensity, but also of heatwave onset and duration.} It involves the computation of evaluation metrics to assess the skills of the models (see Section 2.4.6).

The present study assesses the predictability of heat wave frequency and characteristics in West African cities over the period 2001-2020 using two models part of the S2S project namely, ECMWF and UKMO. \textcolor{red}{To the author's knowledge, this work is the first of its kind in the region and represents a benchmark for future studies.} To achieve our goal, we first analyze the representation of T2m and wet bulb temperature (Tw) in the forecast models with respect to the reanalysis data used as references (see Section 2). Secondly, we evaluate the models on the representation of extreme heat events. Finally, the skill of the models in predicting heat waves is evaluated.

The remainder of this article is organized as follows: in section2, we present the region of study and the data used for this work; the description of the methodology is also provided. Section3 contains the main results of this study following the methodology presented in section 2."

Also, even though there was a previous comment and links provided, still the whole manuscript is full of huge blobs of text without being separated into paragraphs. Finally, the title still contains the word "seasonal" forecasts, which are not included in the current study.

We followed the suggestion of the reviewer, and we splitted the long text in the document into small paragraphs.

Some major concerns were also raised in the previous review, but I guess they were not convincing. The authors state as their main finding in the abstract, text, and conclusions that the model shows a very good Brier Score, it is therefore able to detect heatwaves for lead weeks 2 to 5. This is completely wrong. The Brier score is biased when the categories evaluated are unbalanced, which is the case here with 10% of your sample size being heatwaves. The rarer an event, the easier it is for the Brier Score to get low values, simply because the forecast model predicts well the majority category ----and not the heatwave category---. See here a screenshot of a presentation by a verification workshop by the ECMWF where they explicitly state that the rarer an event the better can the BS get... which is not surprising, as if you look at the equation it contains no sensitivity regarding the climatological frequency of the event. Here is the link to the presentation: https://www.ecmwf.int/sites/default/files/elibrary/2007/15489-verification-probability-forecasts.pdf

I also run a very simple code and I get the same values as your study's Brier score. The model in this code just predicts no heatwaves. Does this mean that my model is able to detect heatwaves?!

Thanks to the reviewer for this good remark and demonstration.

We have changed the conclusion on the Brier score in the document to :

" The Brier score values obtained using ERA5 reanalysis as reference, are very low between 0.05 to 0.175. We could think that the models show skills in forecasting hot days but this is quite difficult to affirm because the brier score is sensitive to the climatological frequency of an event: the more rare an event, the easier it is to get a good BS without having any real skill (https://www.ecmwf.int/sites/default/files/elibrary/2007/15489-verification-probability-forecasts.pdf)."

I will not comment again on the conclusions related to FAR, GSS, and hit rate.

We are aware that the hit-rate and GSS values are very low, but this is not surprising given that heat waves are extremely rare and difficult to predict because the persistence factor comes into play. Furthermore, we also know that the forecasting models underperform in tropical regions due to a poor representation of convective processes in their physical parameterization. Consequently, these scores, which are low but greater than the climatology, are significant for assessing the skill of the models in predicting heat waves in tropical regions which remains a complex task

Nevertheless, the addition of figures and discussion about the CRPS score being similar between lead week 2 and 5 adds nicely to the manuscript. However, I am not sure whether the strange CRPS scores have to do with the way it is calculated. It would be good to add which exact initialisations you consider for the calculation of CRPS, for example in January, and add this info in the Appendix. The fact that I am wondering about this, means that the readers will wonder as well. Therefore, being clear with the calculation will add validity to the results.

We added the following to the manuscript :

"We used for this specific analysis, the UKMO forecasts initialized on the $1^{st}$ of each month". We also added this information to the other legends in the document where it was useful.

Regarding the calculation of the 90th percentile, it was great that the authors added an Appendix Figure. However, the procedure of calculations still needs to be clarified. The authors should keep in mind that the readers and reviewers are not supposed to guess the steps followed for such a calculation. Specifically:

Line 243 states: "For example, using ECMWF, the daily climatological 90th percentile is calculated over the study period separately for hindcasts run every Thursday of the month (see [Fig.S1] in supplement material)."

What do the authors mean by separately? Does the word separately refer to the separation between Monday and Thursday runs?

We clarified this point in the document by changing to :

" The 90th percentile threshold is computed independently for the reanalyses and the hindcasts. As mentionned previously, the hindcasts are run at least once every week for a 6-week duration. For each initialization date within a month and for each lead time, we computed the daily climatological 90th percentile over the study period (2001-2020). We provided in supplement material an illustration of the computation of the 90th percentile threshold using ECMWF hindcasts initialized in January 04th (see Fig.S1 in supplement material). Heat waves are detected independently for each initialization date within a month, using the threshold values computed from this initialization (see Fig.S1_n in supplement material). For example, to detect heat waves in ECMWF hindcasts run on January $04^{th}$, the daily climatological $90^{th}$ percentile of each day of the January $04^{th}$ run is applied to each corresponding day of the runs 04.January.2001, 04.January.2002 to 04.January.2020."

Then we go to the supplementary material and read the caption of Fig S1: "Fig.S1: Evolution of the 90th daily climatological percentile over AT region using T2m_min ECMWF hindcasts for: (i) the first and (ii) the second hindcast initialization dates from January to December."

1. Do the authors mean by the first and second hindcast initialization dates of each month the first Thursday initialization and the second Thursday initialization

of the month? Because one of the 2 first initializations is a Monday and the authors stated in the methods that they do not use Monday runs at all. Here it would help to clarify and to add an example, i.e., for January subpanel (i) refers to e.g. Thursday 03.January and subpanel (ii) refers to e.g. Thursday 10.January.

We clarified this point in the document by changing to :

"Seasonal evolution of the daily climatological 90th percentile threshold over AT region using T2m_min ECMWF hindcasts run on: the first thursday of the month (e.g. Thursday 04th for January) (i) and the second thursday of the month (e.g. Thursday 11th for January)(ii)."

2. After the authors calculate for each initialization (that is for each model run with 46 days) the DAILY 90th percentile of that run considering all years, then how do they apply the 90th percentile threshold to assess heatwave occurrence? Do they apply the 90th percentile of each day of the 03.January initialization on each corresponding day of the run 03.January.2000, 03.January.2001 etc?

Yes, that is actually what we did and we clarified it in the previous comment.

Or do they pool together all Thursday runs initialized in January then calculate a lead time depend climatology based on all daily thresholds and of course separately for lead week 1, lead week 2, etc.? The calculation of percentiles is not straight forward in S2S therefore it would be crucial to provide a schematic explaining clearly and in detail the steps 1 and 2 of the above.

We clarified this point by adding this figure in supplement material in the document.

| Initialization dates | Lead times | | | | | |
|---|---|---|---|---|---|---|
| 04th January 2001 | D1 | D2 | D3 | … | D41 | D42 |
| 04th January 2002 | D1 | D2 | D3 | … | D41 | D42 |
| 04th January 2003 | D1 | D2 | D3 | … | D41 | D42 |
| … | … | … | … | … | … | … |
| 04th January 2019 | D1 | D2 | D3 | … | D41 | D42 |
| 04th January 2020 | D1 | D2 | D3 | … | D41 | D42 |
| Computation of the daily 90th percentile | $Q_{90}(D1)$ | $Q_{90}(D2)$ | $Q_{90}(D3)$ | | $Q_{90}(D41)$ | $Q_{90}(D42)$ |

Figure S1: Computation of the daily climatological 90th percentile threshold for hindcasts initialized on January 04th for all lead times using ECMWF model. We used the same approach to compute the threshold for the other initialization dates and the rest of the months.

Some typos and other comments:

Line 31: typo in "A Heat wave"
We replaced "A Heat wave" by "A heat wave" in the manuscript.

Line 42: I think that here you should replace "min, mean or max" with the full words, being "minimum, mean, maximum"
We replaced "min, mean or max" by "minimum, mean, maximum" in the manuscript.

Line 4:it should be refer "Heat stress indices refers"
We replaced "Heat stress indices refers" by 'it refers'.

Line 101: "This work is carried out in West Africa " The authors should change "in" to "for", otherwise I think that the current sentence means that they carried out the study while being physically in West Africa.

We replaced "This work is carried out in West Africa" by "This work is carried out for West Africa".

Line 120: Why is there ";" after stations? I do not think this is correct.

We removed the ";" in the sentence.

126: Replace this: (NOAA); (in the following, we will use "MERRA" to refer to MERRA-2) as our references for the evaluation of the forecast models. With this: (NOAA). In the following, we will use "MERRA" to refer to MERRA-2 as our reference for the evaluation of the forecast models.

We replaced the previous sentence by :

"In the following, we will use "MERRA" to refer to MERRA-2 which, with ERA5, are the references for the evaluation of the forecast models."

165-167: grammar is not correct

We replaced the previous sentence by the following:

"Hindcasts are forecasts produced for past dates using the most recent version of the forecasting system. They are useful for evaluating the performance of the current version of the model over a past period."

Line 87: I do not think that the word robust here is correct. The difference is that the Lavaysse method evaluates all heatwave characteristics whereas the other methods are focused on the evaluation of intensity. That does not make the Lavaysse method more robust. I propose to the authors to rephrase into: "This method offers a complete evaluation of heatwave characteristics including not only the evaluation of heatwave intensity, but also of heatwave onset and duration. "

We followed the suggestion of the reviewer.

Line 160 is missing a verb.

We replaced the previous sentence by :

The extended-range ECMWF forecast model is running on the Integrated Forecast System (IFS) cycle CY47R3 released on October 10th, 2021.

171: I do not think that this is a valid reason for what the authors chose: "We only analyzed the hindcasts produced on Thursday. This is because we firstly want to carry out a multi-model analysis." The authors analyse 2 models... is this a multi-model analysis?

We clarified this point in the document. We added the following :

"We only analyzed the hindcasts produced on Thursdays. In fact, an initial survey of the initialization dates of the hindcasts revealed that most of the models were initialized on the same date as ECMWF (Thursday of each week). Subsequently, we realized that all those models, with the exception of UKMO, did not cover the study period."

211: Do the authors here mean by "developed approach" the sampling of the closest grid point to a station? If yes, then this is not a developed approach rather a method followed.

We replaced by :

"Following the same approach as in \citet{ngoungue2023heat}, local temperatures ……. "

217: should be "occur" not occurred

We followed the suggestion of the reviewer.

219: are detected and not is detected

We followed the suggestion of the reviewer.

Lines: 307-309 do not need to be repeated. There are already in the introduction.

We removed the expression in the document.

Legend in Figure 2 does not read nicely. The term maxima/minima temperature should be changed to maximum/minimum temperature. Also, this is not grammatically correct: 'With pool' refers to the pooling of two (or more) … Maybe change into: The term "pooling" refers to …

We followed the suggestion of the reviewer.

Mixing tences: line 251 "The predictability of heat waves is assessed .." versus line 258: "The intensity of a heat wave was defined" That may not seem important, but line 258 could mean that the intensity was defined like that in the cited study and not in the current study.

We clarified this point in the document by adding the following:

"The intensity of a heat wave was defined as the sum of the daily exceedances of the indicator values to a daily threshold during the event".

The naming of section 2.4 as "Metrics" is not proper,as more than half of this section does not describe metrics but: 2.4.1 Estimation of temperatures at the city scale , 2.4.2 Heat wave detection etc..

We changed the section name "2.4 Metrics" by "2.4 Methods"

In addition to the points of reviewer 1, I add hereby the comments from the second reviewer:

While I can see that many of my comments were addressed, some of the questions requesting more explanation, e.g., the definitions of the heatwaves themselves and any links to impacts, are not answered. For example, the definition of wet/dry heatwaves – it seems now, day/night heatwaves are changed to dry/wet, and it seems odd that these would be interchangeable.

We clarified the definition of heat wave in the document by adding the following:

" In the present study, two types of heat waves are investigated : dry and wet heat waves. Dry heat waves are mostly driven by incoming solar radiation and occur during the day. The detection of dry heat waves is processed using

maximum values of T2m (T2m_max) as indicator. The most lethal heat waves are due not only to high temperatures but also to the effect of humidity \citep{steadman_assessment_1979,steadman_assessment_1979-1}. Humidity is an important driver of wet heat waves. Wet heat waves are detected using minimum values of T2m (T2m_min) and mean Tw as indicators. T2m_min is also chosen for wet heat waves because relative humidity is higher at night and decreases during the day due the changes in temperature."

The split into the four seasons is not well explained, given that the authors describe the area as having one dry season and one wet season. I can see that effort has been made to include additional information, but still work is needed to improve the structure, narrative, and methodology of the paper.

We clarified this explanation by changing the old paragraph to :

"Heat waves in the Sahel region occur mainly in spring due to the high temperatures in the region at that time (Barbier et al.,2018; Guigma et al., 2020). In this study, the region of interest was extended to the Guinean region in which heat waves are mainly driven by humidity coming from the Atlantic ocean. This advection of humidity over the Guinean region is active during the season. Therefore, the detection of heat waves is performed over the whole season and not just in spring, to cover the wet and dry seasons in the region. "

I am still unconvinced on several points raised in the first review, around the heatwave definitions, the treatment of ensemble members, and the other review articulates very well the concerns and confusion around the verification aspects. The authors appear to also include results regarding heatwave intensity, while also commenting that their threshold approach focuses not intensity but on number of events.

We added some information to clarify the treatment of ensemble members in the text :

"Probabilistic scores exist for evaluating ensemble forecasting systems, but when it comes to the evaluation of specific events such as heat waves, the task

becomes more complex. For example, how do you evaluate a model for which some ensemble members are forecasting a heat wave? To solve this issue, ensemble forecasts are converted into boolean files using threshold values based on the percentage of members correctly forecasting heat wave days. Three threshold values were tested to find the optimal boolean files (see [Table\ref{tab:Table2}])"

For the verification aspects mainly the Brier score, we addressed it in the reviewer 1 comments.

---

## Editor Decision (ED2)

NHESS: Subseasonal forecasts of heat wave in west African cities
Revisions 2 / review 3

While I have always found the topic of this study interesting and important, and I find this version improved over previous versions, I still have some concerns regarding the methodology and results, and structure of the paper. Some aspects of the methodology and data descriptions are still unclear, and the structure, while improved, is still at times confusing. For example, the skill analysis could do with being split into different parts, each clearly stating the aim, scores chosen, and results, as there seem to be multiple parts to the analysis but it is currently confusing to disentangle them. (e.g. there is one part discussing turning probabilistic forecast to deterministic, and then a description of probabilistic skill scores, with no real clear distinction between the different parts of the analysis and which parts of the results/discussion refer to the different aspects of the study). I have provided some more detailed comments and suggestions below which I hope can be of help to the authors in revising the paper.

**Abstract**

The abstract is now very long following the additions. While it is great to see more results highlighted in the abstract, it needs to be reduced in length overall. The authors should also consider what information is clear without the context of the full paper. It is not clear what the 15% / 30% and week 5 / week 2 refer to exactly, so this could be omitted as you really need the context of the full results. For example, I would recommend to rewrite the newly added part (end of line 14 onwards) as follows: "The results suggest that at subseasonal timescales, the forecast models provide a better forecast than climatology, but the hit rate and false alarm rate are sub-optimal and the forecasts may be overestimating the duration of heatwaves, while under-predicting the intensity. Nevertheless, the use of subseasonal forecasts in west African cities can be recommended for prediction of heatwave onset up to two weeks in advance."

**Introduction**

Line 75- 76: The authors state that Batte showed that the Meteo-France model can predict heatwaves up to one week in advance, in the previous paragraph. Here, they say that this approach can't provide information about the onset and duration of heatwaves. Why is that? It is not clear – if a forecasting system can predict a heatwave up to one week in advance, it implies that the onset could be predicted up to one week in advance. Did Batte et al show specifically that onset and duration are not well predicted? If so, that should be highlighted. Perhaps the authors wish to highlight that while studies have shown that forecast models have skill in predicting heatwaves in the short to medium-range, such information can be well supported by longer-range forecasts which may be able to provide even earlier information about the potential onset of a heatwave event, allowing more time to prepare, and that is what is studied here.

*Section 2*

The description of the ECMWF forecasts is still a bit confusing to identify which data exactly have been used by the authors, as several different datasets are described. Additionally, the model version used by the authors is not the most recent version of the ECMWF model, which may be confusing in parts because the authors refer to 'is running', and that hindcasts are using 'the most recent version'. This should be re-worded to indicate that the study uses forecasts from the IFS cycle 47r3, which was operational from October 2021 to June 2023.

*(for info, the current forecast version is 48r1, released in June 2023. This version consists of a 51-member ensemble forecast with a horizontal resolution of around 9km out to 15 days, and another 51-member ensemble forecast out to 46 days at 36km horizontal resolution. The version will be upgraded again later in 2024)*

In the UKMO forecast data description, it states that the initialisation dates are not the same as ECMWF, which does not match with your statement in the ECMWF section that only Thursdays are used to be consistent with the other models. I still do not find a valid reason from the authors for only using the Thursday reforecasts of ECMWF, and not also using the Monday forecasts to increase the sample size. Additionally, the UKMO section is confusing because it states that the hindcasts cover 1993 to 2016, and then starts describing forecasts prior to and since 2017. This needs to be rewritten to be more clear. I would recommend to state the full period of time covered by any UKMO hindcasts, and then break it down into the differences in the forecasts from 2001-2016, and 2017-2021.

Line 203: The authors may wish to state that the land-sea mask of ERA5 has values on a scale of 0 to 1, and what this means, since it is not necessarily standard practice (many masks provide a binary 0 or 1, so it should be clarified what setting a threshold of 0.5 actually means, because it may not be obvious to the reader if they don't work with the ERA5 data and mask themselves)

**Heat wave detection** – is it a regional characteristic, and documented, that humidity is typically lower during the day? Is the humidity always dry enough that it doesn't have an impact on heat stress during the day? This should be clarified, because it is not the case everywhere – typically, it is important also to consider the humidity during the day in combination with the peak temperatures, in order to truly capture the risk of heat stress. The authors may also want to comment on the impact of both together – it can read as though dry and wet heat waves are two different types of event, but really they are just the daytime and nighttime components of one heatwave event, they are not completely distinct, the authors are just disregarding humidity during the day. The authors may wish to comment on the importance of nighttime temperatures and humidity following extreme daytime temperatures, due to the chance (or lack of) for recovery overnight. What is the reference temperature in the relative humidity formula?

Line 240: "The 90$^{th}$ percentile appears to be a sufficient threshold for monitoring heat waves affecting human health." How did the authors determine this? Can the authors

cite the literature where this is determined? Other studies I have seen use the 95th percentile as a relevant threshold for human health.

Line 251 – 253: is 'hot day' an accurate description, when T2m_min deals with nighttime temperatures? Or is a 'hot day' one where **all three** of the variables are all exceeding their respective 90th percentiles during a 24-hour period?

Lines 256 – 264: I am completely confused by this paragraph, sorry – I don't understand any aspects surrounding the definition of heatwave intensity or how it relates to other aspects. This paragraph needs to be rewritten, but I'm afraid I don't understand well enough to offer any advice on it.

Line 279: "the probability of a member predicting a day as being part of a heatwave" – one ensemble member cannot have a probability attached, as the probability itself comes from the full set of ensemble members; I'm not sure what the authors are trying to say here.

Line 278: why is the threshold of 20% used? The authors state it is inspired by another paper, but a prediction of just a 20% probability of a heatwave could be seen as quite low. It would be important to provide a reason for why 20% is decided to be the optimal threshold and how a different threshold may impact the study.

Lines 283 – 289: what is "the whole season" – what part of the year?

Section 2.4.5: it is confusing to read in the previous section that the authors turn the probabilistic forecast into a deterministic one, because it's too complex to assess probabilistic forecasts, and then immediately the next section introduces probabilistic skill scores.

**Results**
Line 413-414: "the main differences... are found with ECMWF" – what are the differences between T2m and Tw that are found?

**Discussion**
Lines 499-503: While the UKMO may place special emphasis on data relevant to the UK and surrounding regions, it does also assimilate a wide range of global and regional data from satellites, ground-based measurements etc. It is unclear how this would influence the surface-ocean interactions. Do the authors mean to discuss differences in ocean coupling between ECMWF and UKMO?

Line 504-505: Is this the correct resolutions for the forecasts that the authors have analysed in this study? The ECMWF extended range is 36km spatial resolution.

**Conclusions**
Lines 551-554: If these models can be useful only up to two weeks in advance, can the authors comment on why they should be considered for use in heatwave forecasting, if other studies have shown that medium-range forecasts are also accurate up to 2 weeks

ahead for forecasting heatwaves? If choosing between a 2-week medium-range forecast at high resolution, and a lower-resolution subseasonal forecasts up to 2 weeks ahead, why choose the subseasonal forecast?

**Figures**
Figure 4: the colour scale used in this figure is inappropriate for the data for 2 reasons: (1) the scale shows only negative values from 0 to -14, it is therefore not appropriate to use a diverging colourscale with a centre point at -7, it should use a continuous colour scale, not diverging. (2) the colour scale includes both green and red together, which applying a filter for colour blindness accessibility indicates it is not accessible for those with colour vision deficiencies. Also, are there really not any positive bias values across the entire region, the bias is only negative everywhere? I would find this surprising. I couldn't find any discussion of the results.

**Minor comments and typos**

Abstract, line 2: remove 'dramatic' (not every heatwave is dramatic, and it is an odd choice of adjective for a heatwave)
*Introduction*
line 29: change "due to urban heat islands" to "due to the urban heat island effect."
line 30: change "by several heat extreme events" to "by several extreme heat events"
line 31: remove "in the shade" (assuming this is based on an official temperature reading, any such reading should always be in the shade)
lines 45-46: the authors mention that EWS integrate shorter and medium-range forecasts, then say 'this window refers to subseasonal timescale from 2 up to 6 weeks' – but short to medium-range timescales would be defined as up to 2 weeks, and then subseasonal as 2 to 6 weeks. Please rewrite these sentences, for example "Many early warning systems integrate short and medium-range foreacsts of potential weather hazards up to two weeks ahead. The subseasonal forecast range, from 2 to 6 weeks ahead, is also highly relevant for actions aimed at mitigating the consequences of extreme heat"
Line 52: "wet/dry temperatures" is not a typical phrase and is confusing. Recommend changing to "Heat waves are often associated with extreme heat, which can be exacerbated by other factors such as humidity levels."
Line 61: "It refers to…" should be "Heat stress indices are used to combine relevant atmospheric variables (such as temperature, humidity, solar and thermal radiation, wind speed) to indicate the impact of the environment on the human body. Examples include…"
Line 64: Universal Thermal Comfort Index should be "University Thermal Climate Index"
Line 71: "ECMWF extended long-range forecasting system" should be "ECMWF extended-range forecasting system"
Line 74: wet and dry heatwaves are introduced here, but are not defined yet. Consider adding short definitions in brackets e.g. "wet heat waves (those combined with high humidity levels) are more predictable than dry heatwaves (those combined with low humidity levels)."
Line 81: add "subseasonal" before predictability to clarify to the reader the forecast range which you will evaluate

Line 108: "reanalysis data have a high resolution compared to observations" – this is an odd statement, as it is strange to compare a gridded product resolution with the observation network, if sparse. There is also something incorrect here – reanalysis products have too **low** of a resolution to be able to detect the highest temperatures (potentially) at point locations, and the urban heat island effect.

---

## Author Response (AR3)

**NHESS: Subseasonal forecasts of heat wave in west African cities  Revisions 2 / review 3**

While I have always found the topic of this study interesting and important, and I find this version improved over previous versions, I still have some concerns regarding the methodology and results, and structure of the paper. Some aspects of the methodology and data descriptions are still unclear, and the structure, while improved, is still at times confusing. For example, the skill analysis could do with being split into different parts, each clearly stating the aim, scores chosen, and results, as there seem to be multiple parts to the analysis but it is currently confusing to disentangle them. (e.g. there is one part discussing turning probabilistic forecast to deterministic, and then a description of probabilistic skill scores, with no real clear distinction between the different parts of the analysis and which parts of the results/discussion refer to the different aspects of the study). I have provided some more detailed comments and suggestions below which I hope can be of help to the authors in revising the paper.

*Thanks to the reviewer for taking his time to review this work once more. We improved the structure of the document by following the recommendations of the reviewer.*

**Abstract**
The abstract is now very long following the additions. While it is great to see more results highlighted in the abstract, it needs to be reduced in length overall. The authors should also consider what information is clear without the context of the full paper. It is not clear what the 15% / 30% and week 5 / week 2 refer to exactly, so this could be omitted as you really need the context of the full results. For example, I would recommend to rewrite the newly added part (end of line 14 onwards) as follows: "The results suggest that at subseasonal timescales, the forecast models provide a better forecast than climatology, but the hit rate and false alarm rate are sub-optimal and the forecasts may be overestimating the duration of heatwaves, while under-predicting the intensity. Nevertheless, the use of subseasonal forecasts in west African cities can be recommended for prediction of heatwave onset up to two weeks in advance."

We followed the suggestion of the reviewer.
We replaced : "Heat waves are one of the most dangerous climatic hazards for human and ecosystem health worldwide. Accurate forecasts of these dramatic events are useful for policy makers and climate services to anticipate risks and develop appropriate responses. Subseasonal forecasts are of great importance for actions to mitigate the human and health consequences of extreme heat. In this perspective, the present study addresses the predictability of heat waves at subseasonal time scales in West African cities over the period 2001-2020. The cities were grouped in three climatic regions based on their climate variability: the continental region 'CO', the Atlantic region 'AT' and the Guinea region 'GU'. Two types of heat waves were analyzed : dry heat waves using maximum values of 2-meter temperature (T2m_max), and wet heat waves using minimum values of 2-meter temperature (T2m_min) and wet bulb temperature (Tw) respectively. Two models that are part of the subseasonal to seasonal (S2S) forecasting project, namely the European Centre for Medium-Range Weather Forecasts (ECMWF) and the United Kingdom Meteorological Office (UKMO) models, were

evaluated using two state-of-the-art reanalysis products, namely the fifth generation ECMWF reanalysis (ERA5) and the Modern-Era Retrospective analysis for Research and Application. The performance of the models in predicting heat waves is assessed through the computation of categorical metrics such as the hit-rate, the Gilbert Skill Score (GSS) and the false alarm ratio (FAR). The forecast models show low skills in predicting heat wave days especially for medium-range forecasts (two weeks ahead) in the three climatic regions. The hit-rate and GSS values are very weak, while the FAR are higher. On average, only 15% / 30% of the predicted heat wave days are actually observed for Week 5 / Week 2, respectively. This suggests that the models overestimate the duration of heat waves with respect to ERA5. Nevertheless, the hit rate and GSS values are superior to the climatology and significant for predicting heat waves in tropical regions, which remains a complex task. Regarding these results, we can recommend the use of subseasonal forecasts in African cities to predict the onset or some days during the heat waves period up to two weeks in advance. Such informations are very useful for the population, hospitals and decision-makers in order to develop some adaptation strategies to reduce the impacts of heat waves in the region." by

*"Heat waves are one of the most dangerous climatic hazards for human and ecosystem health worldwide. Accurate forecasts of these events are useful for policy makers and climate services to anticipate risks and develop appropriate responses. Subseasonal forecasts are of great importance for actions to mitigate the human and health consequences of extreme heat. In this perspective, the present study addresses the predictability of heat waves at subseasonal timescales in West African cities over the period 2001-2020. The cities were grouped in three climatic regions based on their climate variability: the continental region 'CO', the Atlantic region 'AT' and the Guinea region 'GU'. Two types of heat waves were analyzed : dry heat waves using 2-meter temperature and wet heat waves using average wet bulb temperature. Two models that are part of the subseasonal to seasonal forecasting project, namely the European Centre for Medium-Range Weather Forecasts (ECMWF) and the United Kingdom Meteorological Office models, were evaluated using two state-of-the-art reanalysis products, namely the fifth generation ECMWF reanalysis and the Modern-Era Retrospective analysis for Research and Application. The performance of the models in predicting heat waves is assessed through the computation of categorical metrics such as the hit-rate, the Gilbert Skill Score and the False Alarm Ratio. The results suggest that at subseasonal timescales, the forecast models provide a better forecast than climatology, but the hit rate and false alarm rate are sub-optimal and the forecasts may be overestimating the duration of heatwaves, while under- predicting the intensity. Nevertheless, the use of subseasonal forecasts in west African cities can be recommended for prediction of heatwave onset up to two weeks in advance."*

**Introduction**

Line 75-76: The authors state that Batte showed that the Meteo-France model can predict heatwaves up to one week in advance, in the previous paragraph. Here, they say that this approach can't provide information about the onset and duration of heatwaves. Why is that? It is not clear – if a forecasting system can predict a heatwave up to one week in advance, it implies that the onset could be predicted up to one week in advance. Did Batte et al show specifically that onset and duration are not well predicted? If so, that should be highlighted. Perhaps the authors wish to highlight that while studies have shown that forecast models have skill in predicting heatwaves in the short to medium-range, such information can be well supported by longer-range forecasts which may be able to provide even earlier information about the potential onset of a heatwave event, allowing more time to prepare, and that is what is studied here.

*We clarified this point in the document, we replaced :*
"Batté et al. (2018) assessed heat waves predictability using T2m and AT anomalies. While this approach provides information about the weather situation for the future days, it cannot provide useful information about the onset and duration of heat waves." by

*" Batté et al. (2018) assessed heat waves predictability using T2m and AT anomalies. While this approach provides information about the evolution of T2m and AT for the subsequent days, it is not sufficient to determine heat waves characteristics such as the duration and frequency. "*

**Section 2**
The description of the ECMWF forecasts is still a bit confusing to identify which data exactly have been used by the authors, as several different datasets are described. Additionally, the model version used by the authors is not the most recent version of the ECMWF model, which may be confusing in parts because the authors refer to 'is running', and that hindcasts are using 'the most recent version'. This should be re-worded to indicate that the study uses forecasts from the IFS cycle 47r3, which was operational from October 2021 to June 2023. (for info, the current forecast version is 48r1, released in June 2023. This version consists of a 51-member ensemble forecast with a horizontal resolution of around 9km out to 15 days, and another 51-member ensemble forecast out to 46 days at 36km horizontal resolution. The version will be upgraded again later in 2024)

*We clarified this point in the document, we changed :*
*"The extended-range ECMWF forecast model is running on the Integrated Forecast System (IFS) cycle CY47R3 released on October 10th, 2021. The native spatial resolution of the ECMWF model is Tco639 L137 (about 16 km) up to day 15 and Tco319 (about 32 km) after day 15, but the downloaded data are interpolated to a regular 0.25°x 0.25° latitude/longitude grid to match the resolution of ERA5 for evaluation. It contains 91 sigma levels from the surface to 80 km. ECMWF provides two types of outputs for the S2S program: real-time forecasts and reforecasts called "hindcasts". Real-time forecasts are forecasts for the coming days. Hindcasts are forecasts produced for past dates using the most recent version of the forecasting system, and allow analysis of how the current system would have performed, alongside a consistent dataset covering a longer time period for evaluation.*

*ECMWF extended-range real-time forecasts are run with 51 ensemble members (50 perturbed and 1 unperturbed), while hindcasts are run with 11 members. In this study, we focus on hindcasts only. ECMWF extended-range hindcasts are produced twice a week, on Monday and Thursday at 00Z. This means that for each week a new set of hindcasts is produced to calibrate the real-time ensemble forecasts for Monday and Thursday of the following week using the latest version of the IFS. We only analyzed the hindcasts produced on Thursdays. In fact, an initial survey of the initialization dates of the hindcasts revealed that most of the models were initialized on the same date as ECMWF (Thursday of each week). Subsequently, we realized that all those models, with the exception of UKMO, did not cover the study period. It is no longer possible to carry out a multi-model evaluation and we have therefore limited the evaluation to ECMWF and UKMO. The 11-member ensemble hindcasts start on the same day and month as the real-time forecast, but covering the last 20 years. In our case, the forecast year is 2021 and we focus on the previous 20 years from that date, and the hindcasts run from 0-46 days. The variables of interest in the ECMWF S2S are T2m(max,min) over the last 6 hours, daily average T2m and d2m from which the daily average Tw was derived. The data are open access and available on the S2S project website (https://apps.ecmwf.int/datasets/ data/s2s-realtime-instantaneous-accum-ecmf/levtype=sfc/type=cf/).”*

 *to*

*"The extended-range ECMWF forecast model used in this work is run on the Integrated Forecast System (IFS) cycle CY47R3 released on October 10th, 2021. The native spatial resolution of the extended-range ECMWF model is Tco639 L137 (about 16 km) up to day 15 and Tco319 (about 32 km) after day 15, but the downloaded data are interpolated to a regular 0.25°x 0.25° latitude/longitude grid to match the resolution of ERA5 for evaluation. It contains 137 sigma levels from the surface to 80 km. ECMWF provides two types of outputs for the S2S program: real-time forecasts and reforecasts or "hindcasts". Real-time forecasts are forecasts for the coming days. Hindcasts are forecasts produced for past dates and allow analysis of how the current system would have performed, alongside a consistent dataset covering a longer time period for evaluation. They are useful for the calibration of the model and post treatment analyses.*

*The ECMWF extended-range hindcasts are run with 11 members  (10 perturbed and 1 unperturbed). ECMWF extended-range hindcasts are produced twice a week, on Monday and Thursday at 00Z. This means that for each week a new set of hindcasts is produced to calibrate the real-time ensemble forecasts for Monday and Thursday of the following week. We have only analyzed the hindcasts produced on Thursdays, as a preliminary investigation into the initialization dates of the hindcasts showed that most models were launched on Thursdays. In the database, some models use fixed dates of the month (01st, 09th, 17th for example) and others, specific days of the week (Monday, Thursday for example), which generate some difficulties to handle. Most of the models do not cover the period under study, except for UKMO which is available but uses fixed initialisation dates of the month (see section below for more details). It is therefore no longer possible to carry out a multi-model evaluation as we had*

*planned and we have limited the evaluation to ECMWF and UKMO. The ECWMF hindcasts produced on Monday cover the same period of the ones produced on Thursday. Thus, using the Monday hindcasts we have no significant changes in the frequency of the events detected. The 11-member ensemble hindcasts start on the same day and month as the real-time forecast, but covering the last 20 years. In our case, the forecast year is 2021 and we focus on the previous 20 years from that date, and the hindcasts run from 0-46 days. The variables of interest in the ECMWF S2S are T2m(max,min) over the last 6 hours, daily average T2m and d2m from which the daily average Tw was derived. The data are open access and available on the S2S project website (https://apps.ecmwf.int/datasets/data/s2s-realtime-instantaneous-accum-ecmf/levtype=sfc/type=cf/)."*

In the UKMO forecast data description, it states that the initialisation dates are not the same as ECMWF, which does not match with your statement in the ECMWF section that only Thursdays are used to be consistent with the other models. I still do not find a valid reason from the authors for only using the Thursday reforecasts of ECMWF, and not also using the Monday forecasts to increase the sample size. Additionally, the UKMO section is confusing because it states that the hindcasts cover 1993 to 2016, and then starts describing forecasts prior to and since 2017. This needs to be rewritten to be more clear. I would recommend to state the full period of time covered by any UKMO hindcasts, and then break it down into the differences in the forecasts from 2001-2016, and 2017-2021.

*We clarified this point in the document, we changed :*
"The UKMO model runs on the HadGEM3 GC2.0 model which simulates the uncertainties of the initial conditions using a lagged initialisation and the uncertainties of the model using a stochastic scheme. The native spatial resolution of the UKMO model is N216: 0.83°x0.56° (about 60 km at mid-latitudes). It contains 85 vertical levels from the surface to 85 km and 4 soil levels: level 1 (0 - 0.1 m), level 2 (0.1 - 0.35 m), level 3 (0.35 - 1 m) and level 4 (1- 3 m). Similar to ECMWF, UKMO provides to the S2S program real-time forecasts and hindcasts. The UKMO real-time forecast consists of a set of 4 members (3 perturbed members and 1 control member) run daily for a period of 60 days. The UKMO hindcasts are produced 4 times per month, on the 1 st , 9 th , 17 th and 25 th , and cover a 24-year period from 1993 to 2016. We are aware that these initialization dates are not the same as those of ECMWF, but we are interested in this work on the predictability of heat waves in a broad perspective, not on specific events. Prior to 2017, specifically on March 25 th , the UKMO ensemble hindcasts were composed of 3 members per cycle (2 perturbed and 1 control). Since 2017, the number of members has increased from 3 to 7 (6 perturbed and 1 control). Our target period is going from January 2001 to February 2021, and as mentioned earlier, the UKMO hindcasts are not available after the year 2016. To solve this problem and get more robust statistical results, we recompose the products to obtain a new composite that covers the whole target period. "

to

*"The UKMO model runs on the HadGEM3 GC2.0 model which simulates the uncertainties of the initial conditions using a lagged initialisation and the uncertainties of the model using a stochastic scheme. The native spatial resolution of the UKMO model is N216: 0.83°x0.56° (about 60 km at mid-latitudes), but the downloaded data are extrapolated to a regular latitude/ longitude grid of 0.25° x 0.25°. It contains 85 vertical levels from the surface to 85 km and 4 soil levels: level 1 (0 - 0.1 m), level 2 (0.1 - 0.35 m), level 3 (0.35 - 1 m) and level 4 (1- 3 m). Similar to ECMWF, UKMO provides to the S2S program real-time forecasts and hindcasts. The UKMO real-time forecast consists of a set of 4 members (3 perturbed members and 1 control member) run daily for a period of 60 days. The UKMO hindcasts analyzed here are run using the model version released in 2023 which produces 7 members per cycle (6 perturbed and 1 control) for the period 1993-2016 (no UKMO hindcasts available after 2016). They are produced 4 times per month, on the 1 st , 9 th , 17 th and 25 th. We are aware that these initialisation dates are not the same as those of ECMWF, but we are interested in this work on the predictability of heat waves in a broad perspective, not on specific events. Our target period is going from January 2001 to February 2021, and as mentioned earlier, the UKMO hindcasts are not available after the year 2016. To solve this problem and get more robust statistical results, we recomposed the products to obtain a new composite that covers the whole target period. "*

Line 203: The authors may wish to state that the land-sea mask of ERA5 has values on a scale of 0 to 1, and what this means, since it is not necessarily standard practice (many masks provide a binary 0 or 1, so it should be clarified what setting a threshold of 0.5 actually means, because it may not be obvious to the reader if they don't work with the ERA5 data and mask themselves)

*We clarified this point in the text, by adding the following in section 2.4.1 :*

*" Following the same approach as in Ngoungue Langue et al. (2023), local temperatures over the cities were derived from the reanalysis using the reanalysis grid point closest to the station that satisfies a land-sea mask (lsm) of at least 0.5 ([Table2] shows the lsm values for all the cities considered in this study, the same technique was applied for the forecast models). The lsm indicates the proportion of land contained in a grid point. If the lsm is less than 0.5, this means that the grid point is mainly covered by the ocean, while a lsm greater than 0.5 implies more land coverage. "*

Heat wave detection – is it a regional characteristic, and documented, that humidity is typically lower during the day? Is the humidity always dry enough that it doesn't have an impact on heat stress during the day? This should be clarified, because it is not the case everywhere – typically, it is important also to consider the humidity during the day in combination with the peak temperatures, in order to truly capture the risk of heat stress. The authors may also want to comment on the impact of both together – it can read as though dry and wet heat waves are two different types of event, but really they are just the daytime and nighttime components of one heatwave event, they are not completely distinct, the authors are just disregarding humidity during the day. The authors may wish to comment on the importance of nighttime temperatures

and humidity following extreme daytime temperatures, due to the chance (or lack of) for recovery overnight.

*We clarified this point in the text, we changed:*

"In the present study, two types of heat waves are investigated : dry and wet heat waves. Dry heat waves are mostly driven by incoming solar radiation and occur during the day. The detection of dry heat waves is processed using maximum values of T2m (T2m_max) as indicator. The most lethal heat waves are due not only to high temperatures but also to the effect of humidity (Steadman, 1979a, b). Humidity is an important driver of wet heat waves. Wet heat waves are detected using minimum values of T2m (T2m_min) and mean Tw as indicators. T2m_min is also chosen for wet heat waves because relative humidity is higher at night and decreases during the day due the changes in temperature." by

*"In the present study, two types of heat waves are investigated : dry and wet heat waves. Dry heat waves are associated with high temperatures and low humidity conditions. The detection of dry heat waves is processed using 2-meter temperature (T2m) as an indicator. We distinguished two categories of dry heat waves : those that occur during the daytime and are detected using maximum values of T2m (T2m_max), and those that occur during the night and are detected using minimum values of T2m (T2m_min). Concomitant heat waves, those that occur during daytime and the night, are extremely dangerous because the body does not have the time to recover from the daytime heat waves during the night (Li et al.,2017; Wang et al., 2020). The most lethal heat waves are due not only to high temperatures but also to the effect of humidity (Steadman,1979a, b ; Heo et al.,2019 ; Yu et al., 2021). Humidity is an important driver of wet heat waves. The combination of high heat and humidity can compromise the human body's main cooling mechanism: transpiration. The evaporation of sweat from skin cools our bodies, but higher humidity levels limit evaporative cooling. As a result, we can suffer heat stress and illness, and the consequences can even be fatal. Wet heat waves are detected here using average Tw as an indicator (Yu et al. 2021)".*

What is the reference temperature in the relative humidity formula?
*We added this information in the text : "the reference temperature is the thermodynamic temperature of the triple point of water"*

Line 240: "The 90th percentile appears to be a sufficient threshold for monitoring heat waves affecting human health." How did the authors determine this? Can the authors cite the literature where this is determined? Other studies I have seen use the 95th percentile as a relevant threshold for human health.
*We added some references supporting the choice the 90th percentile for the monitoring of heat waves affecting human health : " Perkins and Alexander 2003 ; Déqué et al., 2007; Fontaine et al., 2013; Russo et al., 2014 ; Oueslati et al., 2017; Barbier et al.,2018; Lavaysse et al., 2019 "*

Line 251 – 253: is 'hot day' an accurate description, when T2m_min deals with nighttime temperatures? Or is a 'hot day' one where all three of the variables are all exceeding their respective 90th percentiles during a 24-hour period?

*We aware that this can lead to confusion, but to simplify text, we have defined it as follows :*
*"Hot days are days belonging to heat waves with the values of one of the 3 indicators (T2m_min, T2m_max, Tw) above their 90th percentile. "*

Lines 256 – 264: I am completely confused by this paragraph, sorry – I don't understand any aspects surrounding the definition of heatwave intensity or how it relates to other aspects. This paragraph needs to be rewritten, but I'm afraid I don't understand well enough to offer any advice on it.

Thank you for your comment and sorry for the confusion. We clarified this point by changing :

"The intensity of a heat wave was defined as the sum of the daily exceedances of the indicator values to a daily threshold during the event. We would like to point out that the threshold values used to compute heat waves intensity are not the same for heat waves detection. In the scope of this study, part of the project Agence National de la Recherche STEWARd (STatistical Early WArning systems of weather-related Risks from probabilistic forecasts, over cities in West Africa), we are interested in heat waves, which can be harmful to human health. As mentionned previously and based on previous works, the 90 th percentile is a suitable threshold to detect heat waves affecting human health. In order to assess the severity of the events using the same reference, we have chosen a constant daily threshold for the computation of heat waves intensity. It is defined as the minimum of the daily climatology 90 th percentile over the study period. Therefore, most dangerous heat waves will have higher intensity values."

to

*"The intensity of a heat wave was defined as the sum of the daily exceedances of the indicators (T2m_min, T2m_max and Tw) values to a constant threshold for the duration of the event. It should be emphasized that this constant threshold is not used for heat waves detection, but only to compute their intensity. In this study, which is part of the project Agence National de la Recherche STEWARd (STatistical Early WArning systems of weather-related Risks from probabilistic forecasts, over cities in West Africa), we are interested in heat waves, which can be harmful to human health. Therefore, the constant threshold mentioned above is defined as the minimum of the daily climatological 90th percentile over the study period. The choice of a constant threshold for the computation of heat waves intensity is very important because it takes into account the seasonal cycle. This makes it possible to assess the severity of the events using the same reference. In fact, the most dangerous heat waves will have higher intensity values."*

Line 279: "the probability of a member predicting a day as being part of a heatwave" – one ensemble member cannot have a probability attached, as the probability itself comes from the full set of ensemble members; I'm not sure what the authors are trying to say here.

*We clarified this point in the text by replacing :*

*" Given a threshold of 20% (i.e. at least 20% of ensemble members predict heat wave days), our methodology consists of computing the probability of a member predicting a day as being part of a heat wave. If the probability is greater than the threshold value, the day will be counted as "1" and "0" otherwise" by*

*"Assuming a threshold of 20% for example (i.e. that at least 20% of ensemble system members are forecasting heatwave days), we determined the probability of the ensemble forecast systems predicting a day as being part of a heatwave. If the probability is greater than the threshold value, the day will be counted as "1" and "0" otherwise."*

Line 278: why is the threshold of 20% used? The authors state it is inspired by another paper, but a prediction of just a 20% probability of a heatwave could be seen as quite low. It would be important to provide a reason for why 20% is decided to be the optimal threshold and how a different threshold may impact the study.

*We clarified it in the previous comment (" line 279"), at this stage in the study the "20%" threshold is an example of threshold value to explain the methodology. Nevertheless, in the document, we provided some analyses obtained using different threshold values.*

Lines 283 – 289: what is "the whole season" – what part of the year?
*We replaced the "whole season" by " from January to December"*

**Section 2.4.5**: it is confusing to read in the previous section that the authors turn the probabilistic forecast into a deterministic one, because it's too complex to assess probabilistic forecasts, and then immediately the next section introduces probabilistic skill scores.

*We reorganized the sections 2.4.4 and 2.4.5 in the document as follows :*

*"2.4.4 Evaluation of probabilistic forecasts*

*2.4.5 From probabilistic to deterministic forecasts*

*2.4.6 Evaluation of deterministic forecasts*
*"*

**Results**
Line 413-414: "the main differences… are found with ECMWF" – what are the differences between T2m and Tw that are found?

*We added these differences in the text :*
*"In winter, for example, there is an underestimation of the duration of heat waves associated with Tw in the ECMWF over the Guinea region, while an overestimation is observed with T2m_min and T2m_max."*

**Discussion**
Lines 499-503: While the UKMO may place special emphasis on data relevant to the UK and surrounding regions, it does also assimilate a wide range of global and regional data from satellites, ground-based measurements etc. It is unclear how this would influence the surface-ocean interactions. Do the authors mean to discuss differences in ocean coupling between ECMWF and UKMO?
*We clarified this point in the text, we changed :*

*"Secondly, the models used the same data assimilation methods (4D-Var) for control analyses but the data and initial conditions are completely different. ECMWF assimilates a wide range of global and regional observational data, including satellite,radar and ground-based measurements. The UKMO focuses on observation data relevant to the United Kingdom and surrounding regions. These differences may influence the representation of surface-ocean interactions in the models. The differences observed in the representation of T2m_min over the Atlantic ocean [Fig.3] can result from these types configurations."  to*

*"Secondly, the models use the same data assimilation methods (4D-Var) for control analyses, but the data and initial conditions are completely different. ECMWF assimilates a wide range of global and regional observational data, including satellite,radar and ground-based measurements. The UKMO focuses on observation data relevant to the United Kingdom and surrounding regions. ECMWF atmospheric model is coupling with the NEMO3.4.1 ocean model, while UKMO uses the NEMO3.6 ocean model. The two systems are not using the same atmospheric and ocean models, which implies different parameterisations. The differences observed in the representation of T2m_min over the Atlantic ocean [Fig.3] could result from representation of surface-ocean interactions in the models."*

Line 504-505: Is this the correct resolutions for the forecasts that the authors have analysed in this study? The ECMWF extended range is 36km spatial resolution.
*We added the correct resolution in the text :*
*"Thirdly, the spatial resolution of the atmospheric component of the two models : ECMWF has a higher spatial resolution than UKMO (0.32°x0.32° Vs 0.83°x0.56°), which means that it can capture local-scale variability or atmospheric processes and provide more accurate forecasts for specific regions. (https://confluence.ecmwf.int/display/S2S/ECMWF+Model)"*

**Conclusions**
Lines 551-554: If these models can be useful only up to two weeks in advance, can the authors comment on why they should be considered for use in heatwave forecasting, if other studies have shown that medium-range forecasts are also accurate up to 2 weeks ahead for forecasting

heatwaves? If choosing between a 2-week medium-range forecast at high resolution, and a lower-resolution subseasonal forecasts up to 2 weeks ahead, why choose the subseasonal forecast?

*In this study, we evaluated the skills of the subseasonal forecasts with a focus on medium (week2) and long (week5) time scales. In general, we found that the subseasonal models perform better at week2 than at week5. This means that sub-seasonal forecasts are able to predict heat waves at least two weeks in advance. Subseasonal models offer a wide forecasting window and make it possible to detect an event long time in advance compared to medium range forecasts. We clarified this point in the text by adding the following :*

*"Regarding these results, we can recommend the use of subseasonal forecasts in African cities to predict the onset and frequency of heat waves, and some days during the heat waves period at least to two weeks in advance, but as far as their intensity is concerned, it is still challenging."*

**Figures**
Figure 4: the colour scale used in this figure is inappropriate for the data for 2 reasons: (1) the scale shows only negative values from 0 to -14, it is therefore not appropriate to use a diverging colourscale with a centre point at -7, it should use a continuous colour scale, not diverging. (2) the colour scale includes both green and red together, which applying a filter for colour blindness accessibility indicates it is not accessible for those with colour vision deficiencies. Also, are there really not any positive bias values across the entire region, the bias is only negative everywhere? I would find this surprising. I couldn't find any discussion of the results. *We changed the color scale according to the reviewer comment.*

[Figure]

Figure 4 : Spatial variability of the climatological bias between the forecast models ensemble mean and ERA5 reanalysis over the period 2001-2020 for Tw during the seasons : (a,e) winter; (b,f) spring; (c,g) summer and (d,h) autumn. The bias is computed as the difference between the forecast models and ERA5 considering all the lead times. The color indicates the bias values in degrees Kelvin. The X and Y axes represent the longitude and latitude respectively.

**Minor comments and typos**

Abstract, line 2: remove 'dramatic' (not every heatwave is dramatic, and it is an odd choice of adjective for a heatwave)
*We removed "dramatic" in the text*

Introduction
line 29: change "due to urban heat islands" to "due to the urban heat island effect."
*We followed the recommendation of the reviewer.*

line 30: change "by several heat extreme events" to "by several extreme heat events"
*We followed the recommendation of the reviewer.*

line 31: remove "in the shade" (assuming this is based on an official temperature reading, any such reading should always be in the shade)
*We followed the recommendation of the reviewer.*

lines 45-46: the authors mention that EWS integrate shorter and medium-range forecasts, then say 'this window refers to subseasonal timescale from 2 up to 6 weeks' – but short to medium-range timescales would be defined as up to 2 weeks, and then subseasonal as 2 to 6 weeks. Please rewrite these sentences, for example "Many early warning systems integrate short and medium-range forecasts of potential weather hazards up to two weeks ahead. The subseasonal forecast range, from 2 to 6 weeks ahead, is also highly relevant for actions aimed at mitigating the consequences of extreme heat"
*Thanks to the reviewer for this suggestion, we integrated it in the text.*

Line 52: "wet/dry temperatures" is not a typical phrase and is confusing. Recommend changing to "Heat waves are often associated with extreme heat, which can be exacerbated by other factors such as humidity levels."
*We followed the recommendation of the reviewer.*

Line 61: "It refers to…" should be "Heat stress indices are used to combine relevant atmospheric variables (such as temperature, humidity, solar and thermal radiation, wind speed) to indicate the impact of the environment on the human body. Examples include…"
*Thanks to the reviewer for this suggestion, we integrated it in the text.*

Line 64: Universal Thermal Comfort Index should be "University Thermal Climate Index"
*We followed the recommendation of the reviewer.*

Line 71: "ECMWF extended long-range forecasting system" should be "ECMWF extended-range forecasting system"
*We followed the recommendation of the reviewer.*

Line 74: wet and dry heatwaves are introduced here, but are not defined yet. Consider adding short definitions in brackets e.g. "wet heat waves (those combined with high humidity levels) are more predictable than dry heatwaves (those combined with low humidity levels)."
*We clarified this point in the document by adding these short definitions in brackets.*

Line 81: add "subseasonal" before predictability to clarify to the reader the forecast range which you will evaluate
*We followed the recommendation of the reviewer.*

Line 108: "reanalysis data have a high resolution compared to observations" – this is an odd statement, as it is strange to compare a gridded product resolution with the observation network, if sparse. There is also something incorrect here – reanalysis products have too low of a resolution to be able to detect the highest temperatures (potentially) at point locations, and the urban heat island effect.
*We clarified this point in the text, we changed :*

"We are aware that reanalysis data have a high resolution compared to local observations and therefore cannot represent the urban heat island effect which exacerbates heat stress during heat waves. The resolution of the reanalyses makes it impossible to detect the highest temperatures at specific locations."   by

*"We are aware that reanalysis data have a low resolution for detecting the highest temperatures at a point location, as well as the urban heat island effect."*